# SLINGSHOT PERTURBATION TO LEARNING IN MONO-TONE GAMES

## ABSTRACT

This paper addresses the problem of learning Nash equilibria in *monotone games* where the gradient of the payoff functions is monotone in the strategy profile space, potentially containing additive noise. The optimistic family of learning algorithms, exemplified by optimistic Follow-the-Regularized-Leader and optimistic Mirror Descent, successfully achieves last-iterate convergence in scenarios devoid of noise, leading the dynamics to a Nash equilibrium. A recent emerging trend underscores the promise of the perturbation approach, where payoff functions are perturbed based on the distance from an anchoring, or *slingshot*, strategy. In response, we first establish a unified framework for learning equilibria in monotone games, accommodating both full and noisy feedback. Second, we construct the convergence rates toward an approximated equilibrium, irrespective of noise presence. Thirdly, we introduce a twist by updating the slingshot strategy, anchoring the current strategy at finite intervals. This innovation empowers us to identify the Nash equilibrium of the underlying game with guaranteed rates. The proposed framework is all-encompassing, integrating existing payoff-perturbed algorithms. Finally, empirical demonstrations affirm that our algorithms, grounded in this framework, exhibit significantly accelerated convergence.

## 1 INTRODUCTION

This study delves a variant of online learning algorithms, such as the Follow the Regularized Leader (FTRL) (McMahan, 2017) and mirror descent (MD) (Nemirovskij & Yudin, 1983; Beck & Teboulle, 2003), within the realm of monotone games. In these games, the gradient of the payoff functions exhibits monotonicity concerning the strategy profile space. This encompasses diverse games, including Cournot competition (Bravo et al., 2018), $\lambda$-cocoercive games (Lin et al., 2020), concave-convex games, and zero-sum polymatrix games (Cai & Daskalakis, 2011; Cai et al., 2016). Due to their extensive applicability, various learning algorithms have been developed and scrutinized to compute a Nash equilibrium efficiently.

Traditionally, no-regret learning algorithms, such as FTRL, are widely used because their averaged strategy profile over iterations converges to an equilibrium (referred to as *average-iterate convergence*). Nevertheless, research has shown that the actual trajectory of update strategy profiles fails to converge even in two-player zero-sum games, a specific class within monotone games (Mertikopoulos et al., 2018; Bailey & Piliouras, 2018). On the contrary, optimistic learning algorithms, incorporating recency bias, have shown success. The day-to-day strategy profile converges to a Nash equilibrium (Daskalakis et al., 2018; Daskalakis & Panageas, 2019; Mertikopoulos et al., 2019; Wei et al., 2021), termed *last-iterate convergence*.

However, the optimistic approach faces challenges with feedback contaminated by some noise. Typically, each agent updates his or her strategy according to the perfect gradient feedback of the payoff function at each iteration, denoted as *full feedback*. In a more realistic scenario, noise might distort this feedback. With *noisy feedback*, optimistic learning algorithms perform suboptimally. For instance, Abe et al. (2023) empirically demonstrated that optimistic Multiplicative Weights Update (OMWU) fails to converge to an equilibrium, orbiting around it.

Alternatively, perturbation of payoffs has emerged as a pivotal concept for achieving convergence (Perolat et al., 2021; Liu et al., 2023), even under noise (Abe et al., 2023). Payoff functions are

perturbed using the distance from an anchoring, or *slingshot*, strategy. However, perturbing payoffs alone fall short of reaching a Nash equilibrium of the underlying game.

This paper proposes a unified framework for perturbing approaches that ensure last-iterate convergence to the equilibrium of the underlying game with both full and noisy feedback. This framework extends Perolat et al. (2021); Abe et al. (2022) by utilizing the strongly convex divergence beyond the KL divergence, encompassing Bregman divergence, reverse KL, $\alpha$-divergence, and Rényi divergence. By using this framework, we can instantiate algorithms that rapidly converge to a near-equilibrium of the underlying game. Subsequently, in order to reach an equilibrium of the underlying game, the slingshot strategy is updated at regular intervals. Interestingly, the integration of these two components ensures last-iterate convergence at quantitatively guaranteed rates.

Our contributions are manifold. Firstly, we establish a unified framework covering a wide array of existing payoff-regularized algorithms (Tuyls et al., 2006; Perolat et al., 2021; Abe et al., 2022). Secondly, we provide convergence results to an approximate Nash equilibrium, which can be defined as a Nash equilibrium for the perturbed game with both full and noisy feedback. Thirdly, we introduce the concept of the slingshot strategy update, which allows us to demonstrate the last-iterate convergence properties. Over the course of the entire $T$ iterations, and given full feedback, our algorithm converges toward a Nash equilibrium of the underlying game at a rate of $\mathcal{O}(\ln T/\sqrt{T})$. In the case of noisy feedback, the convergence rate is $\mathcal{O}(\ln T/T^{\frac{1}{10}})$. Finally, through empirical illustrations, we show that our algorithms converge more rapidly than OMWU in a variety of games.

## 2 PRELIMINARIES

**Monotone games.** This paper focuses on a continuous game, which is denoted by $([N], (\mathcal{X}_i)_{i \in [N]}, (v_i)_{i \in [N]})$. $[N] = \{1, 2, \cdots, N\}$ represents the set of $N$ players, $\mathcal{X}_i \subseteq \mathbb{R}^{d_i}$ represents the $d_i$-dimensional compact convex strategy space for player $i \in [N]$, and we write $\mathcal{X} = \prod_{i \in [N]} \mathcal{X}_i$. Each player $i$ chooses a *strategy* $\pi_i$ from $\mathcal{X}_i$ and aims to maximize her differentiable payoff function $v_i : \mathcal{X} \to \mathbb{R}$. We write $\pi_{-i} \in \prod_{j \neq i} \mathcal{X}_i$ as the strategies of all players except player $i$, and denote the *strategy profile* by $\pi = (\pi_i)_{i \in [N]} \in \mathcal{X}$. We assume that $\mathcal{X}_i$ is an affine subset, i.e., there exists a matrix $A \in \mathbb{R}^{k_i \times d_i}$, a vector $b \in \mathbb{R}^{k_i}$ such that $A\pi_i = b$ for all $\pi_i \in \mathcal{X}_i$. This study particularly considers a *smooth monotone game*, where the gradient $(\nabla_{\pi_i} v_i)_{i \in [N]}$ of the payoff functions is monotone:

$$\forall \pi, \pi' \in \mathcal{X}, \ \sum_{i=1}^{N} \langle \nabla_{\pi_i} v_i(\pi_i, \pi_{-i}) - \nabla_{\pi_i} v_i(\pi'_i, \pi'_{-i}), \pi_i - \pi'_i \rangle \leq 0, \tag{1}$$

and $L$-Lipschitz:

$$\forall \pi, \pi' \in \mathcal{X}, \ \sum_{i=1}^{N} \|\nabla_{\pi_i} v_i(\pi_i, \pi_{-i}) - \nabla_{\pi_i} v_i(\pi'_i, \pi'_{-i})\|^2 \leq L^2 \|\pi - \pi'\|^2, \tag{2}$$

where $\|\cdot\|$ is the $\ell^2$-norm.

Monotone games include many common and well-studied classes of games, such as concave-convex games, zero-sum polymatrix games, and Cournot competition.

**Example 2.1** (Concave-Convex Games). Let us consider a max-min game $([2], (\mathcal{X}_1, \mathcal{X}_2), (v, -v))$, where $v : \mathcal{X}_1 \times \mathcal{X}_2 \to \mathbb{R}$. Player 1 aims to maximize $v$, while player 2 aims to minimize $v$. If $v$ is concave in $x_1 \in \mathcal{X}_1$ and convex in $x_2 \in \mathcal{X}_2$, the game is called a concave-convex game or minimax optimization problem, and it is easy to confirm that the game is monotone.

**Example 2.2** (Zero-Sum Polymatrix Games). In a zero-sum polymatrix game, each player's payoff function can be decomposed as $v_i(\pi) = \sum_{j \neq i} u_i(\pi_i, \pi_j)$, where $u_i : \mathcal{X}_i \times \mathcal{X}_j \to \mathbb{R}$ is represented by $u_i(\pi_i, \pi_j) = \pi_i^\top M^{(i,j)} \pi_j$ with some matrix $M^{(i,j)} \in \mathbb{R}^{d_i \times d_j}$, and satisfies $u_i(\pi_i, \pi_j) = -u_j(\pi_j, \pi_i)$. In this game, each player $i$ can be interpreted as playing a two-player zero-sum game with each other player $j \neq i$. From the linearity and zero-sum property of $u_i$, we can easily show that $\sum_{i=1}^{N} \langle \nabla_{\pi_i} v_i(\pi_i, \pi_{-i}) - \nabla_{\pi_i} v_i(\pi'_i, \pi'_{-i}), \pi_i - \pi'_i \rangle = 0$. Thus, the zero-sum polymatrix game is a special case of monotone games.

**Nash equilibrium and exploitability.** A *Nash equilibrium* (Nash, 1951) is a common solution concept of a game, which is a strategy profile where no player can improve her payoff by deviating from her specified strategy. Formally, a Nash equilibrium $\pi^* \in \mathcal{X}$ satisfies the following condition:

$$\forall i \in [N], \forall \pi_i \in \mathcal{X}_i, \ v_i(\pi_i^*, \pi_{-i}^*) \geq v_i(\pi_i, \pi_{-i}^*).$$

We denote the set of Nash equilibria by $\Pi^*$. Note that a Nash equilibrium always exists for any smooth monotone game since these games are also concave games (Debreu, 1952). Furthermore, we define $\mathrm{exploit}(\pi) := \sum_{i=1}^N \left( \max_{\tilde{\pi}_i \in \mathcal{X}_i} v_i(\tilde{\pi}_i, \pi_{-i}) - v_i(\pi) \right)$ as *exploitability* of a given strategy profile $\pi$. Exploitability is a metric of proximity to Nash equilibrium for a given strategy profile $\pi$ (Johanson et al., 2011; 2012; Lockhart et al., 2019; Abe & Kaneko, 2021). From the definition, $\mathrm{exploit}(\pi) \geq 0$ for any $\pi \in \mathcal{X}$, and the equality holds if and only if $\pi$ is a Nash equilibrium.

**Problem setting.** In this study, we consider the online learning setting where the following process is repeated for $T$ iterations: 1) At each iteration $t \geq 0$, each player $i \in [N]$ chooses her strategy $\pi_i^t \in \mathcal{X}_i$ based on the previously observed feedback; 2) Each player $i$ receives the gradient feedback $\widehat{\nabla}_{\pi_i} v_i(\pi_i^t, \pi_{-i}^t)$ as feedback. This study considers two feedback models: *full feedback* and *noisy feedback*. In the full feedback setting, each player receives the perfect gradient vector as feedback, i.e., $\widehat{\nabla}_{\pi_i} v_i(\pi_i^t, \pi_{-i}^t) = \nabla_{\pi_i} v_i(\pi_i^t, \pi_{-i}^t)$. In the noisy feedback setting, each player's feedback is given by $\widehat{\nabla}_{\pi_i} v_i(\pi_i^t, \pi_{-i}^t) = \nabla_{\pi_i} v_i(\pi_i^t, \pi_{-i}^t) + \xi_i^t$, where $\xi_i^t \in \mathbb{R}^{d_i}$ is a noise vector. Specifically, we focus on the zero-mean and bounded-variance noise vectors.

**Other notations.** We denote a $d$-dimensional probability simplex by $\Delta^d = \{p \in [0,1]^d \mid \sum_{j=1}^d p_j = 1\}$. We define $\mathrm{diam}(\mathcal{X}) := \sup_{\pi, \pi' \in \mathcal{X}} \|\pi - \pi'\|$ as the diameter of $\mathcal{X}$. For a strictly convex and differentiable function $\psi$, the associated *Bregman divergence* is defined by $D_\psi(\pi_i, \pi_i') = \psi(\pi_i) - \psi(\pi_i') - \langle \nabla \psi(\pi_i'), \pi - \pi_i' \rangle$. We denote the *Kullback-Leibler* (KL) divergence by $\mathrm{KL}(\pi_i, \pi_i') = \sum_{j=1}^{d_i} \pi_{ij} \ln \frac{\pi_{ij}}{\pi_{ij}'}$. Besides, with a slight abuse of notation, we represent the sum of Bregman divergences and the sum of KL divergences by $D_\psi(\pi, \pi') = \sum_{i=1}^N D_\psi(\pi_i, \pi_i')$, and $\mathrm{KL}(\pi, \pi') = \sum_{i=1}^N \mathrm{KL}(\pi_i, \pi_i')$, respectively.

## 3 FOLLOW THE REGULARIZED LEADER WITH SLINGSHOT PERTURBATION

In this section, we introduce Follow the Regularized Leader with Slingshot Perturbation (FTRL-SP), which is an extension of standard FTRL algorithms. Letting us define the differentiable divergence function $G(\cdot, \cdot) : \mathcal{X}_i \times \mathcal{X}_i \to [0, \infty)$ and the *slingshot* strategy $\sigma_i \in \mathcal{X}_i$, FTRL-SP perturbs each player's payoff by the divergence from the current strategy $\pi_i^t$ to the slingshot strategy $\sigma_i$, i.e., $G(\pi_i^t, \sigma_i)$. Specifically, each player's update rule of FTRL-SP is given by:

$$\pi_i^{t+1} = \arg\max_{x \in \mathcal{X}_i} \left\{ \sum_{s=0}^t \eta_s \left\langle \widehat{\nabla}_{\pi_i} v_i(\pi_i^s, \pi_{-i}^s) - \mu \nabla_{\pi_i} G(\pi_i^s, \sigma_i), x \right\rangle - \psi(x) \right\}, \quad (3)$$

where $\eta_s \in (0, \infty)$ is the learning rate at iteration $s$, $\mu \in (0, \infty)$ is the *perturbation strength*, $\psi : \mathcal{X}_i \to \mathbb{R}$ is the *regularization function*, and $\nabla_{\pi_i} G$ denotes differentiation with respect to first argument. We assume that $G(\cdot, \sigma_i)$ is strictly convex for every $\sigma_i \in \mathcal{X}_i$, and takes a minimum value of 0 at $\sigma_i$. Furthermore, we assume that $\psi$ is differentiable and $\rho$-strongly convex on $\mathcal{X}_i$ with $\rho \in (0, \infty)$, and Legendre (Rockafellar, 1997; Lattimore & Szepesvári, 2020). The pseudo-code of FTRL-SP is Algorithm 1.

The conventional FTRL updates its strategy based on the gradient feedback of the payoff function and the regularization term. The regularization term adjusts the next strategy so that it does not deviate significantly from the current strategy. The main idea of our framework is to perturb the gradient vector so that the next strategy is pulled toward a predefined strategy, which we call the *slingshot* strategy. The intuition is that perturbation with the slingshot strategy stabilizes the learning dynamics. Indeed, Mutant FTRL instantiated in Example 3.1 encompass replicator-mutator dynamics, which is guaranteed to an approximate equilibrium in two-player zero-sum games (Abe et al., 2022). FTRL-SP inherits the nice features of Mutant FTRL beyond two-player zero-sum games.

FTRL-SP can reproduce some existing learning algorithms that incorporate payoff perturbation. For example, the following learning algorithms can be viewed as instantiations of FTRL-SP.

---

**Algorithm 1** FTRL-SP with slingshot strategy update for player $i$. $T_\sigma = \infty$ corresponds to fixing the slingshot strategy.

---

**Require:** Learning rate sequence $\{\eta_t\}_{t \geq 0}$, divergence function for perturbation $G$, perturbation strength $\mu$, update interval $T_\sigma$, initial strategy $\pi_i^0$, initial slingshot strategy $\sigma_i^0$
1: $k \leftarrow 0$, $\tau \leftarrow 0$, $y_i^0 \leftarrow 0$
2: **for** $t = 0, 1, 2, \cdots$ **do**
3:     Receive the gradient feedback $\widehat{\nabla}_{\pi_i} v_i(\pi_i^t, \pi_{-i}^t)$
4:     Update the cumulative gradient vector $y_i^{t+1} \leftarrow y_i^t + \eta_t \left( \widehat{\nabla}_{\pi_i} v_i(\pi_i^t, \pi_{-i}^t) - \mu \nabla_{\pi_i} G(\pi_i^t, \sigma_i^k) \right)$
5:     Update the strategy by $\pi_i^{t+1} = \underset{x \in \mathcal{X}_i}{\arg\max} \left\{ \langle y_i^{t+1}, x \rangle - \psi(x) \right\}$
6:     $\tau \leftarrow \tau + 1$
7:     **if** $\tau = T_\sigma$ **then**
8:         $k \leftarrow k + 1$, $\tau \leftarrow 0$
9:         $\sigma_i^k \leftarrow \pi_i^t$
10:     **end if**
11: **end for**

---

**Example 3.1** (Mutant FTRL (Abe et al., 2022)). Let us define $\mathcal{X}_i = \Delta^{d_i}$, and assume that the reverse KL divergence is used as $G$, i.e., $G(\pi_i, \pi_i') = \text{KL}(\pi_i', \pi_i) = \sum_{j=1}^{d_i} \pi_{ij}' \ln \frac{\pi_{ij}'}{\pi_{ij}}$. Then, (3) can be rewriten as:

$$\pi_i^{t+1} = \underset{x \in \Delta^{d_i}}{\arg\max} \left\{ \sum_{s=0}^t \eta_s \sum_{j=1}^{d_i} x_j \left( q_{ij}^{\pi^s} + \frac{\mu}{\pi_{ij}^s} \left( \sigma_{ij} - \pi_{ij}^s \right) \right) - \psi(x) \right\},$$

where $q_{ij}^{\pi^s} = (\widehat{\nabla}_{\pi_i} v_i(\pi_i^s, \pi_{-i}^s))_j$. This is equivalent to Mutant FTRL (Abe et al., 2022).

**Example 3.2** (Reward transformed FTRL (Perolat et al., 2021)). Let us consider the continuous-time FTRL-SP dynamics with $N = 2$, $\mathcal{X}_i = \Delta^{d_i}$, and $G(\pi_i, \pi_i') = \text{KL}(\pi_i, \pi_i')$. Defining $q_{ij}^{\pi^s} = (\widehat{\nabla}_{\pi_i} v_i(\pi_i^s, \pi_{-i}^s))_j$, FTRL-SP dynamics can be described as:

$$\pi_{ij}^t = \underset{x \in \Delta^{d_i}}{\arg\max} \left\{ \int_0^t \left( \sum_{k=1}^{d_i} x_k q_{ik}^{\pi^s} - \mu \sum_{k=1}^{d_i} x_k \ln \frac{\pi_{ik}^s}{\sigma_{ik}} + \mu \sum_{k=1}^{d_{-i}} \pi_{-ik}^s \ln \frac{\pi_{-ik}^s}{\sigma_{-ik}} \right) ds - \psi(x) \right\}.$$

This algorithm is equivalent to FTRL with reward transformation (Perolat et al., 2021).

**Example 3.3** (Boltzmann Q-Learning (Tuyls et al., 2006)). In Example 3.2, we specifically assume that the regularizer is entropy and the slingshot strategy is a uniform distribution, i.e., $\psi(\pi_i) = \sum_{j=1}^{d_i} \pi_{ij} \ln \pi_{ij}$ and $\sigma_i = (1/d_i)_{j \in [d_i]}$, we have

$$\frac{d}{dt} \pi_{ij}^t = \pi_{ij}^t \left( q_{ij}^{\pi^t} - \sum_{k=1}^{d_i} \pi_{ik}^t q_{ik}^{\pi^t} \right) - \mu \pi_{ij}^t \left( \ln \pi_{ij}^t - \sum_{k=1}^{d_i} \pi_{ik}^t \ln \pi_{ik}^t \right),$$

which is equivalent to Boltzman Q-learning (Tuyls et al., 2006; Bloembergen et al., 2015).

## 4 CONVERGENCE TO AN APPROXIMATE NASH EQUILIBRIUM

Let us define $\pi^{\mu,\sigma}$ as the Nash equilibrium of the game perturbed by $\mu G(\cdot, \sigma_i)$:

$$\forall i \in [N], \ \pi_i^{\mu,\sigma} = \underset{\pi_i \in \mathcal{X}_i}{\arg\max} \left\{ v_i(\pi_i, \pi_{-i}^{\mu,\sigma}) - \mu G(\pi_i, \sigma_i) \right\}. \tag{4}$$

This section proves that the strategy $\pi^t$ updated by FTRL-SP converges to $\pi^{\mu,\sigma}$. Note that $\pi^{\mu,\sigma}$ always exists since the perturbed game is still monotone.

In this section, we fix the slingshot strategy. Specifically, we consider FTRL-SP as Algorithm 1 with the assumption that $T_\sigma = \infty$. We also assume a specific condition on the divergence function $G$:

**Assumption 4.1.** *$G(\cdot, \sigma_i)$ is $\beta$-smooth and $\gamma$-strongly convex relative to $\psi$, i.e., for any $\pi_i, \pi_i' \in \mathcal{X}_i$,*
$\gamma D_\psi(\pi_i', \pi_i) \leq G(\pi_i', \sigma_i) - G(\pi_i, \sigma_i) - \langle \nabla_{\pi_i} G(\pi_i, \sigma_i), \pi_i' - \pi_i \rangle \leq \beta D_\psi(\pi_i', \pi_i)$ *holds.*

Note that these assumptions are always satisfied with $\beta = \gamma = 1$ when $G$ is identical to $D_\psi$; thus, these are not strong assumptions.

## 4.1 FULL FEEDBACK SETTING

First, we present the convergence rate to $\pi^{\mu,\sigma}$ of FTRL-SP with *full feedback*. In this case, each player receives the perfect gradient vector $\widehat{\nabla}_{\pi_i} v_i(\pi_i^t, \pi_{-i}^t) = \nabla_{\pi_i} v_i(\pi_i^t, \pi_{-i}^t)$, at each iteration $t$. We show in Theorem 4.2 that FTRL-SP with a constant learning rate $\eta_t = \eta$ converges exponentially fast towards $\pi^{\mu,\sigma}$ for any fixed $\mu$ and $\sigma$.

**Theorem 4.2.** *Let $\pi^{\mu,\sigma} \in \mathcal{X}$ be the strategy profile that satisfies (4). Suppose that Assumption 4.1 holds with $\beta, \gamma \in (0, \infty)$. If we use the constant learning rate $\eta_t = \eta \in (0, \frac{2\mu\gamma\rho^2}{\mu^2\gamma\rho^2(\gamma+2\beta)+8L^2})$, the strategy $\pi^t$ updated by FTRL-SP satisfies that for any initial strategy profile $\pi^0 \in \mathcal{X}$ and $t \geq 1$:*

$$D_\psi(\pi^{\mu,\sigma}, \pi^t) \leq D_\psi(\pi^{\mu,\sigma}, \pi^0) \left(1 - \frac{\eta\mu\gamma}{2}\right)^t.$$

Based on this theorem, we can show that the exploitability of $\pi^t$ converges to the value of $\mathcal{O}(\mu)$.

**Theorem 4.3.** *In the same setup of Theorem 4.2, the exploitability for FTRL-SP is bounded as:*

$$\text{exploit}(\pi^t) \leq \mu \cdot \text{diam}(\mathcal{X}) \sqrt{\sum_{i=1}^N \|\nabla_{\pi_i} G(\pi_i^{\mu,\sigma}, \sigma_i)\|^2} + \mathcal{O}\left(\left(1 - \frac{\eta\mu\gamma}{2}\right)^{\frac{t}{2}}\right).$$

We note that lower $\mu$ reduces the exploitability of $\pi^{\mu,\sigma}$ (as in the first term of Theorem 4.3), whereas higher $\mu$ makes $\pi^t$ converge faster (as in the second term of Theorem 4.3). That is, $\mu$ controls a trade-off between the speed of convergence and the exploitability. Recall that $\beta = \gamma = 1$ always holds when $G$ is identical to $D_\psi$. Therefore, we can derive the following corollary:

**Corollary 4.4.** *Assume that $G$ is identical to $D_\psi$. If we set $\eta \in (0, \frac{2\mu\rho^2}{3\mu^2\rho^2+8L^2})$, the strategy $\pi^t$ updated by FTRL-SP satisfies that for $t \geq 1$:*

$$D_\psi(\pi^{\mu,\sigma}, \pi^t) \leq D_\psi(\pi^{\mu,\sigma}, \pi^0) \left(1 - \frac{\eta\mu}{2}\right)^t.$$

We provide the proofs of Theorems 4.2, 4.3, and Corollary 4.4 in Appendix B.

By applying Theorem 4.3 and Corollary 4.4, we demonstrate the convergence results for the existing learning algorithms presented in Examples 3.1-3.3.

**Example 4.5** (Mutant FTRL with log-barrier regularization in Example 3.1). Let us consider the same setup of Example 3.1. Using the log-barrier regularization $\psi(\pi_i) = -\sum_{j=1}^{d_i} \ln(\pi_{ij})$ ($\rho = 1$), we obtain the Itakura-Saito divergence as the associated Bregman divergence: $D_\psi(\pi, \pi') = \text{IS}(\pi, \pi') := \sum_{i=1}^N \sum_{j=1}^{d_i} \left(\frac{\pi_{ij}}{\pi_{ij}'} - \ln \frac{\pi_{ij}}{\pi_{ij}'} - 1\right)$. The Itakura-Saito divergence is not equivalent to $G(\pi, \pi') = \text{KL}(\pi', \pi)$. However, it is not hard to show that $G$ is $\beta$-smooth and $\gamma$-strongly convex relative to the Itakura-Saito divergence, where $\beta = 1$ and $\gamma = \min_{i\in[N], j\in[d_i]} \sigma_{ij}$. Therefore, from Theorem 4.2, the Itakura-Saito divergence from $\pi^{\mu,\sigma}$ to $\pi^t$ decreases exponentially fast: $\text{IS}(\pi^{\mu,\sigma}, \pi^t) \leq \text{IS}(\pi^{\mu,\sigma}, \pi^0) \left(1 - \frac{\eta\mu\gamma}{2}\right)^t$ for $\eta \in (0, \frac{2\mu\gamma}{\mu^2\gamma(\gamma+2)+8L^2})$.

**Example 4.6** (Reward transformed FTRL in Example 3.2). Let us consider the same setup of Example 3.2. If we use entropy regularization $\psi(\pi_i) = \sum_{j=1}^{d_i} \pi_{ij} \ln \pi_{ij}$, associated Bregman divergence is given by $D_\psi(\pi_i, \pi_i') = \text{KL}(\pi_i, \pi_i') = G(\pi_i, \pi_i)$ and $\rho = 1$. Thus, from Cororally 4.4, the KL divergence from $\pi^{\mu,\sigma}$ to $\pi^t$ decreases exponentially fast: $\text{KL}(\pi^{\mu,\sigma}, \pi^t) \leq \text{KL}(\pi^{\mu,\sigma}, \pi^0) (1 - \eta\mu/2)^t$ for $\eta \in (0, \frac{2\mu}{3\mu^2+8L^2})$.

**Example 4.7** (Boltzman Q-learning in Example 3.3). In Example 3.3, we can use a similar argument to the one in Example 4.6. From Cororally 4.4, the KL divergence from $\pi^{\mu,\sigma}$ to $\pi^t$ decreases exponentially fast: $\text{KL}(\pi^{\mu,\sigma}, \pi^t) \leq \text{KL}(\pi^{\mu,\sigma}, \pi^0) (1 - \eta\mu/2)^t$ for $\eta \in (0, \frac{2\mu}{3\mu^2+8L^2})$.

## 4.2 NOISY FEEDBACK SETTING

Next, we consider the *noisy feedback* setting, where each player $i$ receives a gradient vector with additive noise: $\nabla_{\pi_i} v_i(\pi_i^t, \pi_{-i}^t) + \xi_i^t$. Define the sigma-algebra generated by the history of the observations: $\mathcal{F}_t = \sigma\left((\widehat{\nabla}_{\pi_i} v_i(\pi_i^0, \pi_{-i}^0))_{i \in [N]}, \ldots, (\widehat{\nabla}_{\pi_i} v_i(\pi_i^{t-1}, \pi_{-i}^{t-1}))_{i \in [N]}\right), \forall t \geq 1$. We assume that $\xi_i^t \in \mathbb{R}^{d_i}$ is a zero-mean independent random vector with bounded variance:

**Assumption 4.8.** *For all $t \geq 1$ and $i \in [N]$, the noise vector $\xi_i^t$ satisfies the following properties: (a) Zero-mean: $\mathbb{E}[\xi_i^t | \mathcal{F}_t] = (0, \cdots, 0)^\top$; (b) Bounded variance: $\mathbb{E}[\|\xi_i^t\|^2 | \mathcal{F}_t] \leq C^2$.*

This is a standard assumption in learning in games with noisy feedback (Mertikopoulos & Zhou, 2019; Hsieh et al., 2019) and stochastic optimization (Nemirovski et al., 2009; Nedić & Lee, 2014). In this setting, the convergence to the stationary point $\pi^{\mu,\sigma}$ is achieved by FTRL-SP using a decreasing learning rate sequence. The convergence rate obtained by FTRL-SP is $\mathcal{O}(\ln T/T)$:

**Theorem 4.9.** *Let $\theta = \frac{\mu^2 \gamma \rho^2 (\gamma + 2\beta) + 8L^2}{2\mu\gamma\rho^2}$ and $\kappa = \frac{\mu\gamma}{2}$. Suppose that Assumptions 4.1 and 4.8 hold and the strategy $\pi^t$ is updated by FTRL-SP with the learning rate sequence of the form $\eta_t = 1/(\kappa t + 2\theta)$. Then, for all $t \geq 0$,*

$$\mathbb{E}[D_\psi(\pi^{\mu,\sigma}, \pi^{t+1})] \leq \frac{2\theta - \kappa}{\kappa t + 2\theta} D_\psi(\pi^{\mu,\sigma}, \pi^0) + \frac{NC^2}{\rho(\kappa t + 2\theta)} \left(\frac{1}{\kappa} \ln\left(\frac{\kappa}{2\theta} t + 1\right) + \frac{1}{2\theta}\right).$$

The proof is given in Appendix C and is based on the standard argument of stochastic optimization, e.g., Nedić & Lee (2014). However, the proof is made possible by taking into account the monotonicity of the game and the relative (strong and smooth) convexity of the divergence function.

## 4.3 EXTENSION TO MIRROR DESCENT

In previous sections, we introduced and analyzed FTRL-SP, which extends the standard FTRL approach. Similarly, it is possible to extend the mirror descent approach as well. In this section, we present Mirror Descent with Slingshot Perturbation (MD-SP), which incorporates the perturbation term $\mu G(\cdot, \sigma_i)$ into the conventional mirror descent algorithm:

$$\pi_i^{t+1} = \arg\max_{x \in \mathcal{X}_i} \left\{\eta_t \left\langle \widehat{\nabla}_{\pi_i} v_i(\pi_i^t, \pi_{-i}^t) - \mu\nabla_{\pi_i} G(\pi_i^t, \sigma_i), x\right\rangle - D_\psi(x, \pi_i^t)\right\}. \tag{5}$$

MD-SP achieves the same convergence rates as FTRL-SP in full and noisy feedback settings.

**Theorem 4.10.** *Let $\pi^{\mu,\sigma} \in \mathcal{X}$ be the strategy profile that satisfies (4). Suppose that Assumption 4.1 holds with $\beta, \gamma \in (0, \infty)$. If we use the constant learning rate $\eta_t = \eta \in (0, \frac{2\mu\gamma\rho^2}{\mu^2 \gamma \rho^2 (\gamma + 2\beta) + 8L^2})$, the strategy $\pi^t$ updated by MD-SP satisfies that for any initial strategy profile $\pi^0 \in \mathcal{X}$ and $t \geq 1$:*

$$D_\psi(\pi^{\mu,\sigma}, \pi^t) \leq D_\psi(\pi^{\mu,\sigma}, \pi^0) \left(1 - \frac{\eta\mu\gamma}{2}\right)^t.$$

**Theorem 4.11.** *Let $\theta = \frac{\mu^2 \gamma \rho^2 (\gamma + 2\beta) + 8L^2}{2\mu\gamma\rho^2}$ and $\kappa = \frac{\mu\gamma}{2}$. Suppose that Assumptions 4.1 and 4.8 hold and the strategy $\pi^t$ is updated by MD-SP with the learning rate sequence of the form $\eta_t = 1/(\kappa t + 2\theta)$. Then, for all $t \geq 0$,*

$$\mathbb{E}[D_\psi(\pi^{\mu,\sigma}, \pi^{t+1})] \leq \frac{2\theta - \kappa}{\kappa t + 2\theta} D_\psi(\pi^{\mu,\sigma}, \pi^0) + \frac{NC^2}{\rho(\kappa t + 2\theta)} \left(\frac{1}{\kappa} \ln\left(\frac{\kappa}{2\theta} t + 1\right) + \frac{1}{2\theta}\right).$$

The proofs of these theorems can be found in Appendix D. Note that MD-SP does not rely on the assumption that the regularization function $\psi$ is Legendre nor that $\mathcal{X}_i$ is an affine subset. As a result, MD-SP is applicable to a broader class of games compared to FTRL-SP.

## 5 SLINGSHOT STRATEGY UPDATE FOR LAST-ITERATE CONVERGENCE

In the preceding section, we demonstrated the convergence results for our framework with a fixed slingshot strategy profile $\sigma$. Perturbation of the payoff functions enables $\pi^t$ to converge quickly to

$\pi^{\mu,\sigma}$ (as in Theorem 4.2), whereas it also moves $\pi^{\mu,\sigma}$ away from the Nash equilibrium of the original game (as in Theorem 4.3). On the other hand, when the slingshot strategy profile $\sigma$ is close to an equilibrium $\pi^* \in \Pi^*$, the solution of (4), i.e., $\pi^{\mu,\sigma}$, is close to $\pi^*$. Therefore, we iteratively update $\sigma$ to a better strategy so that it converges to an equilibrium. To this end, we overwrite $\sigma$ with the current strategy profile $\pi^t$ every $T_\sigma$ iterations. The pseudo-code of FTRL-SP with this adaptation method corresponds to Algorithm 1 with finite $T_\sigma$.

**Last-Iterate Convergence Rates to Nash equilibria.** We denote $\sigma^k$ as the slingshot strategy after $k$ updates and $K$ as the total number of the slingshot strategy updates over the entire $T(= T_\sigma K)$ iterations. We demonstrate the last-iterate convergence rates for $\sigma^K$. The complete proofs are presented in Appendix E. According to Theorems 4.2 and 4.9, we can observe that $D_\psi(\pi^{\mu,\sigma^k}, \sigma^{k+1}) = D_\psi(\pi^{\mu,\sigma^k}, \sigma^k)\exp(-\mathcal{O}(T_\sigma))$ in the full feedback setting, and $\mathbb{E}[D_\psi(\pi^{\mu,\sigma^k}, \sigma^{k+1})] = \mathcal{O}(\ln T_\sigma/T_\sigma)$ in the noisy feedback setting for $k \geq 0$, respectively. Here, we assume that $G$ is the squared $\ell^2$-distance, meaning that $G(\pi_i, \pi_i') = \frac{1}{2}\|\pi_i - \pi_i\|^2$. We adjust $T_\sigma = \Omega(\ln K)$ and $T_\sigma = \Omega(K^4)$ in the full/noisy feedback setting, respectively. From the first-order optimality condition for $\pi^{\mu,\sigma^k}$, we can show that the distance between $\sigma^k$ and a Nash equilibrium $\pi^*$ does not increase too much, i.e., $\|\sigma^{k+1} - \pi^*\|^2 - \|\sigma^k - \pi^*\|^2 \leq -\|\pi^{\mu,\sigma^k} - \sigma^k\|^2/2 + \mathcal{O}(1/K^2)$. By telescoping this inequality, we get $\sum_{k=0}^{K-1} \|\pi^{\mu,\sigma^k} - \sigma^k\|^2 \leq \mathcal{O}(\|\sigma^0 - \pi^*\|^2 + 1/K^2)$. Again using the first-order optimality condition for $\pi^{\mu,\sigma^k}$ and $\pi^{\mu,\sigma^{k-1}}$, we can derive that $\|\pi^{\mu,\sigma^k} - \sigma^k\|^2 \leq \|\sigma^k - \sigma^{k-1}\|^2 + \mathcal{O}(1/K^2)$. Finally, we observe that $\|\pi^{\mu,\sigma^{K-1}} - \sigma^{K-1}\|^2 \leq \mathcal{O}(1/\sqrt{K})$, leading to the last-iterate convergence rates of $\mathcal{O}(1/\sqrt{K})$ and $\mathcal{O}(\ln K/\sqrt{K})$ in the full/noisy feedback setting, respectively:

**Theorem 5.1.** *Assume that* $\|\pi^{\mu,\sigma^k} - \sigma^{k+1}\| \leq \|\pi^{\mu,\sigma^k} - \sigma^k\| \left(\frac{1}{c}\right)^{T_\sigma}$ *for some* $c > 1$, *and* $\sqrt{\sum_{i=1}^N \|\nabla_{\pi_i} v_i(\pi)\|^2} \leq \zeta$ *for any* $\pi \in \mathcal{X}$. *Then, if we set* $G$ *as squared* $\ell^2$-*distance* $G(\pi_i, \pi_i') = \frac{1}{2}\|\pi_i - \pi_i'\|^2$ *and* $T_\sigma \geq \max(\frac{3}{\ln c}\ln K + \frac{\ln 64}{\ln c}, 1)$, *we have for any Nash equilibrium* $\pi^* \in \Pi^*$:

$$\mathrm{exploit}(\sigma^K) \leq \frac{2\sqrt{2}\left((\mu+L)\cdot\mathrm{diam}(\mathcal{X}) + \zeta\right)}{\sqrt{K}}\sqrt{\|\pi^* - \sigma^0\|\left(8\|\pi^* - \sigma^0\| + \frac{\zeta}{\mu}\right)}.$$

**Theorem 5.2.** *Assume that* $\mathbb{E}[\|\pi^{\mu,\sigma^k} - \sigma^{k+1}\|^2 \mid \sigma^k] \leq c^2\frac{\ln T_\sigma}{T_\sigma}$ *for any* $k \geq 0$ *for some* $c > 0$, *and* $\sqrt{\sum_{i=1}^N \|\nabla_{\pi_i} v_i(\pi)\|^2} \leq \zeta$ *for any* $\pi \in \mathcal{X}$. *Then, if we set* $G$ *as squared* $\ell^2$-*distance* $G(\pi_i, \pi_i') = \frac{1}{2}\|\pi_i - \pi_i'\|^2$ *and* $T_\sigma \geq \max(K^4, 3)$, *we have for any Nash equilibrium* $\pi^* \in \Pi^*$:

$$\mathbb{E}[\mathrm{exploit}(\sigma^K)] \leq \frac{2\left(\mathrm{diam}(\mathcal{X})\cdot\left(cL + \mu\sqrt{c}\sqrt{2\mathrm{diam}(\mathcal{X}) + \frac{\zeta}{\mu} + c}\right) + c\zeta\right)\ln K + \mu\cdot\mathrm{diam}(\mathcal{X})^2}{\sqrt{K}}.$$

We emphasize that we have obtained the overall last-iterative convergence rates of our algorithm for the entire $T$ iterations in both full and noisy feedback settings. Specifically, from Theorem 5.1, we can derive the rate of $\mathrm{exploit}(\pi^T) \leq \mathcal{O}(\ln T/\sqrt{T})$ by setting $T_\sigma = \Theta(\ln T)$ in the full feedback setting. Similarly, from Theorem 5.2, the rate of $\mathbb{E}[\mathrm{exploit}(\pi^T)] \leq \mathcal{O}(\ln T/T^{\frac{1}{10}})$ in the noisy feedback setting can be achieved when $T_\sigma$ is set to $T_\sigma = \Theta(T^{4/5})$.

We also provide the convergence results for our algorithm where $G$ is one of the following divergence functions in Appendix G: 1) Bregman divergence; 2) $\alpha$-divergence; 2) Rényi-divergence; 3) reverse KL divergence. We note that these results cover the algorithms in Example 3.1, 3.2, and 3.3.

Our algorithm is related to Clairvoyant Mirror Descent (Farina et al., 2022; Piliouras et al., 2022; Cevher et al., 2023), which have been considered only in the full feedback setting. In contrast to this algorithm, we achieve last-iterate convergence with rates in both full and noisy feedback settings by utilizing the strong convexity of $G$.

## 6 EXPERIMENTS

This section empirically compares the representative instance of FTRL, namely Multiplicative Weight Update (MWU) and its Optimistic version (OMWU), with our framework. Specifically,

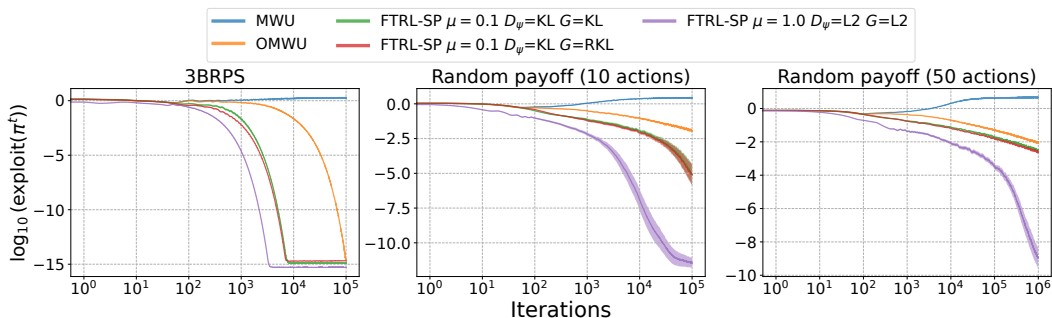

Figure 1: Exploitability of $\pi^t$ for FTRL-SP, MWU, and OMWU with full feedback. The shaded area represents the standard errors. Note that the KL divergence, reverse KL divergence, and squared $\ell^2$-distance are abbreviated to KL, RKL, and L2, respectively.

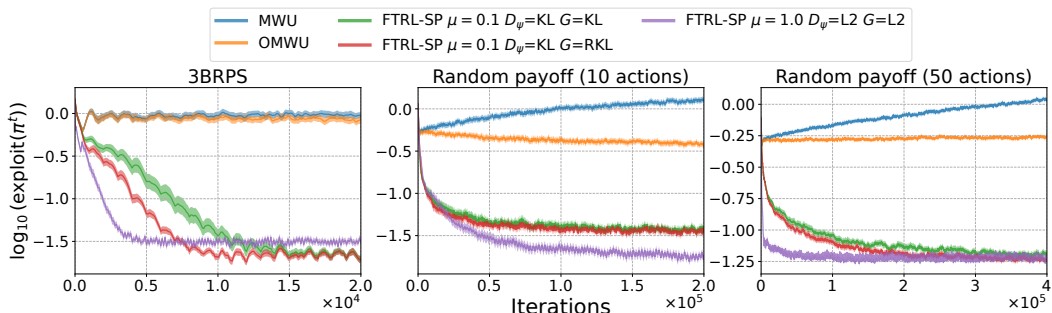

Figure 2: Exploitability of $\pi^t$ for FTRL-SP, MWU, and OMWU with noisy feedback.

we consider the following three instances of FTRL-SP: (i) the divergence function $G$ is the reverse KL divergence, and the Bregman divergence $D_\psi$ is the KL divergence, which matches Mutant FTRL in Example 3.1. (ii) both $G$ and $D_\psi$ are the KL divergence, which is also an instance of Reward transformed FTRL in Example 3.2. Note that if the slingshot strategy is fixed to a uniformly random strategy, this algorithm corresponds to Boltzmann Q-Learning in Example 3.3; (iii) both are the squared $\ell^2$-distance.

We focus on two zero-sum polymatrix games: Three-Player Biased Rock-Paper-Scissors (3BRPS) and three-player random payoff games with 10 and 50 actions. For the 3BRPS game, each player participates in two instances of the game in Table 2 in Appendix H simultaneously with two other players. For the random payoff games, each player $i$ participates in two instances of the game with two other players $j$ simultaneously. The payoff matrix for each instance is denoted as $M^{(i,j)}$. Each entry of $M^{(i,j)}$ is drawn independently from a uniform distribution on the interval $[-1, 1]$.

Figures 1 and 2 illustrate the logarithmic exploitability averaged over 100 instances with different random seeds. We assume that the initial slingshot strategy $\pi^0$ is chosen uniformly at random in the interior of the strategy space $\mathcal{X} = \prod_{i=1}^3 \Delta^{d_i}$ in each instance for 3BRPS, while $\pi^0$ is chosen as $(1/d_i)_{j \in [d_i]}$ for $i \in [3]$ in every instances for the random payoff games.

First, Figure 1 depicts the case of full feedback. Unless otherwise specified, we use a constant learning rate $\eta = 0.1$ and a perturbation strength $\mu = 0.1$ for FTRL-SP. Further details and additional experiments can be found in Appendix H. Figure 1 shows that FTRL-SP outperforms MWU and OMWU in all three games. Notably, FTRL-SP exhibits the fastest convergence in terms of exploitability when using the squared $\ell^2$-distance as both $G$ and $D_\psi$. Next, Figure 2 depicts the case of noisy feedback. We assume that the noise vector $\xi_i^t$ is generated from the multivariate Gaussian distribution $\mathcal{N}(0, 0.1^2 \mathbf{I})$ in an i.i.d. manner. To account for the noise, we use a lower learning rate $\eta = 0.01$ than the full feedback case. In OMWU, we use the noisy gradient vector $\widehat{\nabla}_{\pi_i} v_i(\pi_i^{t-1}, \pi_{-i}^{t-1})$ at the previous step $t - 1$ as the prediction vector for the current iteration $t$. We

observe the same trends as with full feedback. While MWU and OMWU exhibit worse performance, FTRL-SP maintains its fast convergence, as predicted by the theoretical analysis.

# 7 RELATED LITERATURE

Recent progress in achieving no-regret learning with full feedback has been driven by optimistic learning (Rakhlin & Sridharan, 2013a;b). Optimistic versions of well-known algorithms like Follow the Regularized Leader (Shalev-Shwartz & Singer, 2006) and Mirror Descent (Zhou et al., 2017; Hsieh et al., 2021) have been proposed to admit last-iterate convergence in a wide range of game settings. These optimistic algorithms have been successfully applied to various classes of games, including bilinear games (Daskalakis et al., 2018; Daskalakis & Panageas, 2019; Liang & Stokes, 2019; de Montbrun & Renault, 2022), cocoercive games (Lin et al., 2020), and saddle point problems (Daskalakis & Panageas, 2018; Mertikopoulos et al., 2019; Golowich et al., 2020b; Wei et al., 2021; Lei et al., 2021; Yoon & Ryu, 2021; Lee & Kim, 2021; Cevher et al., 2023). Recent advancements have provided solutions to monotone games and have established convergence rates (Golowich et al., 2020a; Cai et al., 2022a;b; Gorbunov et al., 2022; Cai & Zheng, 2023).

The exploration of literature with noisy feedback poses significant challenges, in contrast to full feedback. In situations where feedback is imprecise or limited, algorithms must estimate action values at each iteration. There has been significant progress in achieving last-iterate convergence in specific classes of games when noisy feedback is present. This progress is particularly noticeable in potential games (Cohen et al., 2017), strongly monotone games (Giannou et al., 2021b;a), and two-player zero-sum games (Abe et al., 2023). Previous results with noisy feedback have often relied on strict (or strong) monotonicity (Bravo et al., 2018; Kannan & Shanbhag, 2019; Hsieh et al., 2019; Anagnostides & Panageas, 2022) and strict variational stability (Mertikopoulos et al., 2019; Azizian et al., 2021; Mertikopoulos & Zhou, 2019; Mertikopoulos et al., 2022). Even if these restrictions are not imposed, convergence is demonstrated in an asymptotic sense and the rate is not quantified (Koshal et al., 2010; 2013; Yousefian et al., 2017; Hsieh et al., 2022; Abe et al., 2023). As a result, prohibitively long rounds may be required to achieve this convergence.

From an algorithmic perspective, our work is closely related to payoff-regularized learning, where the payoff or utility functions of each player are perturbed or regularized through entropy functions or $\ell^2$-distance functions (Cen et al., 2021; 2023; Cai et al., 2023; Pattathil et al., 2023). Previous studies have successfully achieved convergence to stationary points or approximate equilibria characterized by the payoff regularizing parameter. For instance, Sokota et al. (2023) showed that their mirror descent-based algorithm converges to a quantal response equilibrium (McKelvey & Palfrey, 1995; 1998), which can be viewed as an approximate equilibrium. Similar results have been obtained with the Boltzmann Q-learning dynamics (Tuyls et al., 2006) in continuous-time settings (Hussain et al., 2023). Our framework provides a comprehensive understanding of the relationship between convergence points of dynamics and approximate equilibria. We can argue that our framework unifies the existing payoff-regularized learning algorithms, each of which perturbs payoffs by different functions, e.g., strongly convex functions (Facchinei & Pang, 2003; Liu et al., 2023; Bernasconi et al., 2022; Sokota et al., 2023) or divergence functions (Perolat et al., 2021; Abe et al., 2022; 2023). Although Sokota et al. (2023) employs a similar perturbation to our slingshot strategy, which they call *magnetic* strategy, they only provide convergence toward an approximate equilibrium. On the contrary, we manage to derive convergence toward a Nash equilibrium of the underlying game by incorporating iterative updates of the slingshot strategy. This non-trivial convergence with full and noisy feedback arises from the interplay between our simple, but distinct payoff perturbation technique and the iterative updates.

# 8 CONCLUSION

We developed a slingshot framework to compute equilibria in monotone games, where the slingshot perturbation enables convergence even when the noise is present. This research could lead to several intriguing future studies, such as finding the best perturbation strength for the optimal convergence rate and achieving convergence with more limited feedback, for example, using bandit feedback (Bravo et al., 2018; Tatarenko & Kamgarpour, 2019; Drusvyatskiy et al., 2022).

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

# A  NOTATIONS

In this section, we summarize the notations we use in Table 1.

Table 1: Notations

| Symbol | Description |
|---|---|
| $N$ | Number of players |
| $\mathcal{X}_i$ | Strategy space for player $i$ |
| $\mathcal{X}$ | Joint strategy space: $\mathcal{X} = \prod_{i=1}^{N} \mathcal{X}_i$ |
| $v_i$ | Payoff function for player $i$ |
| $\pi_i$ | Strategy for player $i$ |
| $\pi$ | Strategy profile: $\pi = (\pi_i)_{i \in [N]}$ |
| $\xi_i^t$ | Noise vector for player $i$ at iteration $t$ |
| $\pi^*$ | Nash equilibrium |
| $\Pi^*$ | Set of Nash equilibria |
| $\text{exploit}(\pi)$ | Exploitability of $\pi$: $\text{exploit}(\pi) = \sum_{i=1}^{N} \left( \max_{\tilde{\pi}_i \in \mathcal{X}_i} v_i(\tilde{\pi}_i, \pi_{-i}) - v_i(\pi) \right)$ |
| $\Delta^d$ | $d$-dimensional probability simplex: $\Delta^d = \{ p \in [0,1]^d \mid \sum_{j=1}^{d} p_j = 1 \}$ |
| $\text{diam}(\mathcal{X})$ | Diameter of $\mathcal{X}$: $\text{diam}(\mathcal{X}) = \sup_{\pi, \pi' \in \mathcal{X}} \|\pi - \pi'\|$ |
| $\text{KL}(\cdot, \cdot)$ | Kullback-Leibler divergence |
| $D_\psi(\cdot, \cdot)$ | Bregman divergence associated with $\psi$ |
| $\nabla_{\pi_i} v_i$ | Gradient of $v_i$ with respect to $\pi_i$ |
| $\eta_t$ | Learning rate at iteration $t$ |
| $\mu$ | Perturbation strength |
| $\sigma$ | Slingshot strategy profile |
| $G(\cdot, \cdot)$ | Divergence function for payoff perturbation |
| $\nabla_{\pi_i} G$ | Gradient of $G$ with respect to first argument |
| $T_\sigma$ | Update interval for the slingshot strategy |
| $K$ | Total number of the slingshot strategy updates |
| $\pi^{\mu, \sigma}$ | Stationary point satisfies (4) for given $\mu$ and $\sigma$ |
| $\pi^t$ | Strategy profile at iteration $t$ |
| $\sigma^k$ | Slingshot strategy after $k$ updates |
| $L$ | Smoothness parameter of $(v_i)_{i \in [N]}$ |
| $\rho$ | Strongly convex parameter of $\psi$ |
| $\beta$ | Smoothness parameter of $G(\cdot, \sigma_i)$ relative to $\psi$ |
| $\gamma$ | Strongly convex parameter of $G(\cdot, \sigma_i)$ relative to $\psi$ |

# B   PROOFS FOR SECTION 4.1

## B.1   PROOF OF THEOREM 4.2

*Proof of Theorem 4.2.*   First, we introduce the following lemma:

**Lemma B.1.** *Let us define* $T(y_i) = \arg\max_{x \in \mathcal{X}_i}\{\langle y_i, x \rangle - \psi(x)\}$. *Assuming* $\psi : \mathcal{X}_i \to \mathbb{R}$ *be a convex function of the Legendre type, we have for any* $\pi_i \in \mathcal{X}_i$:

$$D_\psi(\pi_i, T(y_i)) = \psi(\pi_i) - \psi(T(y_i)) - \langle y_i, \pi_i - T(y_i) \rangle.$$

Defining $y_i^t = \eta \sum_{s=0}^t \left( \nabla_{\pi_i} v_i(\pi_i^s, \pi_{-i}^s) - \mu \nabla_{\pi_i} G(\pi_i^s, \sigma_i) \right)$ and letting $\pi_i = \pi_i^{\mu,\sigma}$, $y_i = y_i^t$ in Lemma B.1, we have:

$$D_\psi(\pi_i^{\mu,\sigma}, \pi_i^{t+1}) = \psi(\pi_i^{\mu,\sigma}) - \psi(\pi_i^{t+1}) - \langle y_i^t, \pi_i^{\mu,\sigma} - \pi_i^{t+1} \rangle.$$

Using this equation, we get for any $t \geq 0$:

$$
\begin{aligned}
&D_\psi(\pi_i^{\mu,\sigma}, \pi_i^{t+1}) - D_\psi(\pi_i^{\mu,\sigma}, \pi_i^t) + D_\psi(\pi_i^{t+1}, \pi_i^t) \\
&= \psi(\pi_i^{\mu,\sigma}) - \psi(\pi_i^{t+1}) - \langle y_i^t, \pi_i^{\mu,\sigma} - \pi_i^{t+1} \rangle - \psi(\pi_i^{\mu,\sigma}) + \psi(\pi_i^t) + \langle y_i^{t-1}, \pi_i^{\mu,\sigma} - \pi_i^t \rangle \\
&\quad + \psi(\pi_i^{t+1}) - \psi(\pi_i^t) - \langle y_i^{t-1}, \pi_i^{t+1} - \pi_i^t \rangle \\
&= -\langle y_i^t, \pi_i^{\mu,\sigma} - \pi_i^{t+1} \rangle + \langle y_i^{t-1}, \pi_i^{\mu,\sigma} - \pi_i^{t+1} \rangle \\
&= \langle y_i^t - y_i^{t-1}, \pi_i^{t+1} - \pi_i^{\mu,\sigma} \rangle \\
&= \eta \langle \nabla_{\pi_i} v_i(\pi_i^t, \pi_{-i}^t) - \mu \nabla_{\pi_i} G(\pi_i^t, \sigma_i), \pi_i^{t+1} - \pi_i^{\mu,\sigma} \rangle. \quad (6)
\end{aligned}
$$

Next, we derive the following convergence result for $\pi^t$:

**Lemma B.2.** *Suppose that Assumption 4.1 holds with* $\beta, \gamma \in (0, \infty)$, *and the updated strategy profile* $\pi^t$ *satisfies the following condition: for any* $t \geq 0$,

$$
\begin{aligned}
&D_\psi(\pi^{\mu,\sigma}, \pi^{t+1}) - D_\psi(\pi^{\mu,\sigma}, \pi^t) + D_\psi(\pi^{t+1}, \pi^t) \\
&\leq \eta \sum_{i=1}^N \langle \nabla_{\pi_i} v_i(\pi_i^t, \pi_{-i}^t) - \mu \nabla_{\pi_i} G(\pi_i^t, \sigma_i), \pi_i^{t+1} - \pi_i^{\mu,\sigma} \rangle.
\end{aligned}
$$

*Then, for any* $t \geq 0$:

$$D_\psi(\pi^{\mu,\sigma}, \pi^t) \leq D_\psi(\pi^{\mu,\sigma}, \pi^0) \left( 1 - \frac{\eta\mu\gamma}{2} \right)^t.$$

It is easy to confirm that (6) satisfies the assumption in Lemma B.2. Thus, we conclude the statement. □

## B.2   PROOF OF THEOREM 4.3

*Proof of Theorem 4.3.*   Since $v_i(\cdot, \pi_{-i}^t)$ is concave, we can upper bound the exploitability as:

$$
\begin{aligned}
\mathrm{exploit}(\pi^t) &= \sum_{i=1}^N \left( \max_{\tilde{\pi}_i \in \mathcal{X}_i} v_i(\tilde{\pi}_i, \pi_{-i}^t) - v_i(\pi^t) \right) \\
&\leq \sum_{i=1}^N \max_{\tilde{\pi}_i \in \mathcal{X}_i} \langle \nabla_{\pi_i} v_i(\pi^t), \tilde{\pi}_i - \pi_i^t \rangle \\
&= \max_{\tilde{\pi} \in \mathcal{X}} \sum_{i=1}^N \left( \langle \nabla_{\pi_i} v_i(\pi^{\mu,\sigma}), \tilde{\pi}_i - \pi_i^{\mu,\sigma} \rangle - \langle \nabla_{\pi_i} v_i(\pi^{\mu,\sigma}), \pi_i^t - \pi_i^{\mu,\sigma} \rangle \right. \\
&\qquad\qquad \left. + \langle \nabla_{\pi_i} v_i(\pi^t) - \nabla_{\pi_i} v_i(\pi^{\mu,\sigma}), \tilde{\pi}_i - \pi_i^t \rangle \right). \quad (7)
\end{aligned}
$$

Here, we introduce the following lemma from Cai et al. (2022a):

**Lemma B.3** (Lemma 2 of Cai et al. (2022a)). *For any $\pi \in \mathcal{X}$, we have:*

$$\max_{\tilde{\pi} \in \mathcal{X}} \sum_{i=1}^{N} \langle \nabla_{\pi_i} v_i(\pi), \tilde{\pi}_i - \pi_i \rangle \leq \operatorname{diam}(\mathcal{X}) \cdot \min_{(a_i) \in N_{\mathcal{X}}(\pi)} \sqrt{\sum_{i=1}^{N} \| -\nabla_{\pi_i} v_i(\pi) + a_i \|^2},$$

*where $N_{\mathcal{X}}(\pi) = \{(a_i)_{i \in [N]} \in \prod_{i=1}^{N} \mathbb{R}^{d_i} \mid \sum_{i=1}^{N} \langle a_i, \pi_i' - \pi_i \rangle \leq 0, \ \forall \pi' \in \mathcal{X}\}$.*

From Lemma B.3, the first term of (7) can be upper bounded as:

$$\max_{\tilde{\pi} \in \mathcal{X}} \sum_{i=1}^{N} \langle \nabla_{\pi_i} v_i(\pi^{\mu,\sigma}), \tilde{\pi}_i - \pi_i^{\mu,\sigma} \rangle \leq \operatorname{diam}(\mathcal{X}) \cdot \min_{(a_i) \in N_{\mathcal{X}}(\pi^{\mu,\sigma})} \sqrt{\sum_{i=1}^{N} \| -\nabla_{\pi_i} v_i(\pi^{\mu,\sigma}) + a_i \|^2}.$$

From the first-order optimality condition for $\pi^{\mu,\sigma}$, we have for any $\pi \in \mathcal{X}$:

$$\sum_{i=1}^{N} \langle \nabla_{\pi_i} v_i(\pi^{\mu,\sigma}) - \mu \nabla_{\pi_i} G(\pi_i^{\mu,\sigma}, \sigma_i), \pi_i - \pi_i^{\mu,\sigma} \rangle \leq 0,$$

and then $(\nabla_{\pi_i} v_i(\pi^{\mu,\sigma}) - \mu \nabla_{\pi_i} G(\pi_i^{\mu,\sigma}, \sigma_i))_{i \in [N]} \in N_{\mathcal{X}}(\pi^{\mu,\sigma})$. Thus,

$$\max_{\tilde{\pi} \in \mathcal{X}} \sum_{i=1}^{N} \langle \nabla_{\pi_i} v_i(\pi^{\mu,\sigma}), \tilde{\pi}_i - \pi_i^{\mu,\sigma} \rangle$$

$$\leq \operatorname{diam}(\mathcal{X}) \sqrt{\sum_{i=1}^{N} \| -\nabla_{\pi_i} v_i(\pi^{\mu,\sigma}) + \nabla_{\pi_i} v_i(\pi^{\mu,\sigma}) - \mu \nabla_{\pi_i} G(\pi_i^{\mu,\sigma}, \sigma_i) \|^2}$$

$$= \mu \cdot \operatorname{diam}(\mathcal{X}) \sqrt{\sum_{i=1}^{N} \| \nabla_{\pi_i} G(\pi_i^{\mu,\sigma}, \sigma_i) \|^2}. \tag{8}$$

Next, from Cauchy–Schwarz inequality, the second term of (7) can be bounded as:

$$-\sum_{i=1}^{N} \langle \nabla_{\pi_i} v_i(\pi^{\mu,\sigma}), \pi_i^t - \pi_i^{\mu,\sigma} \rangle \leq \| \pi^t - \pi^{\mu,\sigma} \| \sqrt{\sum_{i=1}^{N} \| \nabla_{\pi_i} v_i(\pi^{\mu,\sigma}) \|^2}. \tag{9}$$

Again from Cauchy–Schwarz inequality, the third term of (7) is bounded by:

$$\sum_{i=1}^{N} \langle \nabla_{\pi_i} v_i(\pi^t) - \nabla_{\pi_i} v_i(\pi^{\mu,\sigma}), \tilde{\pi}_i - \pi_i^t \rangle \leq \| \tilde{\pi} - \pi^t \| \sqrt{\sum_{i=1}^{N} \| \nabla_{\pi_i} v_i(\pi^t) - \nabla_{\pi_i} v_i(\pi^{\mu,\sigma}) \|^2}$$

$$\leq \operatorname{diam}(\mathcal{X}) \sqrt{\sum_{i=1}^{N} \| \nabla_{\pi_i} v_i(\pi^t) - \nabla_{\pi_i} v_i(\pi^{\mu,\sigma}) \|^2}$$

$$\leq L \cdot \operatorname{diam}(\mathcal{X}) \| \pi^t - \pi^{\mu,\sigma} \|, \tag{10}$$

where the third inequality follows from (2).

By combining (7), (8), (9), and (10), we get:

$$\operatorname{exploit}(\pi^t)$$

$$\leq \mu \cdot \operatorname{diam}(\mathcal{X}) \sqrt{\sum_{i=1}^{N} \| \nabla_{\pi_i} G(\pi_i^{\mu,\sigma}, \sigma_i) \|^2} + \left( L \cdot \operatorname{diam}(\mathcal{X}) + \sqrt{\sum_{i=1}^{N} \| \nabla_{\pi_i} v_i(\pi^{\mu,\sigma}) \|^2} \right) \| \pi^t - \pi^{\mu,\sigma} \|.$$

Thus, from Theorem 4.2 and the strong convexity of $\psi$, we have:

$$\text{exploit}(\pi^t) \leq \mu \cdot \text{diam}(\mathcal{X}) \sqrt{\sum_{i=1}^{N} \|\nabla_{\pi_i} G(\pi_i^{\mu,\sigma}, \sigma_i)\|^2}$$

$$+ \left( L \cdot \text{diam}(\mathcal{X}) + \sqrt{\sum_{i=1}^{N} \|\nabla_{\pi_i} v_i(\pi^{\mu,\sigma})\|^2} \right) \sqrt{\frac{2 D_\psi(\pi^{\mu,\sigma}, \pi^0)}{\rho} \left(1 - \frac{\eta \mu \gamma}{2}\right)^t}.$$

$\square$

### B.3 PROOF OF COROLLARY 4.4

*Proof of Corollary 4.4.* From the definition of the Bregman divergence, we have for all $\pi_i, \pi_i' \in \mathcal{X}_i$:

$$D_\psi(\pi_i', \sigma_i) - D_\psi(\pi_i, \sigma_i) - \langle \nabla_{\pi_i} D_\psi(\pi_i, \sigma_i), \pi_i' - \pi_i \rangle$$
$$= \psi(\pi_i') - \psi(\sigma_i) - \langle \nabla \psi(\sigma_i), \pi_i' - \sigma_i \rangle - \psi(\pi_i) + \psi(\sigma_i) + \langle \nabla \psi(\sigma_i), \pi_i - \sigma_i \rangle$$
$$\quad - \langle \nabla \psi(\pi_i) - \nabla \psi(\sigma_i), \pi_i' - \pi_i \rangle$$
$$= \psi(\pi_i') - \psi(\pi_i) - \langle \nabla \psi(\pi_i), \pi_i' - \pi_i \rangle$$
$$= D_\psi(\pi_i', \pi_i).$$

Therefore, Assumption 4.1 holds with $\beta = \gamma = 1$, and then we conclude the statement. $\square$

### B.4 DETAILS OF EXAMPLE 4.5

*Details of Example 4.5.* Defining $\psi(\pi_i) = -\sum_{j=1}^{d_i} \ln(\pi_{ij})$, we have $D_\psi(\pi_i, \pi_i') = \text{IS}(\pi_i, \pi_i') :=$ $\sum_{j=1}^{d_i} \left( \frac{\pi_{ij}}{\pi_{ij}'} - \ln \frac{\pi_{ij}}{\pi_{ij}'} - 1 \right)$ for all $\pi_i, \pi_i' \in \Delta^{d_i}$. Furthermore, $\psi$ is twice differentiable and has a diagonal Hessian:

$$\nabla^2 \psi(\pi_i) = \begin{bmatrix} \frac{1}{\pi_{i1}^2} & & \\ & \ddots & \\ & & \frac{1}{\pi_{id_i}^2}, \end{bmatrix}$$

and thus, its smallest eigenvalue is lower bounded by 1. That is, $\rho = 1$.

Next, we have:

$$G(\pi_i', \sigma_i) - G(\pi_i, \sigma_i) - \langle \nabla_{\pi_i} G(\pi_i, \sigma_i), \pi_i' - \pi_i \rangle$$
$$= \sum_{j=1}^{d_i} \sigma_{ij} \ln \left( \frac{\sigma_{ij}}{\pi_{ij}'} \right) - \sum_{j=1}^{d_i} \sigma_{ij} \ln \left( \frac{\sigma_{ij}}{\pi_{ij}} \right) + \sum_{j=1}^{d_i} (\pi_{ij}' - \pi_{ij}) \frac{\sigma_{ij}}{\pi_{ij}}$$
$$= \sum_{j=1}^{d_i} \sigma_{ij} \ln \left( \frac{\pi_{ij}}{\pi_{ij}'} \right) + \sum_{j=1}^{d_i} \sigma_{ij} \frac{\pi_{ij}' - \pi_{ij}}{\pi_{ij}}$$
$$= \sum_{j=1}^{d_i} \sigma_{ij} \left( \frac{\pi_{ij}'}{\pi_{ij}} - \ln \left( \frac{\pi_{ij}'}{\pi_{ij}} \right) - 1 \right).$$

Thus, $G$ satisfies that:

$$G(\pi_i', \sigma_i) - G(\pi_i, \sigma_i) - \langle \nabla_{\pi_i} G(\pi_i, \sigma_i), \pi_i' - \pi_i \rangle \geq \min_{j \in [d_i]} \sigma_{ij} \text{IS}(\pi_i', \pi_i),$$

$$G(\pi_i', \sigma_i) - G(\pi_i, \sigma_i) - \langle \nabla_{\pi_i} G(\pi_i, \sigma_i), \pi_i' - \pi_i \rangle \leq \text{IS}(\pi_i', \pi_i).$$

Therefore, Assumption 4.1 holds with $\beta = 1$ and $\gamma = \min_{i \in [N], j \in [d_i]} \sigma_{ij}$. Hence, from Theorem 4.2, we have for $\eta \in (0, \frac{2\mu\gamma}{\mu^2 \gamma(\gamma+2) + 8L^2})$:

$$\text{IS}(\pi^{\mu,\sigma}, \pi^t) \leq \text{IS}(\pi^{\mu,\sigma}, \pi^0) \left(1 - \frac{\eta \mu \gamma}{2}\right)^t.$$

$\square$

## B.5 DETAILS OF EXAMPLE 4.6

*Details of Example 4.6.* When $\psi(\pi_i) = \sum_{j=1}^{d_i} \pi_{ij} \ln(\pi_{ij})$, we have $D_\psi(\pi_i, \pi_i') = \mathrm{KL}(\pi_i, \pi_i') = G(\pi_i, \pi_i')$ for all $\pi_i, \pi_i' \in \Delta^{d_i}$. Furthermore, $\psi$ is twice differentiable and has a diagonal Hessian:

$$\nabla^2 \psi(\pi_i) = \begin{bmatrix} \frac{1}{\pi_{i1}} & & \\ & \ddots & \\ & & \frac{1}{\pi_{id_i}}, \end{bmatrix}$$

and thus, its smallest eigenvalue is lower bounded by 1. Therefore, $\psi$ is 1-strongly convex, and then we have $\rho = 1$. By plugging $\rho = 1$ into Corollary 4.4, we have for $\eta \in (0, \frac{2\mu}{3\mu^2 + 8L^2})$:

$$\mathrm{KL}(\pi^{\mu,\sigma}, \pi^t) \le \mathrm{KL}(\pi^{\mu,\sigma}, \pi^0) \left(1 - \frac{\eta\mu}{2}\right)^t.$$

$\square$

## B.6 DETAILS OF EXAMPLE 4.7

*Details of Example 4.7.* Let us define $\sigma_i = (1/d_i)_{j \in [d_i]}$ in Example 4.6. Since the convergence rate holds for any $\sigma \in \mathcal{X}$, we have for $\eta \in (0, \frac{2\mu}{3\mu^2 + 8L^2})$:

$$\mathrm{KL}(\pi^{\mu,\sigma}, \pi^t) \le \mathrm{KL}(\pi^{\mu,\sigma}, \pi^0) \left(1 - \frac{\eta\mu}{2}\right)^t.$$

$\square$

# C PROOFS FOR SECTION 4.2

## C.1 PROOF OF THEOREM 4.9

*Proof of Theorem 4.9.* Writing $y_i^t = \sum_{s=0}^t \eta_s (\nabla_{\pi_i} v_i(\pi_i^s, \pi_{-i}^s) + \xi_i^s - \mu \nabla_{\pi_i} G(\pi_i^s, r_i))$ and using Lemma B.1 in Appendix B.1,

$$D_\psi(\pi_i^{\mu,\sigma}, \pi_i^{t+1}) - D_\psi(\pi_i^{\mu,\sigma}, \pi_i^t) + D_\psi(\pi_i^{t+1}, \pi_i^t)$$
$$= \langle y_i^t - y_i^{t-1}, \pi_i^{t+1} - \pi_i^{\mu,\sigma} \rangle$$
$$= \eta_t \langle \nabla_{\pi_i} v_i(\pi_i^t, \pi_{-i}^t) - \mu \nabla_{\pi_i} G(\pi_i^t, \sigma_i) + \xi_i^t, \pi_i^{t+1} - \pi_i^{\mu,\sigma} \rangle.$$

We have the following Lemma that replaces the gradient with Bregman divergences:

**Lemma C.1.** *Under the noisy feedback setting, suppose that Assumption 4.1 holds with $\beta, \gamma \in (0, \infty)$, and the updated strategy profile $\pi^t$ satisfies the following condition: for any $t \ge 0$,*

$$D_\psi(\pi^{\mu,\sigma}, \pi^{t+1}) - D_\psi(\pi^{\mu,\sigma}, \pi^t) + D_\psi(\pi^{t+1}, \pi^t)$$
$$\le \eta_t \sum_{i=1}^N \langle \nabla_{\pi_i} v_i(\pi_i^t, \pi_{-i}^t) - \mu \nabla_{\pi_i} G(\pi_i^t, \sigma_i) + \xi_i^t, \pi_i^{t+1} - \pi_i^{\mu,\sigma} \rangle.$$

*Then, for any $t \ge 0$:*

$$D_\psi(\pi^{\mu,\sigma}, \pi^{t+1}) - D_\psi(\pi^{\mu,\sigma}, \pi^t) + D_\psi(\pi^{t+1}, \pi^t)$$
$$\le \eta_t \left( \left( \frac{\mu^2 \gamma \rho^2 (\gamma + 2\beta) + 8L^2}{2\mu\gamma\rho^2} \right) D_\psi(\pi^{t+1}, \pi^t) - \frac{\mu\gamma}{2} D_\psi(\pi^{\mu,\sigma}, \pi^t) \right) + \eta_t \sum_{i=1}^N \langle \xi_i^t, \pi_i^{t+1} - \pi_i^{\mu,\sigma} \rangle.$$

The proof of Lemma C.1 is given in Appendix F.3. Note that $\frac{\mu^2 \gamma \rho^2 (\gamma + 2\beta) + 8L^2}{2\mu\gamma\rho^2} > \frac{\mu\gamma}{2}$. The following lemma completes the proof.

**Lemma C.2.** *Suppose that with some constants $\theta > \kappa > 0$, for all $t \geq 0$, the following inequality holds:*

$$D_\psi(\pi^{\mu,\sigma}, \pi^{t+1}) - D_\psi(\pi^{\mu,\sigma}, \pi^t) + D_\psi(\pi^{t+1}, \pi^t)$$

$$\leq \eta_t(\theta D_\psi(\pi^{t+1}, \pi^t) - \kappa D_\psi(\pi^{\mu,\sigma}, \pi^t)) + \eta_t \sum_{i=1}^{N} \langle \xi_i^t, \pi_i^{t+1} - \pi_i^{\mu,\sigma} \rangle.$$

*Then, under Assumption 4.8, for all $t \geq 0$,*

$$\mathbb{E}[D_\psi(\pi^{\mu,\sigma}, \pi^{t+1})] \leq \frac{2\theta - \kappa}{\kappa t + 2\theta} D_\psi(\pi^{\mu,\sigma}, \pi^0) + \frac{NC^2}{\rho(\kappa t + 2\theta)} \left( \frac{1}{\kappa} \ln\left( \frac{\kappa}{2\theta} t + 1 \right) + \frac{1}{2\theta} \right).$$

The proof of Lemma C.2 is given in Appendix F.4 $\qquad\qquad\qquad\qquad\qquad\square$

# D  PROOFS FOR SECTION 4.3

## D.1  PROOF OF THEOREM 4.10

*Proof of Theorem 4.10.* From the definition of the Bregman divergence, we have:

$$D_\psi(\pi_i^{\mu,\sigma}, \pi_i^{t+1}) - D_\psi(\pi_i^{\mu,\sigma}, \pi_i^t) + D_\psi(\pi_i^{t+1}, \pi_i^t)$$
$$= \psi(\pi_i^{\mu,\sigma}) - \psi(\pi_i^{t+1}) - \langle \nabla\psi(\pi_i^{t+1}), \pi_i^{\mu,\sigma} - \pi_i^{t+1} \rangle$$
$$- \psi(\pi_i^{\mu,\sigma}) + \psi(\pi_i^t) + \langle \nabla\psi(\pi_i^t), \pi_i^{\mu,\sigma} - \pi_i^t \rangle$$
$$+ \psi(\pi_i^{t+1}) - \psi(\pi_i^t) - \langle \nabla\psi(\pi_i^t), \pi_i^{t+1} - \pi_i^t \rangle$$
$$= \langle \nabla\psi(\pi_i^t) - \nabla\psi(\pi_i^{t+1}), \pi_i^{\mu,\sigma} - \pi_i^{t+1} \rangle. \qquad (11)$$

From the first-order optimality condition for $\pi_i^{t+1}$, we get:

$$\langle \eta(\nabla_{\pi_i} v_i(\pi_i^t, \pi_{-i}^t) - \mu\nabla_{\pi_i} G(\pi_i^t, \sigma_i)) - \nabla\psi(\pi_i^{t+1}) + \nabla\psi(\pi_i^t), \pi_i^{\mu,\sigma} - \pi_i^{t+1} \rangle \leq 0. \qquad (12)$$

By combining (11) and (12), we have:

$$D_\psi(\pi_i^{\mu,\sigma}, \pi_i^{t+1}) - D_\psi(\pi_i^{\mu,\sigma}, \pi_i^t) + D_\psi(\pi_i^{t+1}, \pi_i^t)$$
$$\leq \eta\langle \nabla_{\pi_i} v_i(\pi_i^t, \pi_{-i}^t) - \mu\nabla_{\pi_i} G(\pi_i^t, \sigma_i), \pi_i^{t+1} - \pi_i^{\mu,\sigma} \rangle.$$

Thus, we can apply Lemma B.2, and we conclude the statement. $\qquad\qquad\qquad\square$

## D.2  PROOF OF THEOREM 4.11

*Proof of Theorem 4.11.* Writing $g_i^t = \nabla_{\pi_i} v_i(\pi_i^t, \pi_{-i}^t) - \mu\nabla_{\pi_i} G(\pi_i^t, \sigma_i)$, from the first-order optimality condition for $\pi_i^{t+1}$, we get:

$$\langle \eta_t(g_i^t + \xi_i^t) - \nabla\psi(\pi_i^{t+1}) + \nabla\psi(\pi_i^t), \pi_i^{\mu,\sigma} - \pi_i^{t+1} \rangle \leq 0. \qquad (13)$$

By combining (11) and (13), we have:

$$D_\psi(\pi_i^{\mu,\sigma}, \pi_i^{t+1}) - D_\psi(\pi_i^{\mu,\sigma}, \pi_i^t) + D_\psi(\pi_i^{t+1}, \pi_i^t) \leq \eta_t\langle g_i^t + \xi_i^t, \pi_i^{t+1} - \pi_i^{\mu,\sigma} \rangle.$$

Thus, we can apply Lemma C.1, and we have for all $t \geq 0$,

$$D_\psi(\pi^{\mu,r}, \pi^{t+1}) - D_\psi(\pi^{\mu,\sigma}, \pi^t) + D_\psi(\pi^{t+1}, \pi^t)$$

$$\leq \eta_t\left( \left( \frac{\mu^2\gamma\rho^2(\gamma + 2\beta) + 8L^2}{2\mu\gamma\rho^2} \right) D_\psi(\pi^{t+1}, \pi^t) - \frac{\mu\gamma}{2} D_\psi(\pi^{\mu,\sigma}, \pi^t) \right) + \eta_t \sum_{i=1}^{N} \langle \xi_i^t, \pi_i^{t+1} - \pi_i^{\mu,\sigma} \rangle$$

$$= \eta_t(\theta D_\psi(\pi^{t+1}, \pi^t) - \kappa D_\psi(\pi^{\mu,\sigma}, \pi^t)) + \eta_t \sum_{i=1}^{N} \langle \xi_i^t, \pi_i^{t+1} - \pi_i^{\mu,\sigma} \rangle.$$

By applying Lemma C.2,

$$\mathbb{E}[D_\psi(\pi^{\mu,\sigma}, \pi^{t+1})] \leq \frac{2\theta - \kappa}{\kappa t + 2\theta} D_\psi(\pi^{\mu,\sigma}, \pi^0) + \frac{NC^2}{\rho(\kappa t + 2\theta)} \left( \frac{1}{\kappa} \ln\left( \frac{\kappa}{2\theta} t + 1 \right) + \frac{1}{2\theta} \right).$$

This concludes the proof. $\qquad\qquad\qquad\qquad\qquad\qquad\qquad\qquad\qquad\square$

# E PROOFS FOR SECTION 5

## E.1 PROOF OF THEOREM 5.1

*Proof of Theorem 5.1.* First, we prove that exploitability can be decomposed as follows:

**Lemma E.1.** *Assume that* $\sqrt{\sum_{i=1}^{N}\|\nabla_{\pi_i}v_i(\pi)\|^2} \leq \zeta$ *for any* $\pi \in \mathcal{X}$*. Then, we have for any* $\pi, \pi' \in \mathcal{X}$*:*

$$\text{exploit}(\pi) \leq \text{diam}(\mathcal{X}) \cdot \min_{(a_i) \in N_{\mathcal{X}}(\pi')} \sqrt{\sum_{i=1}^{N}\| - \nabla_{\pi_i}v_i(\pi') + a_i\|^2} + (L \cdot \text{diam}(\mathcal{X}) + \zeta)\|\pi - \pi'\|.$$

From Lemma E.1, we have:

$$\text{exploit}(\sigma^K)$$

$$\leq \text{diam}(\mathcal{X}) \cdot \min_{(a_i) \in N_{\mathcal{X}}(\pi^{\mu,\sigma^{K-1}})} \sqrt{\sum_{i=1}^{N}\| - \nabla_{\pi_i}v_i(\pi^{\mu,\sigma^{K-1}}) + a_i\|^2} + (L \cdot \text{diam}(\mathcal{X}) + \zeta)\|\sigma^K - \pi^{\mu,\sigma^{K-1}}\|.$$

$$(14)$$

From the first-order optimality condition for $\pi^{\mu,\sigma^{K-1}}$, we have for any $\pi \in \mathcal{X}$:

$$\sum_{i=1}^{N}\langle\nabla_{\pi_i}v_i(\pi^{\mu,\sigma^{K-1}}) - \mu\left(\pi_i^{\mu,\sigma^{K-1}} - \sigma_i^{K-1}\right), \pi_i - \pi_i^{\mu,\sigma^{K-1}}\rangle \leq 0,$$

and then $\left(\nabla_{\pi_i}v_i(\pi^{\mu,\sigma^{K-1}}) - \mu\left(\pi_i^{\mu,\sigma^{K-1}} - \sigma_i^{K-1}\right)\right)_{i \in [N]} \in N_{\mathcal{X}}(\pi^{\mu,\sigma^{K-1}})$. Thus, the first term of (14) can be bounded as:

$$\min_{(a_i) \in N_{\mathcal{X}}(\pi^{\mu,\sigma^{K-1}})} \sqrt{\sum_{i=1}^{N}\| - \nabla_{\pi_i}v_i(\pi^{\mu,\sigma^{K-1}}) + a_i\|^2}$$

$$\leq \sqrt{\sum_{i=1}^{N}\| - \nabla_{\pi_i}v_i(\pi^{\mu,\sigma^{K-1}}) + \nabla_{\pi_i}v_i(\pi^{\mu,\sigma^{K-1}}) - \mu\left(\pi_i^{\mu,\sigma^{K-1}} - \sigma_i^{K-1}\right)\|^2}$$

$$= \mu\|\pi^{\mu,\sigma^{K-1}} - \sigma^{K-1}\|.$$

Here, $\|\pi^{\mu,\sigma^{K-1}} - \sigma^{K-1}\|$ can be upper bounded as follows:

**Lemma E.2.** *Assume that* $\|\pi^{\mu,\sigma^k} - \sigma^{k+1}\| \leq \|\pi^{\mu,\sigma^k} - \sigma^k\|\left(\frac{1}{c}\right)^{T_\sigma}$ *for some* $c > 1$*, and* $\sqrt{\sum_{i=1}^{N}\|\nabla_{\pi_i}v_i(\pi)\|^2} \leq \zeta$ *for any* $\pi \in \mathcal{X}$*. Then, if we set* $T_\sigma = \max(\frac{3}{\ln c}\ln K + \frac{\ln 64}{\ln c}, 1)$*, we have for any Nash equilibrium* $\pi^* \in \Pi^*$*:*

$$\|\pi^{\mu,\sigma^{K-1}} - \sigma^{K-1}\| \leq \frac{2\sqrt{2}}{\sqrt{K}}\sqrt{\|\pi^* - \sigma^0\|\left(8\|\pi^* - \sigma^0\| + \frac{\zeta}{\mu}\right)}.$$

By using Lemma E.2, the first term of (14) can be upper bounded as:

$$\min_{(a_i) \in N_{\mathcal{X}}(\pi^{\mu,\sigma^{K-1}})} \sqrt{\sum_{i=1}^{N}\| - \nabla_{\pi_i}v_i(\pi^{\mu,\sigma^{K-1}}) + a_i\|^2} \leq \frac{2\sqrt{2}\mu}{\sqrt{K}}\sqrt{\|\pi^* - \sigma^0\|\left(8\|\pi^* - \sigma^0\| + \frac{\zeta}{\mu}\right)}$$

$$(15)$$

Next, from the assumption that $\|\pi^{\mu,\sigma^k} - \sigma^{k+1}\| \leq \|\pi^{\mu,\sigma^k} - \sigma^k\| \left(\frac{1}{c}\right)^{T_\sigma}$ and Lemma E.2, the second term of (14) can be bounded as:

$$
\begin{aligned}
\|\sigma^K - \pi^{\mu,\sigma^{K-1}}\| &\leq \frac{\|\sigma^{K-1} - \pi^{\mu,\sigma^{K-1}}\|}{c^{T_\sigma}} \\
&\leq \frac{2\sqrt{2}\zeta}{c^{T_\sigma}\sqrt{K}} \sqrt{\|\pi^* - \sigma^0\| \left(8\|\pi^* - \sigma^0\| + \frac{\zeta}{\mu}\right)} \\
&\leq \frac{2\sqrt{2}\zeta}{\sqrt{K}} \sqrt{\|\pi^* - \sigma^0\| \left(8\|\pi^* - \sigma^0\| + \frac{\zeta}{\mu}\right)}.
\end{aligned}
\tag{16}
$$

By combining (14), (15), and (16), we get:

$$
\text{exploit}(\sigma^K) \leq \frac{2\sqrt{2}\left((\mu + L) \cdot \text{diam}(\mathcal{X}) + \zeta\right)}{\sqrt{K}} \sqrt{\|\pi^* - \sigma^0\| \left(8\|\pi^* - \sigma^0\| + \frac{\zeta}{\mu}\right)}.
$$

$\square$

### E.2 PROOF OF THEOREM 5.2

*Proof of Theorem 5.2.* From Lemma E.1, we have:

$\mathbb{E}[\text{exploit}(\sigma^K)]$

$$
\leq \mathbb{E}\left[\text{diam}(\mathcal{X}) \cdot \min_{(a_i) \in N_\mathcal{X}(\pi^{\mu,\sigma^{K-1}})} \sqrt{\sum_{i=1}^{N} \| - \nabla_{\pi_i} v_i(\pi^{\mu,\sigma^{K-1}}) + a_i\|^2} + (L \cdot \text{diam}(\mathcal{X}) + \zeta)\|\sigma^K - \pi^{\mu,\sigma^{K-1}}\|\right].
\tag{17}
$$

From the first-order optimality condition for $\pi^{\mu,\sigma^{K-1}}$, for any $\pi \in \mathcal{X}$:

$$
\sum_{i=1}^{N} \langle \nabla_{\pi_i} v_i(\pi^{\mu,\sigma^{K-1}}) - \mu\left(\pi_i^{\mu,\sigma^{K-1}} - \sigma_i^{K-1}\right), \pi_i - \pi_i^{\mu,\sigma^{K-1}} \rangle \leq 0,
$$

and then $\left(\nabla_{\pi_i} v_i(\pi^{\mu,\sigma^{K-1}}) - \mu\left(\pi_i^{\mu,\sigma^{K-1}} - \sigma_i^{K-1}\right)\right)_{i \in [N]} \in N_\mathcal{X}(\pi^{\mu,\sigma^{K-1}})$. The first term of (17) can be bounded as:

$$
\begin{aligned}
&\mathbb{E}\left[\min_{(a_i) \in N_\mathcal{X}(\pi^{\mu,\sigma^{K-1}})} \sqrt{\sum_{i=1}^{N} \| - \nabla_{\pi_i} v_i(\pi^{\mu,\sigma^{K-1}}) + a_i\|^2}\right] \\
&\leq \mathbb{E}\left[\sqrt{\sum_{i=1}^{N} \| - \nabla_{\pi_i} v_i(\pi^{\mu,\sigma^{K-1}}) + \nabla_{\pi_i} v_i(\pi^{\mu,\sigma^{K-1}}) - \mu\left(\pi_i^{\mu,\sigma^{K-1}} - \sigma_i^{K-1}\right)\|^2}\right] \\
&= \mu\mathbb{E}\left[\|\pi^{\mu,\sigma^{K-1}} - \sigma^{K-1}\|\right].
\end{aligned}
$$

Here, $\mathbb{E}\left[\mu\|\pi^{\mu,\sigma^{K-1}} - \sigma^{K-1}\|\right]$ can be upper bounded as follows:

**Lemma E.3.** *Assume that $\mathbb{E}[\|\pi^{\mu,\sigma^k} - \sigma^{k+1}\|^2 \mid \sigma^k] \leq c^2 \frac{\ln T_\sigma}{T_\sigma}$ for any $k \geq 0$ for some $c > 0$, and $\sqrt{\sum_{i=1}^{N} \|\nabla_{\pi_i} v_i(\pi)\|^2} \leq \zeta$ for any $\pi \in \mathcal{X}$. Then, if we set $T_\sigma \geq \max(K^4, 3)$, we have for any Nash equilibrium $\pi^* \in \Pi^*$:*

$$
\mathbb{E}[\|\pi^{\mu,\sigma^{K-1}} - \sigma^{K-1}\|] \leq \sqrt{\frac{\|\pi^* - \sigma^0\|^2 + 4c\left(2\text{diam}(\mathcal{X}) + \frac{\zeta}{\mu} + c\right)\ln K}{K}}.
$$

By using Lemma E.3, the first term of (17) can be bounded as:

$$
\mathbb{E}\left[\min_{(a_i)\in N_{\mathcal{X}}(\pi^{\mu,\sigma^{K-1}})}\sqrt{\sum_{i=1}^{N}\| -\nabla_{\pi_i}v_i(\pi^{\mu,\sigma^{K-1}})+a_i\|^2}\right]
$$
$$
\leq \mu\sqrt{\frac{\|\pi^*-\sigma^0\|^2+4c\left(2\mathrm{diam}(\mathcal{X})+\frac{\varsigma}{\mu}+c\right)\ln K}{K}}. \tag{18}
$$

Next, from the assumption that $\mathbb{E}[\|\pi^{\mu,\sigma^k}-\sigma^{k+1}\| \mid \sigma^k]\leq c\sqrt{\frac{\ln T_\sigma}{T_\sigma}}\leq 2c\frac{\sqrt{\ln K}}{K^2}$ and Lemma E.3, the second term of (17) can be bounded as:

$$
\mathbb{E}[\|\sigma^K-\pi^{\mu,\sigma^{K-1}}\|]\leq c\sqrt{\frac{\ln T_\sigma}{T_\sigma}}\leq 2c\frac{\sqrt{\ln K}}{K^2}. \tag{19}
$$

By combining (17), (18), and (19), we get:

$\mathbb{E}[\mathrm{exploit}(\sigma^K)]$

$$
\leq \mu\cdot\mathrm{diam}(\mathcal{X})\sqrt{\frac{\|\pi^*-\sigma^0\|^2+4c\left(2\mathrm{diam}(\mathcal{X})+\frac{\varsigma}{\mu}+c\right)\ln K}{K}}
$$
$$
+2c\left(L\cdot\mathrm{diam}(\mathcal{X})+\zeta\right)\frac{\sqrt{\ln K}}{K^2}
$$
$$
\leq \frac{\mu\cdot\mathrm{diam}(\mathcal{X})\|\pi^*-\sigma^0\|}{\sqrt{K}}+2\mu\cdot\mathrm{diam}(\mathcal{X})\sqrt{\frac{c\left(2\mathrm{diam}(\mathcal{X})+\frac{\varsigma}{\mu}+c\right)\ln K}{K}}
$$
$$
+2c\left(L\cdot\mathrm{diam}(\mathcal{X})+\zeta\right)\frac{\sqrt{\ln K}}{K^2}
$$
$$
\leq \frac{\mu\cdot\mathrm{diam}(\mathcal{X})\|\pi^*-\sigma^0\|}{\sqrt{K}}+2\left(\mathrm{diam}(\mathcal{X})\cdot\left(cL+\mu\sqrt{c\left(2\mathrm{diam}(\mathcal{X})+\frac{\varsigma}{\mu}+c\right)}\right)+c\zeta\right)\sqrt{\frac{\ln K}{K}}
$$
$$
\leq \frac{2\left(\mathrm{diam}(\mathcal{X})\cdot\left(cL+\mu\sqrt{c}\sqrt{2\mathrm{diam}(\mathcal{X})+\frac{\varsigma}{\mu}+c}\right)+c\zeta\right)\ln K+\mu\cdot\mathrm{diam}(\mathcal{X})^2}{\sqrt{K}}.
$$

$\square$

### E.3  PROOF OF THEOREM G.1

*Proof of Theorem G.1.* When $G(\pi_i,\pi_i')=D_{\psi'}(\pi_i,\pi_i')$ for all $i\in[N]$ and $\pi,\pi'\in\mathcal{X}$, we can show that the Bregman divergence from a Nash equilibrium $\pi^*\in\Pi^*$ to $\sigma^{k+1}$ monotonically decreases:

**Lemma E.4.** *Assume that $G$ is a Bregman divergence $D_{\psi'}$ for some strongly convex function $\psi'$. Then, for any Nash equilibrium $\pi^*\in\Pi^*$ of the original game, we have for any $k\geq 0$:*
$$
D_{\psi'}(\pi^*,\sigma^{k+1})-D_{\psi'}(\pi^*,\sigma^k)\leq -D_{\psi'}(\sigma^{k+1},\sigma^k).
$$

By summing the inequality in Lemma E.4 from $k=0$ to $K$, we have:

$$
D_{\psi'}(\pi^*,\sigma^0)\geq \sum_{k=0}^{K}D_{\psi'}(\sigma^{k+1},\sigma^k)\geq \frac{\rho}{2}\sum_{k=0}^{K}\|\sigma^{k+1}-\sigma^k\|^2,
$$

where the second inequality follows from the strong convexity of $\psi'$. Therefore, $\sum_{k=0}^{\infty}\|\sigma^{k+1}-\sigma^k\|^2<\infty$, which implies that $\|\sigma^{k+1}-\sigma^k\|\to 0$ as $k\to\infty$.

By the compactness of $\mathcal{X}$ and Bolzano–Weierstrass theorem, there exists a subsequence $k_n$ and a limit point $\hat{\sigma}\in\mathcal{X}$ such that $\sigma^{k_n}\to\hat{\sigma}$ as $n\to\infty$. Since $\|\sigma^{k_n+1}-\sigma^{k_n}\|\to 0$ as $n\to\infty$, we have $\sigma^{k_n+1}\to\hat{\sigma}$ as $n\to\infty$. Thus, the limit point $\hat{\sigma}$ is the fixed point of the updating rule. From the following lemma, we show that the fixed point $\hat{\sigma}$ is a Nash equilibrium of the original game:

**Lemma E.5.** *Assume that $G$ is a Bregman divergence $D_{\psi'}$ for some strongly convex function $\psi'$, and $\sigma^{k+1} = \pi^{\mu,\sigma^k}$ for $k \geq 0$. If $\sigma^{k+1} = \sigma^k$, then $\sigma^k$ is a Nash equilibrium of the original game.*

On the other hand, by summing the inequality in Lemma E.4 from $k = k_n$ to $k = K - 1$ for $K \geq k_n + 1$, we have:

$$0 \leq D_{\psi'}(\hat{\sigma}, \sigma^K) \leq D_{\psi'}(\hat{\sigma}, \sigma^{k_n}).$$

Since $\sigma^{k_n} \to \hat{\sigma}$ as $n \to \infty$, we have $\sigma^K \to \hat{\sigma}$ as $K \to \infty$. Since $\hat{\sigma}$ is a Nash equilibrium of the original game, we conclude the first statement of the theorem.

$\square$

### E.4 Proof of Theorem G.2

*Proof of Theorem G.2.* We first show that the divergence between $\Pi^*$ and $\sigma^k$ decreases monotonically as $k$ increases:

**Lemma E.6.** *Suppose that the same assumptions in Theorem G.2 hold. For any $k \geq 0$, if $\sigma^k \in \mathcal{X} \setminus \Pi^*$, then:*

$$\min_{\pi^* \in \Pi^*} \mathrm{KL}(\pi^*, \sigma^{k+1}) < \min_{\pi^* \in \Pi^*} \mathrm{KL}(\pi^*, \sigma^k).$$

*Otherwise, if $\sigma^k \in \Pi^*$, then $\sigma^{k+1} = \sigma^k \in \Pi^*$.*

From Lemma E.6, the sequence $\{\min_{\pi^* \in \Pi^*} \mathrm{KL}(\pi^*, \sigma^k)\}_{k \geq 0}$ is a monotonically decreasing sequence and is bounded from below by zero. Thus, $\{\min_{\pi^* \in \Pi^*} \mathrm{KL}(\pi^*, \sigma^k)\}_{k \geq 0}$ converges to some constant $b \geq 0$. We show that $b = 0$ by a contradiction argument.

Suppose $b > 0$ and let us define $B = \min_{\pi^* \in \Pi^*} \mathrm{KL}(\pi^*, \sigma^0)$. Since $\min_{\pi^* \in \Pi^*} \mathrm{KL}(\pi^*, \sigma^k)$ monotonically decreases, $\sigma^k$ is in the set $\Omega_{b,B} = \{\sigma \in \mathcal{X} \mid b \leq \min_{\pi^* \in \Pi^*} \mathrm{KL}(\pi^*, \sigma) \leq B\}$ for all $k \geq 0$. Since $\min_{\pi^* \in \Pi^*} \mathrm{KL}(\pi^*, \cdot)$ is a continuous function on $\mathcal{X}$, the preimage $\Omega_{b,B}$ of the closed set $[b, B]$ is also closed. Furthermore, since $\mathcal{X}$ is compact and then bounded, $\Omega_{b,B}$ is a bounded set. Thus, $\Omega_{b,B}$ is a compact set.

Next, we show that the function which maps the slingshot strategies $\sigma$ to the associated stationary point $\pi^{\mu,\sigma}$ is continuous:

**Lemma E.7.** *Let $F(\sigma) : \mathcal{X} \to \mathcal{X}$ be a function that maps the slingshot strategies $\sigma$ to the stationary point $\pi^{\mu,\sigma}$ defined by (4). In the same setup of Theorem G.2, $F(\cdot)$ is a continuous function on $\mathcal{X}$.*

From Lemma E.7, $\min_{\pi^* \in \Pi^*} \mathrm{KL}(\pi^*, F(\sigma)) - \min_{\pi^* \in \Pi^*} \mathrm{KL}(\pi^*, \sigma)$ is also a continuous function. Since a continuous function has a maximum over a compact set $\Omega_{b,B}$, the maximum $\delta = \max_{\sigma \in \Omega_{b,B}} \{\min_{\pi^* \in \Pi^*} \mathrm{KL}(\pi^*, F(\sigma)) - \min_{\pi^* \in \Pi^*} \mathrm{KL}(\pi^*, \sigma)\}$ exists. From Lemma E.6 and the assumption that $b > 0$, we have $\delta < 0$. It follows that:

$$\min_{\pi^* \in \Pi^*} \mathrm{KL}(\pi^*, \sigma^k) = \min_{\pi^* \in \Pi^*} \mathrm{KL}(\pi^*, \sigma^0) + \sum_{l=0}^{k-1} \left( \min_{\pi^* \in \Pi^*} \mathrm{KL}(\pi^*, \sigma^{l+1}) - \min_{\pi^* \in \Pi^*} \mathrm{KL}(\pi^*, \sigma^l) \right)$$

$$\leq B + \sum_{l=0}^{k-1} \delta = B + k\delta.$$

This implies that $\min_{\pi^* \in \Pi^*} \mathrm{KL}(\pi^*, \sigma^k) < 0$ for $k > \frac{B}{-\delta}$, which contradicts $\min_{\pi^* \in \Pi^*} \mathrm{KL}(\pi^*, \sigma) \geq 0$. Therefore, the sequence $\{\min_{\pi^* \in \Pi^*} \mathrm{KL}(\pi^*, \sigma^k)\}_{k \geq 0}$ converges to 0, and $\sigma^k$ converges to $\Pi^*$.

$\square$

# F PROOFS FOR ADDITIONAL LEMMAS

## F.1 PROOF OF LEMMA B.1

*Proof of Lemma B.1.* First, from the definition of the Bregman divergence, for any $\pi_i \in \mathcal{X}_i$:

$$D_\psi(\pi, T(y_i)) = \psi(\pi) - \psi(T(y_i)) - \langle \nabla \psi(T(y_i)), \pi_i - T(y_i) \rangle. \tag{20}$$

Recall that $\mathcal{X}_i$ satisfies $A\pi_i = b$ for all $\pi_i \in \mathcal{X}_i$ for a matrix $A \in \mathbb{R}^{k_i \times d_i}$ and $b \in \mathbb{R}^{k_i}$. From the assumption for $\psi$ and the first-order optimality condition for the optimization problem of $\arg\max_{x \in \mathcal{X}}\{\langle y_i, x \rangle - \psi(x)\}$, there exists $\nu \in \mathbb{R}^{k_i}$ such that

$$y_i - \nabla \psi(T(y_i)) = A^\top \nu.$$

Thus, we get:

$$\begin{aligned}
\langle y_i, \pi_i - T(y_i) \rangle &= \langle A^\top \nu + \nabla \psi(T(y_i)), \pi_i - T(y_i) \rangle \\
&= \langle \nabla \psi(T(y_i)), \pi_i - T(y_i) \rangle + \nu^\top A \pi_i - \nu^\top A T(y_i) \\
&= \langle \nabla \psi(T(y_i)), \pi_i - T(y_i) \rangle + \nu^\top b - \nu^\top b \\
&= \langle \nabla \psi(T(y_i)), \pi_i - T(y_i) \rangle. 
\end{aligned} \tag{21}$$

By combining (20) and (21), we have:

$$D_\psi(\pi, T(y_i)) = \psi(\pi) - \psi(T(y_i)) - \langle y_i, \pi_i - T(y_i) \rangle.$$

$\square$

## F.2 PROOF OF LEMMA B.2

*Proof of Lemma B.2.* We first decompose the inequality in the assumption as follows:

$$\begin{aligned}
&D_\psi(\pi_i^{\mu,\sigma}, \pi_i^{t+1}) - D_\psi(\pi_i^{\mu,\sigma}, \pi_i^t) + D_\psi(\pi_i^{t+1}, \pi_i^t) \\
&\leq \eta \langle \nabla_{\pi_i} v_i(\pi_i^t, \pi_{-i}^t) - \mu \nabla_{\pi_i} G(\pi_i^t, \sigma_i), \pi_i^{t+1} - \pi_i^{\mu,\sigma} \rangle \\
&= \eta \langle \nabla_{\pi_i} v_i(\pi_i^t, \pi_{-i}^t), \pi_i^{t+1} - \pi_i^{\mu,\sigma} \rangle + \eta\mu \langle \nabla_{\pi_i} G(\pi_i^t, \sigma_i), \pi_i^t - \pi_i^{t+1} \rangle + \eta\mu \langle \nabla_{\pi_i} G(\pi_i^t, \sigma_i), \pi_i^{\mu,\sigma} - \pi_i^t \rangle. 
\end{aligned} \tag{22}$$

From the relative smoothness in Assumption 4.1 and the convexity of $G(\cdot, \sigma_i)$:

$$\begin{aligned}
&\langle \nabla_{\pi_i} G(\pi_i^t, \sigma_i), \pi_i^t - \pi_i^{t+1} \rangle \\
&\leq G(\pi_i^t, \sigma_i) - G(\pi_i^{t+1}, \sigma_i) + \beta D_\psi(\pi_i^{t+1}, \pi_i^t) \\
&\leq G(\pi_i^t, \sigma_i) - G(\pi_i^{\mu,\sigma}, \sigma_i) + \langle \nabla_{\pi_i} G(\pi_i^{\mu,\sigma}, \sigma_i), \pi_i^{\mu,\sigma} - \pi_i^{t+1} \rangle + \beta D_\psi(\pi_i^{t+1}, \pi_i^t). 
\end{aligned} \tag{23}$$

Also, from the relative strong convexity in Assumption 4.1:

$$G(\pi_i^t, \sigma_i) - G(\pi_i^{\mu,\sigma}, \sigma_i) \leq \langle \nabla_{\pi_i} G(\pi_i^t, \sigma_i), \pi_i^t - \pi_i^{\mu,\sigma} \rangle - \gamma D_\psi(\pi_i^{\mu,\sigma}, \pi_i^t). \tag{24}$$

By combining (22), (23), and (24), we have:

$$\begin{aligned}
&D_\psi(\pi_i^{\mu,\sigma}, \pi_i^{t+1}) - D_\psi(\pi_i^{\mu,\sigma}, \pi_i^t) + D_\psi(\pi_i^{t+1}, \pi_i^t) \\
&\leq \eta \langle \nabla_{\pi_i} v_i(\pi_i^t, \pi_{-i}^t), \pi_i^{t+1} - \pi_i^{\mu,\sigma} \rangle + \eta\mu \langle \nabla_{\pi_i} G(\pi_i^{\mu,\sigma}, \sigma_i), \pi_i^{\mu,\sigma} - \pi_i^{t+1} \rangle \\
&\quad - \eta\mu\gamma D_\psi(\pi_i^{\mu,\sigma}, \pi_i^t) + \eta\mu\beta D_\psi(\pi_i^{t+1}, \pi_i^t), 
\end{aligned}$$

and then:

$$\begin{aligned}
&D_\psi(\pi_i^{\mu,\sigma}, \pi_i^{t+1}) - (1 - \eta\mu\gamma) D_\psi(\pi_i^{\mu,\sigma}, \pi_i^t) + (1 - \eta\mu\beta) D_\psi(\pi_i^{t+1}, \pi_i^t) \\
&\leq \eta \langle \nabla_{\pi_i} v_i(\pi_i^t, \pi_{-i}^t), \pi_i^{t+1} - \pi_i^{\mu,\sigma} \rangle + \eta\mu \langle \nabla_{\pi_i} G(\pi_i^{\mu,\sigma}, \sigma_i), \pi_i^{\mu,\sigma} - \pi_i^{t+1} \rangle \\
&= \eta \langle \nabla_{\pi_i} v_i(\pi_i^{t+1}, \pi_{-i}^{t+1}), \pi_i^{t+1} - \pi_i^{\mu,\sigma} \rangle + \eta\mu \langle \nabla_{\pi_i} G(\pi_i^{\mu,\sigma}, \sigma_i), \pi_i^{\mu,\sigma} - \pi_i^{t+1} \rangle \\
&\quad + \eta \langle \nabla_{\pi_i} v_i(\pi_i^t, \pi_{-i}^t) - \nabla_{\pi_i} v_i(\pi_i^{t+1}, \pi_{-i}^{t+1}), \pi_i^{t+1} - \pi_i^{\mu,\sigma} \rangle. 
\end{aligned}$$

Summing this inequality from $i = 1$ to $N$ implies that:

$$D_\psi(\pi^{\mu,\sigma}, \pi^{t+1}) - (1 - \eta\mu\gamma)D_\psi(\pi^{\mu,\sigma}, \pi^t) + (1 - \eta\mu\beta)D_\psi(\pi^{t+1}, \pi^t)$$

$$\leq \eta\sum_{i=1}^{N}\langle\nabla_{\pi_i}v_i(\pi_i^{t+1}, \pi_{-i}^{t+1}), \pi_i^{t+1} - \pi_i^{\mu,\sigma}\rangle + \eta\mu\sum_{i=1}^{N}\langle\nabla_{\pi_i}G(\pi_i^{\mu,\sigma}, \sigma_i), \pi_i^{\mu,\sigma} - \pi_i^{t+1}\rangle$$

$$+ \eta\sum_{i=1}^{N}\langle\nabla_{\pi_i}v_i(\pi_i^t, \pi_{-i}^t) - \nabla_{\pi_i}v_i(\pi_i^{t+1}, \pi_{-i}^{t+1}), \pi_i^{t+1} - \pi_i^{\mu,\sigma}\rangle$$

$$\leq \eta\sum_{i=1}^{N}\langle\nabla_{\pi_i}v_i(\pi_i^{\mu,\sigma}, \pi_{-i}^{\mu,\sigma}) - \mu\nabla_{\pi_i}G(\pi_i^{\mu,\sigma}, \sigma_i), \pi_i^{t+1} - \pi_i^{\mu,\sigma}\rangle$$

$$+ \eta\sum_{i=1}^{N}\langle\nabla_{\pi_i}v_i(\pi_i^t, \pi_{-i}^t) - \nabla_{\pi_i}v_i(\pi_i^{t+1}, \pi_{-i}^{t+1}), \pi_i^{t+1} - \pi_i^{\mu,\sigma}\rangle$$

$$\leq \eta\sum_{i=1}^{N}\langle\nabla_{\pi_i}v_i(\pi_i^t, \pi_{-i}^t) - \nabla_{\pi_i}v_i(\pi_i^{t+1}, \pi_{-i}^{t+1}), \pi_i^{t+1} - \pi_i^{\mu,\sigma}\rangle, \tag{25}$$

where the second inequality follows from (1), and the third inequality follows from the first-order optimality condition for $\pi^{\mu,\sigma}$.

Here, from Young's inequality, we have for any $\lambda > 0$:

$$\sum_{i=1}^{N}\langle\nabla_{\pi_i}v_i(\pi_i^t, \pi_{-i}^t) - \nabla_{\pi_i}v_i(\pi_i^{t+1}, \pi_{-i}^{t+1}), \pi_i^{t+1} - \pi_i^{\mu,\sigma}\rangle$$

$$= \sum_{i=1}^{N}\langle\nabla_{\pi_i}v_i(\pi_i^t, \pi_{-i}^t) - \nabla_{\pi_i}v_i(\pi_i^{t+1}, \pi_{-i}^{t+1}), \pi_i^{t+1} - \pi_i^t\rangle + \sum_{i=1}^{N}\langle\nabla_{\pi_i}v_i(\pi_i^t, \pi_{-i}^t) - \nabla_{\pi_i}v_i(\pi_i^{t+1}, \pi_{-i}^{t+1}), \pi_i^t - \pi_i^{\mu,\sigma}\rangle$$

$$\leq \lambda\sum_{i=1}^{N}\|\nabla_{\pi_i}v_i(\pi_i^{t+1}, \pi_{-i}^{t+1}) - \nabla_{\pi_i}v_i(\pi_i^t, \pi_{-i}^t)\|^2 + \frac{1}{2\lambda}\|\pi^{t+1} - \pi^t\|^2 + \frac{1}{2\lambda}\|\pi^t - \pi^{\mu,\sigma}\|^2$$

$$\leq \left(L^2\lambda + \frac{1}{2\lambda}\right)\|\pi^{t+1} - \pi^t\|^2 + \frac{1}{2\lambda}\|\pi^t - \pi^{\mu,\sigma}\|^2$$

$$\leq \frac{1}{\rho}\left(2L^2\lambda + \frac{1}{\lambda}\right)D_\psi(\pi^{t+1}, \pi^t) + \frac{1}{\rho\lambda}D_\psi(\pi^{\mu,\sigma}, \pi^t). \tag{26}$$

where the second inequality follows from (2), and the fourth inequality follows from the strong convexity of $\psi$.

By combining (25) and (26), we get:

$$D_\psi(\pi^{\mu,\sigma}, \pi^{t+1}) \leq \left(1 - \eta\left(\mu\gamma - \frac{1}{\rho\lambda}\right)\right)D_\psi(\pi^{\mu,\sigma}, \pi^t) - \left(1 - \eta\left(\mu\beta + \frac{2L^2\lambda}{\rho} + \frac{1}{\rho\lambda}\right)\right)D_\psi(\pi^{t+1}, \pi^t).$$

By setting $\lambda = \frac{2}{\mu\gamma\rho}$,

$$D_\psi(\pi^{\mu,\sigma}, \pi^{t+1}) \leq \left(1 - \frac{\eta\mu\gamma}{2}\right)D_\psi(\pi^{\mu,\sigma}, \pi^t) - \left(1 - \eta\left(\frac{\mu(\gamma + 2\beta)}{2} + \frac{4L^2}{\mu\gamma\rho^2}\right)\right)D_\psi(\pi^{t+1}, \pi^t)$$

$$= \left(1 - \frac{\eta\mu\gamma}{2}\right)D_\psi(\pi^{\mu,\sigma}, \pi^t) - \left(1 - \eta\left(\frac{\mu^2\gamma\rho^2(\gamma + 2\beta) + 8L^2}{2\mu\gamma\rho^2}\right)\right)D_\psi(\pi^{t+1}, \pi^t).$$

Thus, when $\eta \leq \frac{2\mu\gamma\rho^2}{\mu^2\gamma\rho^2(\gamma+2\beta)+8L^2} < \frac{2}{\mu\gamma}$, we have for all $t \geq 0$:

$$D_\psi(\pi^{\mu,\sigma}, \pi^{t+1}) \leq \left(1 - \frac{\eta\mu\gamma}{2}\right)D_\psi(\pi^{\mu,\sigma}, \pi^t) \leq \left(1 - \frac{\eta\mu\gamma}{2}\right)^{t+1}D_\psi(\pi^{\mu,\sigma}, \pi^0).$$

$$\square$$

### F.3   PROOF OF LEMMA C.1

*Proof of Lemma C.1.* We first decompose the inequality in the assumption as follows:

$$
\begin{aligned}
&D_\psi(\pi_i^{\mu,\sigma}, \pi_i^{t+1}) - D_\psi(\pi_i^{\mu,\sigma}, \pi_i^t) + D_\psi(\pi_i^{t+1}, \pi_i^t) \\
&\leq \eta_t \langle \nabla_{\pi_i} v_i(\pi_i^t, \pi_{-i}^t) - \mu \nabla_{\pi_i} G(\pi_i^t, \sigma_i) + \xi_i^t, \pi_i^{t+1} - \pi_i^{\mu,\sigma} \rangle \\
&= \eta_t \langle \nabla_{\pi_i} v_i(\pi_i^t, \pi_{-i}^t), \pi_i^{t+1} - \pi_i^{\mu,\sigma} \rangle + \eta_t \mu \langle \nabla_{\pi_i} G(\pi_i^t, \sigma_i), \pi_i^t - \pi_i^{t+1} \rangle \\
&\quad + \eta_t \mu \langle \nabla_{\pi_i} G(\pi_i^t, \sigma_i), \pi_i^{\mu,\sigma} - \pi_i^t \rangle + \langle \xi_i^t, \pi_i^{t+1} - \pi_i^{\mu,\sigma} \rangle.
\end{aligned}
\tag{27}
$$

By combining (23), (24) in Appendix F.2, and (27),

$$
\begin{aligned}
&D_\psi(\pi_i^{\mu,\sigma}, \pi_i^{t+1}) - D_\psi(\pi_i^{\mu,\sigma}, \pi_i^t) + D_\psi(\pi_i^{t+1}, \pi_i^t) \\
&\leq \eta_t \langle \nabla_{\pi_i} v_i(\pi_i^t, \pi_{-i}^t), \pi_i^{t+1} - \pi_i^{\mu,\sigma} \rangle + \eta_t \mu \langle \nabla_{\pi_i} G(\pi_i^{\mu,\sigma}, \sigma_i), \pi_i^{\mu,\sigma} - \pi_i^{t+1} \rangle + \eta_t \mu \beta D_\psi(\pi_i^{t+1}, \pi_i^t) \\
&\quad - \eta_t \mu \gamma D_\psi(\pi_i^{\mu,\sigma}, \pi_i^t) + \langle \xi_i^t, \pi_i^{t+1} - \pi_i^{\mu,\sigma} \rangle.
\end{aligned}
$$

Summing up these inequalities with respect to the player index,

$$
\begin{aligned}
&D_\psi(\pi^{\mu,\sigma}, \pi^{t+1}) - D_\psi(\pi^{\mu,\sigma}, \pi^t) + D_\psi(\pi^{t+1}, \pi^t) \\
&\leq \eta_t \sum_{i=1}^{N} \langle \nabla_{\pi_i} v_i(\pi_i^t, \pi_{-i}^t) - \mu G(\pi_i^{\mu,\sigma}, \sigma_i), \pi_i^{t+1} - \pi_i^{\mu,\sigma} \rangle + \eta_t \mu \beta D_\psi(\pi^{t+1}, \pi^t) \\
&\quad - \eta_t \mu \gamma D_\psi(\pi^{\mu,\sigma}, \pi^t) + \sum_{i=1}^{N} \langle \xi_i^t, \pi_i^{t+1} - \pi_i^{\mu,\sigma} \rangle \\
&= \sum_{i=1}^{N} \eta_t \langle \nabla_{\pi_i} v_i(\pi_i^{t+1}, \pi_{-i}^{t+1}) - \mu \nabla_{\pi_i} G(\pi_i^{\mu,\sigma}, \sigma_i), \pi_i^{t+1} - \pi_i^{\mu,\sigma} \rangle - \eta_t \mu \gamma D_\psi(\pi^{\mu,\sigma}, \pi^t) + \eta_t \mu \beta D_\psi(\pi^{t+1}, \pi^t) \\
&\quad + \eta_t \sum_{i=1}^{N} \langle \nabla_{\pi_i} v_i(\pi_i^t, \pi_{-i}^t) - \nabla_{\pi_i} v_i(\pi_i^{t+1}, \pi_{-i}^{t+1}), \pi_i^{t+1} - \pi_i^{\mu,\sigma} \rangle + \eta_t \sum_{i=1}^{N} \langle \xi_i^t, \pi_i^{t+1} - \pi_i^{\mu,\sigma} \rangle \\
&\leq \sum_{i=1}^{N} \eta_t \langle \nabla_{\pi_i} v_i(\pi_i^{\mu,\sigma}, \pi_{-i}^{\mu,\sigma}) - \mu \nabla_{\pi_i} G(\pi_i^{\mu,\sigma}, \sigma_i), \pi_i^{t+1} - \pi_i^{\mu,\sigma} \rangle - \eta_t \mu \gamma D_\psi(\pi^{\mu,\sigma}, \pi^t) + \eta_t \mu \beta D_\psi(\pi^{t+1}, \pi^t) \\
&\quad + \eta_t \sum_{i=1}^{N} \langle \nabla_{\pi_i} v_i(\pi_i^t, \pi_{-i}^t) - \nabla_{\pi_i} v_i(\pi_i^{t+1}, \pi_{-i}^{t+1}), \pi_i^{t+1} - \pi_i^{\mu,\sigma} \rangle + \eta_t \sum_{i=1}^{N} \langle \xi_i^t, \pi_i^{t+1} - \pi_i^{\mu,\sigma} \rangle \\
&\leq -\eta_t \mu \gamma D_\psi(\pi^{\mu,\sigma}, \pi^t) + \eta_t \mu \beta D_\psi(\pi^{t+1}, \pi^t) \\
&\quad + \eta_t \sum_{i=1}^{N} \langle \nabla_{\pi_i} v_i(\pi_i^t, \pi_{-i}^t) - \nabla_{\pi_i} v_i(\pi_i^{t+1}, \pi_{-i}^{t+1}), \pi_i^{t+1} - \pi_i^{\mu,\sigma} \rangle + \eta_t \sum_{i=1}^{N} \langle \xi_i^t, \pi_i^{t+1} - \pi_i^{\mu,\sigma} \rangle,
\end{aligned}
\tag{28}
$$

where the second inequality follows (1), and the third inequality follows from the first-order optimality condition for $\pi^{\mu,\sigma}$.

By combining (26) in Appendix F.2 and (28), we have for any $\lambda > 0$:

$$
\begin{aligned}
&D_\psi(\pi^{\mu,\sigma}, \pi^{t+1}) - D_\psi(\pi^{\mu,\sigma}, \pi^t) + D_\psi(\pi^{t+1}, \pi^t) \\
&\leq -\eta_t \mu \gamma D_\psi(\pi^{\mu,\sigma}, \pi^t) + \eta_t \mu \beta D_\psi(\pi^{t+1}, \pi^t) \\
&\quad + \frac{\eta_t}{\rho} \left( 2L^2 \lambda + \frac{1}{\lambda} \right) D_\psi(\pi^{t+1}, \pi^t) + \frac{\eta_t}{\rho \lambda} D_\psi(\pi^{\mu,\sigma}, \pi^t) + \eta_t \sum_{i=1}^{N} \langle \xi_i^t, \pi_i^{t+1} - \pi_i^{\mu,\sigma} \rangle.
\end{aligned}
$$

By setting $\lambda = \frac{2}{\mu \gamma \rho}$,

$$
\begin{aligned}
&D_\psi(\pi^{\mu,\sigma}, \pi^{t+1}) - D_\psi(\pi^{\mu,\sigma}, \pi^t) + D_\psi(\pi^{t+1}, \pi^t) \\
&\leq -\frac{\eta_t \mu \gamma}{2} D_\psi(\pi^{\mu,\sigma}, \pi^t) + \eta_t \left( \frac{\mu^2 \gamma \rho^2 (\gamma + 2\beta) + 8L^2}{2\mu \gamma \rho^2} \right) D_\psi(\pi^{t+1}, \pi^t) + \eta_t \sum_{i=1}^{N} \langle \xi_i^t, \pi_i^{t+1} - \pi_i^{\mu,\sigma} \rangle.
\end{aligned}
$$

This concludes the proof. □

### F.4 Proof of Lemma C.2

*Proof of Lemma C.2.* Reforming the inequality in the assumption,

$$D_\psi(\pi^{\mu,\sigma}, \pi^{t+1})$$

$$\leq (1 - \kappa\eta_t)D_\psi(\pi^{\mu,\sigma}, \pi^t) - (1 - \eta_t\theta)D_\psi(\pi^{t+1}, \pi^t) + \eta_t \sum_{i=1}^N \langle \xi_i^t, \pi_i^{t+1} - \pi_i^{\mu,\sigma} \rangle$$

$$= (1 - \kappa\eta_t)D_\psi(\pi^{\mu,\sigma}, \pi^t) - (1 - \eta_t\theta)D_\psi(\pi^{t+1}, \pi^t)$$

$$+ \eta_t \sum_{i=1}^N \langle \xi_i^t, \pi_i^t - \pi_i^{\mu,\sigma} \rangle + \eta_t \sum_{i=1}^N \langle \xi_i^t, \pi_i^{t+1} - \pi_i^t \rangle.$$

By taking the expectation conditioned on $\mathcal{F}_t$ for both sides and using Assumption 4.8 (a),

$$\mathbb{E}[D_\psi(\pi^{\mu,\sigma}, \pi^{t+1})|\mathcal{F}_t]$$

$$\leq (1 - \kappa\eta_t)D_\psi(\pi^{\mu,\sigma}, \pi^t) - (1 - \eta_t\theta)\mathbb{E}[D_\psi(\pi^{t+1}, \pi^t)|\mathcal{F}_t] + \sum_{i=1}^N \mathbb{E}[\langle \eta_t\xi_i^t, \pi_i^{t+1} - \pi_i^t \rangle|\mathcal{F}_t]$$

$$= (1 - \kappa\eta_t)D_\psi(\pi^{\mu,\sigma}, \pi^t) - (1 - \eta_t\theta)\mathbb{E}[D_\psi(\pi^{t+1}, \pi^t)|\mathcal{F}_t]$$

$$+ \sum_{i=1}^N \mathbb{E}\left[ \left\langle \frac{\eta_t\xi_i^t}{\sqrt{\rho(1 - \eta_t\theta)}}, \sqrt{\rho(1 - \eta_t\theta)}(\pi_i^{t+1} - \pi_i^t) \right\rangle |\mathcal{F}_t \right]$$

$$\leq (1 - \kappa\eta_t)D_\psi(\pi^{\mu,\sigma}, \pi^t) - (1 - \eta_t\theta)\mathbb{E}[D_\psi(\pi^{t+1}, \pi^t)|\mathcal{F}_t]$$

$$+ \frac{\eta_t^2}{2\rho(1 - \eta_t\theta)} \sum_{i=1}^N \mathbb{E}[\|\xi_i^t\|^2|\mathcal{F}_t] + \frac{\rho(1 - \eta_t\theta)}{2}\mathbb{E}[\|\pi^{t+1} - \pi^t\|^2|\mathcal{F}_t]$$

$$\leq (1 - \kappa\eta_t)D_\psi(\pi^{\mu,\sigma}, \pi^t) + \frac{\eta_t^2}{2\rho(1 - \eta_t\theta)} \sum_{i=1}^N \mathbb{E}[\|\xi_i^t\|^2|\mathcal{F}_t]$$

$$\leq \left(1 - \frac{1}{t + 2\theta/\kappa}\right)D_\psi(\pi^{\mu,\sigma}, \pi^t) + \frac{\eta_t^2}{\rho} \sum_{i=1}^N \mathbb{E}[\|\xi_i^t\|^2|\mathcal{F}_t].$$

Therefore, rearranging and taking the expectations,

$$(t + 2\theta/\kappa)\mathbb{E}[D_\psi(\pi^{\mu,\sigma}, \pi^{t+1})] \leq (t - 1 + 2\theta/\kappa)\mathbb{E}[D_\psi(\pi^{\mu,\sigma}, \pi^t)] + \frac{NC^2}{\rho\kappa(\kappa t + 2\theta)}.$$

Telescoping the sum,

$$(t + 2\theta/\kappa)\mathbb{E}[D_\psi(\pi^{\mu,\sigma}, \pi^{t+1})] \leq (2\theta/\kappa - 1)D_\psi(\pi^{\mu,\sigma}, \pi^0) + \frac{NC^2}{\rho\kappa} \sum_{s=0}^t \frac{1}{\kappa s + 2\theta}$$

$$\iff \mathbb{E}[D_\psi(\pi^{\mu,\sigma}, \pi^{t+1})] \leq \frac{2\theta - \kappa}{\kappa t + 2\theta}D_\psi(\pi^{\mu,\sigma}, \pi^0) + \frac{NC^2}{\rho(\kappa t + 2\theta)} \sum_{s=0}^t \frac{1}{\kappa s + 2\theta}.$$

Here, we introduce the following lemma, whose proof is given in Appendix F.12, for the evaluation of the sum.

**Lemma F.1.** *For any $\kappa, \theta \geq 0$ and $t \geq 0$,*

$$\sum_{s=0}^t \frac{1}{\kappa s + 2\theta} \leq \frac{1}{2\theta} + \frac{1}{\kappa} \ln\left(\frac{\kappa}{2\theta}t + 1\right).$$

In summary, we obtain the following inequality:

$$\mathbb{E}[D_\psi(\pi^{\mu,\sigma}, \pi^{t+1})] \leq \frac{2\theta - \kappa}{\kappa t + 2\theta}D_\psi(\pi^{\mu,\sigma}, \pi^0) + \frac{NC^2}{\rho(\kappa t + 2\theta)}\left(\frac{1}{\kappa} \ln\left(\frac{\kappa}{2\theta}t + 1\right) + \frac{1}{2\theta}\right).$$

This concludes the proof. □

### F.5 PROOF OF LEMMA E.1

*Proof of Lemma E.1.*

$$
\begin{aligned}
\text{exploit}(\pi) &= \sum_{i=1}^{N} \left( \max_{\tilde{\pi}_i \in \mathcal{X}_i} v_i(\tilde{\pi}_i, \pi_{-i}) - v_i(\pi) \right) \\
&\leq \sum_{i=1}^{N} \max_{\tilde{\pi}_i \in \mathcal{X}_i} \langle \nabla_{\pi_i} v_i(\pi), \tilde{\pi}_i - \pi_i \rangle \\
&= \max_{\tilde{\pi} \in \mathcal{X}} \sum_{i=1}^{N} \left( \langle \nabla_{\pi_i} v_i(\pi'), \tilde{\pi}_i - \pi_i' \rangle - \langle \nabla_{\pi_i} v_i(\pi'), \pi_i - \pi_i' \rangle + \langle \nabla_{\pi_i} v_i(\pi) - \nabla_{\pi_i} v_i(\pi'), \tilde{\pi}_i - \pi_i \rangle \right).
\end{aligned}
\tag{29}
$$

From Lemma B.3, the first term of (29) can be upper bounded as:

$$
\max_{\tilde{\pi} \in \mathcal{X}} \sum_{i=1}^{N} \langle \nabla_{\pi_i} v_i(\pi'), \tilde{\pi}_i - \pi_i' \rangle \leq \text{diam}(\mathcal{X}) \cdot \min_{(a_i) \in N_{\mathcal{X}}(\pi')} \sqrt{\sum_{i=1}^{N} \| - \nabla_{\pi_i} v_i(\pi') + a_i \|^2}
\tag{30}
$$

Next, from Cauchy-Schwarz inequality, the second term of (29) can be upper bounded as:

$$
\begin{aligned}
-\sum_{i=1}^{N} \langle \nabla_{\pi_i} v_i(\pi'), \pi_i - \pi_i' \rangle &\leq \| \pi - \pi' \| \sqrt{\sum_{i=1}^{N} \| \nabla_{\pi_i} v_i(\pi') \|^2} \\
&\leq \zeta \| \pi - \pi' \|.
\end{aligned}
\tag{31}
$$

Again from Cauchy-Schwarz inequality, the third term of (29) can be upper bounded as:

$$
\begin{aligned}
\sum_{i=1}^{N} \langle \nabla_{\pi_i} v_i(\pi) - \nabla_{\pi_i} v_i(\pi'), \tilde{\pi}_i - \pi_i \rangle &\leq \| \tilde{\pi} - \pi \| \sqrt{\sum_{i=1}^{N} \| \nabla_{\pi_i} v_i(\pi) - \nabla_{\pi_i} v_i(\pi') \|^2} \\
&\leq \text{diam}(\mathcal{X}) \sqrt{\sum_{i=1}^{N} \| \nabla_{\pi_i} v_i(\pi) - \nabla_{\pi_i} v_i(\pi') \|^2} \\
&\leq L \cdot \text{diam}(\mathcal{X}) \cdot \| \pi - \pi' \|.
\end{aligned}
\tag{32}
$$

By combining (29), (30), (31), and (32), we get:

$$
\text{exploit}(\pi) \leq \text{diam}(\mathcal{X}) \cdot \min_{(a_i) \in N_{\mathcal{X}}(\pi')} \sqrt{\sum_{i=1}^{N} \| - \nabla_{\pi_i} v_i(\pi') + a_i \|^2} + (L \cdot \text{diam}(\mathcal{X}) + \zeta) \| \pi - \pi' \|.
$$

$\square$

### F.6 PROOF OF LEMMA E.2

*Proof of Lemma E.2.* First, we prove the following lemma:

**Lemma F.2.** *Assume that* $\sqrt{\sum_{i=1}^{N} \| \nabla_{\pi_i} v_i(\pi) \|^2} \leq \zeta$ *for any* $\pi \in \mathcal{X}$. *Then, we have for any* $k \in [K]$:

$$
\| \pi^{\mu, \sigma^k} - \sigma^k \|^2 \leq \| \sigma^k - \sigma^{k-1} \|^2 + \frac{2\zeta}{\mu} \| \pi^{\mu, \sigma^{k-1}} - \sigma^k \|.
$$

From Lemma F.2, we can bound $\|\pi^{\mu,\sigma^k} - \sigma^k\|^2$ as:

$$\|\pi^{\mu,\sigma^k} - \sigma^k\|^2$$

$$\leq \|\sigma^k - \sigma^{k-1}\|^2 + \frac{2\zeta}{\mu}\|\pi^{\mu,\sigma^{k-1}} - \sigma^k\|$$

$$\leq \|\pi^{\mu,\sigma^{k-1}} - \sigma^{k-1}\|^2 + \|\sigma^k - \pi^{\mu,\sigma^{k-1}}\|^2 + 2\|\sigma^k - \pi^{\mu,\sigma^{k-1}}\|\|\pi^{\mu,\sigma^{k-1}} - \sigma^{k-1}\| + \frac{2\zeta}{\mu}\|\pi^{\mu,\sigma^{k-1}} - \sigma^k\|. \tag{33}$$

Next, we upper bound $\|\pi^{\mu,\sigma^k} - \sigma^k\|^2$ using the following lemma:

**Lemma F.3.** *Assume that $\|\pi^{\mu,\sigma^k} - \sigma^{k+1}\| \leq \|\pi^{\mu,\sigma^k} - \sigma^k\|\left(\frac{1}{c}\right)^{T_\sigma}$ for some $c > 1$. Then, if we set $T_\sigma = \max(\frac{3}{\ln c}\ln K + \frac{\ln 64}{\ln c}, 1)$, we have for any Nash equilibrium $\pi^* \in \Pi^*$:*

$$\sum_{k=0}^{K-1} \|\pi^{\mu,\sigma^k} - \sigma^k\|^2 \leq 16\|\pi^* - \sigma^0\|^2.$$

Using Lemma F.3, we have:

$$\|\pi^{\mu,\sigma^k} - \sigma^k\|^2 \leq \sum_{k=0}^{K-1} \|\pi^{\mu,\sigma^k} - \sigma^k\|^2 \leq 16\|\pi^* - \sigma^0\|^2,$$

and then from the assumption, we get:

$$\|\pi^{\mu,\sigma^k} - \sigma^{k+1}\| \leq \|\pi^{\mu,\sigma^k} - \sigma^k\|\left(\frac{1}{c}\right)^{T_\sigma} \leq \frac{4\|\pi^* - \sigma^0\|}{c^{T_\sigma}}. \tag{34}$$

By combining (33) and (34), we have:

$$\|\pi^{\mu,\sigma^k} - \sigma^k\|^2 \leq \|\pi^{\mu,\sigma^{k-1}} - \sigma^{k-1}\|^2 + \frac{16\|\pi^* - \sigma^0\|^2}{c^{2T_\sigma}} + 2\frac{4\|\pi^* - \sigma^0\|}{c^{T_\sigma}}4\|\pi^* - \sigma^0\| + \frac{2\zeta}{\mu}\frac{4\|\pi^* - \sigma^0\|}{c^{T_\sigma}}$$

$$\leq \|\pi^{\mu,\sigma^{k-1}} - \sigma^{k-1}\|^2 + \frac{48\|\pi^* - \sigma^0\|^2}{c^{T_\sigma}} + \frac{8\zeta}{\mu}\frac{\|\pi^* - \sigma^0\|}{c^{T_\sigma}}$$

$$\leq \|\pi^{\mu,\sigma^{k-1}} - \sigma^{k-1}\|^2 + \frac{8\|\pi^* - \sigma^0\|}{c^{T_\sigma}}\left(6\|\pi^* - \sigma^0\| + \frac{\zeta}{\mu}\right)$$

Therefore, we get:

$$\|\pi^{\mu,\sigma^{K-1}} - \sigma^{K-1}\|^2 \leq \|\pi^{\mu,\sigma^{K-2}} - \sigma^{K-2}\|^2 + \frac{8\|\pi^* - \sigma^0\|}{c^{T_\sigma}}\left(6\|\pi^* - \sigma^0\| + \frac{\zeta}{\mu}\right)$$

$$\leq \|\pi^{\mu,\sigma^k} - \sigma^k\|^2 + \frac{8K\|\pi^* - \sigma^0\|}{c^{T_\sigma}}\left(6\|\pi^* - \sigma^0\| + \frac{\zeta}{\mu}\right). \tag{35}$$

By combining (35) and Lemma F.3, we have:

$$K\|\pi^{\mu,\sigma^{K-1}} - \sigma^{K-1}\|^2 \leq \sum_{k=0}^{K-1}\|\pi^{\mu,\sigma^k} - \sigma^k\|^2 + \frac{8K^2\|\pi^* - \sigma^0\|}{c^{T_\sigma}}\left(6\|\pi^* - \sigma^0\| + \frac{\zeta}{\mu}\right)$$

$$\leq 16\|\pi^* - \sigma^0\|^2 + \frac{8K^2\|\pi^* - \sigma^0\|}{c^{T_\sigma}}\left(6\|\pi^* - \sigma^0\| + \frac{\zeta}{\mu}\right)$$

$$\leq 16\|\pi^* - \sigma^0\|^2 + 8\|\pi^* - \sigma^0\|\left(6\|\pi^* - \sigma^0\| + \frac{\zeta}{\mu}\right)$$

$$= 8\|\pi^* - \sigma^0\|\left(8\|\pi^* - \sigma^0\| + \frac{\zeta}{\mu}\right).$$

$$\square$$

### F.7 PROOF OF LEMMA E.3

*Proof of Lemma E.3.* From Lemma F.2 and the assumption of $\mathbb{E}[\|\pi^{\mu,\sigma^k} - \sigma^{k+1}\| \mid \sigma^k] \leq c\sqrt{\frac{\ln T_\sigma}{T_\sigma}} \leq 2c\frac{\sqrt{\ln K}}{K^2}$, we have:

$$\mathbb{E}[\|\pi^{\mu,\sigma^k} - \sigma^k\|^2 \mid \sigma^k]$$

$$\leq \|\sigma^k - \sigma^{k-1}\|^2 + \mathbb{E}\left[\frac{2\zeta}{\mu}\|\pi^{\mu,\sigma^{k-1}} - \sigma^k\| \mid \sigma^k\right]$$

$$\leq \|\sigma^k - \pi^{\mu,\sigma^{k-1}}\|^2 + \|\pi^{\mu,\sigma^{k-1}} - \sigma^{k-1}\|^2 + \mathbb{E}\left[2\|\sigma^k - \pi^{\mu,\sigma^{k-1}}\|\|\pi^{\mu,\sigma^{k-1}} - \sigma^{k-1}\| + \frac{2\zeta}{\mu}\|\pi^{\mu,\sigma^{k-1}} - \sigma^k\| \mid \sigma^k\right]$$

$$\leq \|\pi^{\mu,\sigma^{k-1}} - \sigma^{k-1}\|^2 + c^2\frac{\ln T_\sigma}{T_\sigma} + 2\mathrm{diam}(\mathcal{X}) \cdot c\sqrt{\frac{\ln T_\sigma}{T_\sigma}} + \frac{2\zeta c}{\mu}\sqrt{\frac{\ln T_\sigma}{T_\sigma}}$$

$$\leq \|\pi^{\mu,\sigma^{k-1}} - \sigma^{k-1}\|^2 + \frac{4c^2\ln K}{K^4} + 4c\frac{\sqrt{\ln K}}{K^2}\left(\mathrm{diam}(\mathcal{X}) + \frac{\zeta}{\mu}\right).$$

Therefore, we get:

$$\mathbb{E}[\|\pi^{\mu,\sigma^{K-1}} - \sigma^{K-1}\|^2] \leq \mathbb{E}[\|\pi^{\mu,\sigma^{K-2}} - \sigma^{K-2}\|^2] + \frac{4c^2\ln K}{K^4} + 4c\frac{\sqrt{\ln K}}{K^2}\left(\mathrm{diam}(\mathcal{X}) + \frac{\zeta}{\mu}\right)$$

$$\leq \mathbb{E}[\|\pi^{\mu,\sigma^k} - \sigma^k\|^2] + \frac{4c^2\ln K}{K^3} + 4c\frac{\sqrt{\ln K}}{K}\left(\mathrm{diam}(\mathcal{X}) + \frac{\zeta}{\mu}\right). \quad (36)$$

Here, we derive the following upper bound in terms of $\mathbb{E}[\|\pi^{\mu,\sigma^k} - \sigma^k\|^2]$:

**Lemma F.4.** *Assume that* $\mathbb{E}[\|\pi^{\mu,\sigma^k} - \sigma^{k+1}\|^2 \mid \sigma^k] \leq c^2\frac{\ln T_\sigma}{T_\sigma}$ *for any* $k \geq 0$ *for some* $c > 0$. *Then, if we set* $T_\sigma \geq \max(K^4, 3)$, *we have for any Nash equilibrium* $\pi^* \in \Pi^*$:

$$\mathbb{E}\left[\sum_{k=0}^{K-1}\|\pi^{\mu,\sigma^k} - \sigma^k\|^2\right] \leq \|\pi^* - \sigma^0\|^2 + \mathrm{diam}(\mathcal{X}) \cdot \frac{4c\sqrt{\ln K}}{K^2}.$$

By combining (36) and Lemma F.4, we have:

$$K\mathbb{E}[\|\pi^{\mu,\sigma^{K-1}} - \sigma^{K-1}\|^2]$$

$$\leq \mathbb{E}\left[\sum_{k=0}^{K-1}\|\pi^{\mu,\sigma^k} - \sigma^k\|^2\right] + \frac{4c^2\ln K}{K^2} + 4c\sqrt{\ln K}\left(\mathrm{diam}(\mathcal{X}) + \frac{\zeta}{\mu}\right)$$

$$\leq \|\pi^* - \sigma^0\|^2 + \mathrm{diam}(\mathcal{X}) \cdot \frac{4c\sqrt{\ln K}}{K^2} + \frac{4c^2\ln K}{K^2} + 4c\sqrt{\ln K}\left(\mathrm{diam}(\mathcal{X}) + \frac{\zeta}{\mu}\right)$$

$$\leq \|\pi^* - \sigma^0\|^2 + 4c\left(2\mathrm{diam}(\mathcal{X}) + \frac{\zeta}{\mu} + c\right)\ln K.$$

$$\square$$

### F.8 PROOF OF LEMMA E.4

*Proof of Lemma E.4.* Recall that $G(\pi_i, \pi_i') = D_{\psi'}(\pi_i, \pi_i')$ for any $i \in [N]$ and $\pi_i, \pi_i' \in \mathcal{X}_i$. By the first-order optimality condition for $\sigma^{k+1}$, we have for all $\pi^* \in \Pi^*$:

$$\sum_{i=1}^{N}\langle\nabla_{\pi_i}v_i(\sigma_i^{k+1}, \sigma_{-i}^{k+1}) - \mu(\nabla\psi'(\sigma_i^{k+1}) - \nabla\psi'(\sigma_i^k)), \pi_i^* - \sigma_i^{k+1}\rangle \leq 0.$$

Then,

$$\sum_{i=1}^{N}\langle\nabla\psi'(\sigma_i^{k+1}) - \nabla\psi'(\sigma_i^k), \sigma_i^{k+1} - \pi_i^*\rangle \leq \frac{1}{\mu}\sum_{i=1}^{N}\langle\sigma_i^{k+1} - \pi_i^*, \nabla_{\pi_i}v_i(\sigma_i^{k+1}, \sigma_{-i}^{k+1})\rangle.$$

Moreover, we have for any $\pi^* \in \Pi^*$:

$$D_{\psi'}(\pi^*, \sigma^{k+1}) - D_{\psi'}(\pi^*, \sigma^k)$$

$$= \sum_{i=1}^{N} \left( \psi'(\pi_i^*) - \psi'(\sigma_i^{k+1}) - \langle \nabla \psi'(\sigma_i^{k+1}), \pi_i^* - \sigma_i^{k+1} \rangle - \psi'(\pi_i^*) + \psi'(\sigma_i^k) + \langle \nabla \psi'(\sigma_i^k), \pi_i^* - \sigma_i^k \rangle \right)$$

$$= \sum_{i=1}^{N} \left( -\psi'(\sigma_i^{k+1}) + \psi'(\sigma_i^k) + \langle \nabla \psi'(\sigma_i^k), \sigma_i^{k+1} - \sigma_i^k \rangle - \langle \nabla \psi'(\sigma_i^{k+1}) - \nabla \psi'(\sigma_i^k), \pi_i^* - \sigma_i^{k+1} \rangle \right)$$

$$= -D_{\psi'}(\sigma^{k+1}, \sigma^k) + \sum_{i=1}^{N} \langle \nabla \psi'(\sigma_i^{k+1}) - \nabla \psi'(\sigma_i^k), \sigma_i^{k+1} - \pi_i^* \rangle.$$

By combining these inequalities, we get for any $\pi^* \in \Pi^*$:

$$D_{\psi'}(\pi^*, \sigma^{k+1}) - D_{\psi'}(\pi^*, \sigma^k) \le -D_{\psi'}(\sigma^{k+1}, \sigma^k) + \frac{1}{\mu} \sum_{i=1}^{N} \langle \sigma_i^{k+1} - \pi_i^*, \nabla_{\pi_i} v_i(\sigma_i^{k+1}, \sigma_{-i}^{k+1}) \rangle$$

$$\le -D_{\psi'}(\sigma^{k+1}, \sigma^k) + \frac{1}{\mu} \sum_{i=1}^{N} \langle \sigma_i^{k+1} - \pi_i^*, \nabla_{\pi_i} v_i(\pi_i^*, \pi_{-i}^*) \rangle,$$

where the second inequality follows from (1). Since $\pi^*$ is the Nash equilibrium, from the first-order optimality condition, we get:

$$\sum_{i=1}^{N} \langle \sigma_i^{k+1} - \pi_i^*, \nabla_{\pi_i} v_i(\pi_i^*, \pi_{-i}^*) \rangle \le 0.$$

Thus, we have for $\pi^* \in \Pi^*$:

$$D_{\psi'}(\pi^*, \sigma^{k+1}) - D_{\psi'}(\pi^*, \sigma^k) \le -D_{\psi'}(\sigma^{k+1}, \sigma^k).$$

$\square$

### F.9 PROOF OF LEMMA E.5

*Proof of Lemma E.5.* By using the first-order optimality condition for $\sigma_i^{k+1}$, we have for all $\pi \in \mathcal{X}$:

$$\sum_{i=1}^{N} \langle \nabla_{\pi_i} v_i(\sigma_i^{k+1}, \sigma_{-i}^{k+1}) - \mu(\nabla_{\pi_i} \psi'(\sigma_i^{k+1}) - \nabla_{\pi_i} \psi'(\sigma_i^k)), \pi_i - \sigma_i^{k+1} \rangle \le 0,$$

and then

$$\sum_{i=1}^{N} \langle \nabla_{\pi_i} v_i(\sigma_i^{k+1}, \sigma_{-i}^{k+1}), \pi_i - \sigma_i^{k+1} \rangle \le \mu \sum_{i=1}^{N} \langle \nabla_{\pi_i} \psi'(\sigma_i^{k+1}) - \nabla_{\pi_i} \psi'(\sigma_i^k), \pi_i - \sigma_i^{k+1} \rangle.$$

Under the assumption that $\sigma^{k+1} = \sigma^k$, we have for all $\pi \in \mathcal{X}$:

$$\sum_{i=1}^{N} \langle \nabla_{\pi_i} v_i(\sigma_i^{k+1}, \sigma_{-i}^{k+1}), \pi_i - \sigma_i^{k+1} \rangle \le 0.$$

This is equivalent to the first-order optimality condition for $\pi^* \in \Pi^*$. Therefore, $\sigma^{k+1} = \sigma^k$ is a Nash equilibrium of the original game. $\square$

### F.10 PROOF OF LEMMA E.6

*Proof of Lemma E.6.* First, we prove the first statement of the lemma by using the following lemmas:

**Lemma F.5.** *Assume that $\sigma^{k+1} = \pi^{\mu, \sigma^k}$ for $k \ge 0$, and $G$ is one of the following divergence: 1) $\alpha$-divergence with $\alpha \in (0, 1)$; 2) Rényi-divergence with $\alpha \in (0, 1)$; 3) reverse KL divergence. If $\sigma^{k+1} = \sigma^k$, then $\sigma^k$ is a Nash equilibrium of the original game.*

**Lemma F.6.** *Assume that $\sigma^{k+1} = \pi^{\mu,\sigma^k}$ for $k \geq 0$, and $G$ is one of the following divergence: 1) $\alpha$-divergence with $\alpha \in (0,1)$; 2) Rényi-divergence with $\alpha \in (0,1)$; 3) reverse KL divergence. Then, if $\sigma^{k+1} \neq \sigma^k$, we have for any $\pi^* \in \Pi^*$ and $k \geq 0$:*

$$\mathrm{KL}(\pi^*, \sigma^{k+1}) - \mathrm{KL}(\pi^*, \sigma^k) < 0.$$

From Lemma F.5, when $\sigma^k \in \mathcal{X} \setminus \Pi^*$, $\sigma^{k+1} \neq \sigma^k$ always holds. Let us define $\pi^\star = \arg\min_{\pi^* \in \Pi^*} \mathrm{KL}(\pi^*, r)$. Since $\sigma^{k+1} \neq \sigma^k$, from Lemma F.6, we have:

$$\min_{\pi^* \in \Pi^*} \mathrm{KL}(\pi^*, \sigma^k) = \mathrm{KL}(\pi^\star, \sigma^k) > \mathrm{KL}(\pi^\star, \sigma^{k+1}) \geq \min_{\pi^* \in \Pi^*} \mathrm{KL}(\pi^*, \sigma^{k+1}).$$

Therefore, if $\sigma^k \in \mathcal{X} \setminus \Pi^*$ then $\min_{\pi^* \in \Pi^*} \mathrm{KL}(\pi^*, \sigma^{k+1}) < \min_{\pi^* \in \Pi^*} \mathrm{KL}(\pi^*, \sigma^k)$.

Next, we prove the second statement of the lemma. Assume that there exists $\sigma^k \in \Pi^*$ such that $\sigma^{k+1} \neq \sigma^k$. In this case, we can apply Lemma F.6, hence we have $\mathrm{KL}(\pi^*, \sigma^{k+1}) < \mathrm{KL}(\pi^*, \sigma^k)$ for all $\pi^* \in \Pi^*$. On the other hand, since $\sigma^k \in \Pi^*$, there exists a Nash equilibrium $\pi^\star \in \Pi^*$ such that $\mathrm{KL}(\pi^\star, \sigma^k) = 0$. Therefore, we have $\mathrm{KL}(\pi^\star, \sigma^{k+1}) < \mathrm{KL}(\pi^\star, \sigma^k) = 0$, which contradicts $\mathrm{KL}(\pi^\star, \sigma^{k+1}) \geq 0$. Thus, if $\sigma^k \in \Pi^*$ then $\sigma^{k+1} = \sigma^k$. $\qquad\square$

### F.11 PROOF OF LEMMA E.7

*Proof of Lemma E.7.* For a given $\sigma \in \mathcal{X}$, let us consider that $\pi^t$ follows the following continuous-time dynamics:

$$\pi_i^t = \arg\max_{\pi_i \in \mathcal{X}_i} \left\{ \langle y_i^t, \pi_i \rangle - \psi(\pi_i) \right\}, \tag{37}$$

$$y_{ij}^t = \int_0^t \left( \frac{\partial}{\pi_{ij}} v_i(\pi_i^s, \pi_{-i}^s) - \mu \frac{\partial}{\pi_{ij}} G(\pi_i^s, \sigma_i) \right).$$

We assume that $\psi(\pi_i) = \sum_{j=1}^{d_i} \pi_{ij} \ln \pi_{ij}$. Note that this dynamics is the continuous-time version of FTRL-SP (3), so clearly $\pi^{\mu,\sigma}$ defined by (4) is the stationary point of (37). We have for a given $\sigma' \in \mathcal{X}$ and the associated stationary point $\pi^{\mu,\sigma'} = F(\sigma')$:

$$\begin{aligned}
\frac{d}{dt}\mathrm{KL}(\pi^{\mu,\sigma'}, \pi^t) &= \frac{d}{dt} D_\psi(\pi^{\mu,\sigma'}, \pi^t) \\
&= \sum_{i=1}^N \frac{d}{dt} \left( \psi(\pi_i^{\mu,\sigma'}) - \psi(\pi_i^t) - \langle y_i^t, \pi_i^{\mu,\sigma'} - \pi_i^t \rangle \right) \\
&= \sum_{i=1}^N \frac{d}{dt} \left( \langle y_i^t, \pi_i^t \rangle - \psi(\pi_i^t) - \langle y_i^t, \pi_i^{\mu,\sigma'} \rangle + \psi(\pi_i^{\mu,\sigma'}) \right) \\
&= \sum_{i=1}^N \frac{d}{dt} \left( \psi^*(y_i^t) - \langle y_i^t, \pi_i^{\mu,\sigma'} \rangle \right) \\
&= \sum_{i=1}^N \left( \left\langle \frac{d}{dt} y_i^t, \nabla \psi^*(y_i^t) \right\rangle - \left\langle \frac{d}{dt} y_i^t, \pi_i^{\mu,\sigma'} \right\rangle \right) \\
&= \sum_{i=1}^N \left\langle \frac{d}{dt} y_i^t, \nabla \psi^*(y_i^t) - \pi_i^{\mu,\sigma'} \right\rangle,
\end{aligned}$$

where $\psi^*(y_i^t) = \max_{\pi_i \in \mathcal{X}_i} \left\{ \langle y_i^t, \pi_i \rangle - \psi(\pi_i) \right\}$. When $\psi(\pi_i) = \sum_{j=1}^{d_i} \pi_{ij} \ln \pi_{ij}$, we have

$$\begin{aligned}
\psi^*(y_i^t) &= \sum_{j=1}^{d_i} y_{ij}^t \frac{\exp(y_{ij}^t)}{\sum_{j'=1}^{d_i} \exp(y_{ij'}^t)} - \sum_{j=1}^{d_i} \frac{\exp(y_{ij}^t)}{\sum_{j'=1}^{d_i} \exp(y_{ij'}^t)} \ln \frac{\exp(y_{ij}^t)}{\sum_{j'=1}^{d_i} \exp(y_{ij'}^t)} \\
&= \frac{\ln \sum_{j'=1}^{d_i} \exp(y_{ij'}^t)}{\sum_{j'=1}^{d_i} \exp(y_{ij'}^t)} \sum_{j=1}^{d_i} \exp(y_{ij}^t),
\end{aligned}$$

and then,

$$
\begin{aligned}
\frac{\partial}{\partial y_{ij}} \psi^*(y_i^t) &= \frac{\exp(y_{ij}^t)}{(\sum_{j'=1}^{d_i} \exp(y_{ij'}^t))^2} \sum_{j'=1}^{d_i} \exp(y_{ij'}^t) - \frac{\ln \sum_{j'=1}^{d_i} \exp(y_{ij'}^t)}{(\sum_{j'=1}^{d_i} \exp(y_{ij'}^t))^2} \exp(y_{ij}^t) \sum_{j'=1}^{d_i} \exp(y_{ij'}^t) \\
&\quad + \frac{\ln \sum_{j'=1}^{d_i} \exp(y_{ij'}^t)}{\sum_{j'=1}^{d_i} \exp(y_{ij'}^t)} \exp(y_{ij}^t) \\
&= \frac{\exp(y_{ij}^t)}{\sum_{j'=1}^{d_i} \exp(y_{ij'}^t)} = \pi_{ij}^t.
\end{aligned}
$$

Therefore, we get $\nabla \psi^*(y_i^t) = \pi_i^t$. Hence,

$$
\begin{aligned}
\frac{d}{dt} \mathrm{KL}(\pi^{\mu,\sigma'}, \pi^t) &= \sum_{i=1}^N \left\langle \frac{d}{dt} y_i^t, \pi_i^t - \pi_i^{\mu,\sigma'} \right\rangle \\
&= \sum_{i=1}^N \langle \nabla_{\pi_i} v_i(\pi_i^t, \pi_{-i}^t) - \mu \nabla_{\pi_i} G(\pi_i^t, \sigma_i), \pi_i^t - \pi_i^{\mu,\sigma'} \rangle \\
&= \sum_{i=1}^N \langle \nabla_{\pi_i} v_i(\pi_i^t, \pi_{-i}^t) - \mu \nabla_{\pi_i} G(\pi_i^t, \sigma_i'), \pi_i^t - \pi_i^{\mu,\sigma'} \rangle \\
&\quad + \mu \sum_{i=1}^N \langle \nabla_{\pi_i} G(\pi_i^t, \sigma_i') - \nabla_{\pi_i} G(\pi_i^t, \sigma_i), \pi_i^t - \pi_i^{\mu,\sigma'} \rangle.
\end{aligned}
\tag{38}
$$

The first term of (38) can be written as:

$$
\begin{aligned}
&\sum_{i=1}^N \langle \nabla_{\pi_i} v_i(\pi_i^t, \pi_{-i}^t) - \mu \nabla_{\pi_i} G(\pi_i^t, \sigma_i'), \pi_i^t - \pi_i^{\mu,\sigma'} \rangle \\
&\leq \sum_{i=1}^N \langle \nabla_{\pi_i} v_i(\pi_i^{\mu,\sigma'}, \pi_{-i}^{\mu,\sigma'}) - \mu \nabla_{\pi_i} G(\pi_i^t, \sigma_i'), \pi_i^t - \pi_i^{\mu,\sigma'} \rangle \\
&= \sum_{i=1}^N \langle \nabla_{\pi_i} v_i(\pi_i^{\mu,\sigma'}, \pi_{-i}^{\mu,\sigma'}), \pi_i^t - \pi_i^{\mu,\sigma'} \rangle - \mu \sum_{i=1}^N \langle \nabla_{\pi_i} G(\pi_i^t, \sigma_i'), \pi_i^t - \pi_i^{\mu,\sigma'} \rangle \\
&= \sum_{i=1}^N \langle \nabla_{\pi_i} v_i(\pi_i^{\mu,\sigma'}, \pi_{-i}^{\mu,\sigma'}) - \mu \nabla_{\pi_i} G(\pi_i^{\mu,\sigma'}, \sigma_i'), \pi_i^t - \pi_i^{\mu,\sigma'} \rangle \\
&\quad - \mu \sum_{i=1}^N \langle \nabla_{\pi_i} G(\pi_i^t, \sigma_i') - \nabla_{\pi_i} G(\pi_i^{\mu,\sigma'}, \sigma_i'), \pi_i^t - \pi_i^{\mu,\sigma'} \rangle \\
&\leq -\mu \sum_{i=1}^N \langle \nabla_{\pi_i} G(\pi_i^t, \sigma_i') - \nabla_{\pi_i} G(\pi_i^{\mu,\sigma'}, \sigma_i'), \pi_i^t - \pi_i^{\mu,\sigma'} \rangle.
\end{aligned}
$$

where the first inequality follows from (1), and the second inequality follows from the first-order optimality condition for $\pi^{\mu,\sigma}$. When $G$ is $\alpha$-divergence, $G$ has a diagonal Hessian is given as:

$$
\nabla^2 G(\pi_i, \sigma_i') = \begin{bmatrix} \frac{\sigma_{i1}'}{(\pi_{i1})^{\alpha-2}} & & \\ & \ddots & \\ & & \frac{\sigma_{id_i}'}{(\pi_{id_i})^{\alpha-2}}, \end{bmatrix}
$$

and thus, its smallest eigenvalue is lower bounded by $\min_{j \in [d_i]} \sigma'_{ij}$. Therefore,

$$
\sum_{i=1}^{N} \langle \nabla_{\pi_i} v_i(\pi_i^t, \pi_{-i}^t) - \mu \nabla_{\pi_i} G(\pi_i^t, \sigma'_i), \pi_i^t - \pi_i^{\mu, \sigma'} \rangle
$$

$$
\leq -\mu \sum_{i=1}^{N} \langle \nabla_{\pi_i} G(\pi_i^t, \sigma'_i) - \nabla_{\pi_i} G(\pi_i^{\mu, \sigma'}, \sigma'_i), \pi_i^t - \pi_i^{\mu, \sigma'} \rangle
$$

$$
\leq -\mu \left( \min_{i \in [N], \, j \in [d_i]} \sigma'_{ij} \right) \| \pi^t - \pi^{\mu, \sigma'} \|^2. \tag{39}
$$

On the other hand, by compactness of $\mathcal{X}_i$, the second term of (38) is written as:

$$
\mu \sum_{i=1}^{N} \langle \nabla_{\pi_i} G(\pi_i^t, \sigma'_i) - \nabla_{\pi_i} G(\pi_i^t, \sigma_i), \pi_i^t - \pi_i^{\mu, \sigma'} \rangle
$$

$$
\leq \mu \cdot \mathrm{diam}(\mathcal{X}) \sqrt{ \sum_{i=1}^{N} \| \nabla_{\pi_i} G(\pi_i^t, \sigma'_i) - \nabla_{\pi_i} G(\pi_i^t, \sigma_i) \|^2 } \tag{40}
$$

By combining (38), (39), and (40), we get:

$$
\frac{d}{dt} \mathrm{KL}(\pi^{\mu, \sigma'}, \pi^t)
$$

$$
\leq -\mu \left( \min_{i \in [N], \, j \in [d_i]} \sigma'_{ij} \right) \| \pi^t - \pi^{\mu, \sigma'} \|^2 + \mu \cdot \mathrm{diam}(\mathcal{X}) \sqrt{ \sum_{i=1}^{N} \| \nabla_{\pi_i} G(\pi_i^t, \sigma'_i) - \nabla_{\pi_i} G(\pi_i^t, \sigma_i) \|^2 }.
$$

Recall that $\pi^{\mu, \sigma}$ is the stationary point of (37). Therefore, by setting the start point as $\pi^0 = \pi^{\mu, \sigma}$, we have for all $t \geq 0, \pi^t = \pi^{\mu, \sigma}$. In this case, for all $t \geq 0$, $\frac{d}{dt} \mathrm{KL}(\pi^{\mu, \sigma'}, \pi^t) = 0$ and then:

$$
\left( \min_{i \in [N], \, j \in [d_i]} \sigma'_{ij} \right) \| \pi^{\mu, \sigma'} - \pi^{\mu, \sigma} \|^2 \leq \mathrm{diam}(\mathcal{X}) \sqrt{ \sum_{i=1}^{N} \| \nabla_{\pi_i} G(\pi_i^t, \sigma'_i) - \nabla_{\pi_i} G(\pi_i^t, \sigma_i) \|^2 }.
$$

For a given $\varepsilon > 0$, let us define $\varepsilon' = \frac{\left( \min_{i \in [N], \, j \in [d_i]} \sigma'_{ij} \right)}{\mathrm{diam}(\mathcal{X})} \varepsilon^2$. Since $\nabla_{\pi_i} G(\pi_i^{\mu, \sigma}, \cdot)$ is continuous on $\mathcal{X}_i$, for $\varepsilon'$, there exists $\delta > 0$ such that $\| \sigma' - \sigma \| < \delta \Rightarrow$ $\sqrt{ \sum_{i=1}^{N} \| \nabla_{\pi_i} G(\pi_i^{\mu, \sigma}, \sigma'_i) - \nabla_{\pi_i} G(\pi_i^{\mu, \sigma}, \sigma_i) \|^2 } < \varepsilon' = \frac{\left( \min_{i \in [N], \, j \in [d_i]} \sigma'_{ij} \right)}{\mathrm{diam}(\mathcal{X})} \varepsilon^2$. Thus, for every $\varepsilon > 0$, there exists $\delta > 0$ such that

$$
\| \sigma' - \sigma \| < \delta
$$

$$
\Rightarrow \| \pi^{\mu, \sigma'} - \pi^{\mu, \sigma} \| \leq \sqrt{ \frac{\mathrm{diam}(\mathcal{X})}{\left( \min_{i \in [N], \, j \in [d_i]} \sigma'_{ij} \right)} } \left( \sum_{i=1}^{N} \| \nabla_{\pi_i} G(\pi_i^{\mu, \sigma}, \sigma'_i) - \nabla_{\pi_i} G(\pi_i^{\mu, \sigma}, \sigma_i) \|^2 \right)^{\frac{1}{4}} < \varepsilon.
$$

This implies that $F(\cdot)$ is a continuous function on $\mathcal{X}$ when $G$ is $\alpha$-divergence. A similar argument can be applied to Rényi-divergence and reverse KL divergence. $\qquad \square$

### F.12 PROOF OF LEMMA F.1

*Proof of Lemma F.1.* Since $\frac{1}{\kappa s + 2\theta}$ is a decreasing function for $s \geq 0$, for all $s \geq 1$,

$$
\frac{1}{\kappa s + 2\theta} \leq \int_{s-1}^{s} \frac{1}{\kappa x + 2\theta} dx.
$$

Using this inequality, we can upper bound the sum as follows.

$$
\begin{aligned}
\sum_{s=0}^{t} \frac{1}{\kappa s + 2\theta} &= \frac{1}{2\theta} + \sum_{s=1}^{t} \frac{1}{\kappa s + 2\theta} \\
&\leq \frac{1}{2\theta} + \sum_{s=1}^{t} \int_{s-1}^{s} \frac{1}{\kappa x + 2\theta} dx \\
&= \frac{1}{2\theta} + \int_{0}^{t} \frac{1}{\kappa x + 2\theta} dx \\
&= \frac{1}{2\theta} + \frac{1}{\kappa} \int_{0}^{t} \frac{1}{x + \frac{2\theta}{\kappa}} dx \\
&= \frac{1}{2\theta} + \frac{1}{\kappa} \int_{\frac{2\theta}{\kappa}}^{t+\frac{2\theta}{\kappa}} \frac{1}{u} du \qquad \left( u = x + \frac{2\theta}{\kappa} \right) \\
&= \frac{1}{2\theta} + \frac{1}{\kappa} \ln \left( \frac{\kappa}{2\theta} t + 1 \right).
\end{aligned}
$$

This concludes the proof. $\qquad\qquad\square$

### F.13 PROOF OF LEMMA F.2

*Proof of Lemma F.2.* From the first-order optimality condition for $\pi^{\mu,\sigma^k}$ and $\pi^{\mu,\sigma^{k-1}}$, we have for any $k \geq 1$:

$$
\langle \nabla_{\pi_i} v_i(\pi_i^{\mu,\sigma^k}, \pi_{-i}^{\mu,\sigma^k}) - \mu \left( \pi_i^{\mu,\sigma^k} - \sigma_i^k \right), \sigma_i^k - \pi_i^{\mu,\sigma^k} \rangle \leq 0,
$$
$$
\langle \nabla_{\pi_i} v_i(\pi_i^{\mu,\sigma^{k-1}}, \pi_{-i}^{\mu,\sigma^{k-1}}) - \mu \left( \pi_i^{\mu,\sigma^{k-1}} - \sigma_i^{k-1} \right), \pi_i^{\mu,\sigma^k} - \pi_i^{\mu,\sigma^{k-1}} \rangle \leq 0.
$$

Summing up these inequalities yields:

$$
\begin{aligned}
0 \geq & \sum_{i=1}^{N} \langle \nabla_{\pi_i} v_i(\pi_i^{\mu,\sigma^k}, \pi_{-i}^{\mu,\sigma^k}) - \mu \left( \pi_i^{\mu,\sigma^k} - \sigma_i^k \right), \sigma_i^k - \pi_i^{\mu,\sigma^k} \rangle \\
& + \sum_{i=1}^{N} \langle \nabla_{\pi_i} v_i(\pi_i^{\mu,\sigma^{k-1}}, \pi_{-i}^{\mu,\sigma^{k-1}}) - \mu \left( \pi_i^{\mu,\sigma^{k-1}} - \sigma_i^{k-1} \right), \pi_i^{\mu,\sigma^k} - \pi_i^{\mu,\sigma^{k-1}} \rangle \\
= & \sum_{i=1}^{N} \langle \nabla_{\pi_i} v_i(\pi_i^{\mu,\sigma^k}, \pi_{-i}^{\mu,\sigma^k}), \pi_i^{\mu,\sigma^{k-1}} - \pi_i^{\mu,\sigma^k} \rangle + \sum_{i=1}^{N} \langle \nabla_{\pi_i} v_i(\pi_i^{\mu,\sigma^k}, \pi_{-i}^{\mu,\sigma^k}), \sigma_i^k - \pi_i^{\mu,\sigma^{k-1}} \rangle + \mu \| \pi^{\mu,\sigma^k} - \sigma^k \|^2 \\
& + \sum_{i=1}^{N} \langle \nabla_{\pi_i} v_i(\pi_i^{\mu,\sigma^{k-1}}, \pi_{-i}^{\mu,\sigma^{k-1}}), \pi_i^{\mu,\sigma^k} - \pi_i^{\mu,\sigma^{k-1}} \rangle + \mu \sum_{i=1}^{N} \langle \sigma_i^{k-1} - \pi_i^{\mu,\sigma^{k-1}}, \pi_i^{\mu,\sigma^k} - \pi_i^{\mu,\sigma^{k-1}} \rangle \\
= & \sum_{i=1}^{N} \langle \nabla_{\pi_i} v_i(\pi_i^{\mu,\sigma^k}, \pi_{-i}^{\mu,\sigma^k}) - \nabla_{\pi_i} v_i(\pi_i^{\mu,\sigma^{k-1}}, \pi_{-i}^{\mu,\sigma^{k-1}}), \pi_i^{\mu,\sigma^{k-1}} - \pi_i^{\mu,\sigma^k} \rangle \\
& + \sum_{i=1}^{N} \langle \nabla_{\pi_i} v_i(\pi_i^{\mu,\sigma^k}, \pi_{-i}^{\mu,\sigma^k}), \sigma_i^k - \pi_i^{\mu,\sigma^{k-1}} \rangle + \mu \| \pi^{\mu,\sigma^k} - \sigma^k \|^2 + \mu \sum_{i=1}^{N} \langle \sigma_i^{k-1} - \pi_i^{\mu,\sigma^{k-1}}, \pi_i^{\mu,\sigma^k} - \pi_i^{\mu,\sigma^{k-1}} \rangle \\
\geq & \sum_{i=1}^{N} \langle \nabla_{\pi_i} v_i(\pi_i^{\mu,\sigma^k}, \pi_{-i}^{\mu,\sigma^k}), \sigma_i^k - \pi_i^{\mu,\sigma^{k-1}} \rangle + \mu \| \pi^{\mu,\sigma^k} - \sigma^k \|^2 + \mu \sum_{i=1}^{N} \langle \sigma_i^{k-1} - \pi_i^{\mu,\sigma^{k-1}}, \pi_i^{\mu,\sigma^k} - \pi_i^{\mu,\sigma^{k-1}} \rangle,
\end{aligned}
$$

where the last inequality follows from (1). Then, since

$$
\langle \sigma_i^{k-1} - \pi_i^{\mu,\sigma^{k-1}}, \pi_i^{\mu,\sigma^{k-1}} - \pi_i^{\mu,\sigma^k} \rangle = \frac{\| \pi_i^{\mu,\sigma^k} - \sigma_i^{k-1} \|^2}{2} - \frac{\| \pi_i^{\mu,\sigma^k} - \pi_i^{\mu,\sigma^{k-1}} \|^2}{2} - \frac{\| \pi_i^{\mu,\sigma^{k-1}} - \sigma_i^{k-1} \|^2}{2},
$$

we have:

$$0 \geq \sum_{i=1}^{N} \langle \nabla_{\pi_i} v_i(\pi_i^{\mu,\sigma^k}, \pi_{-i}^{\mu,\sigma^k}), \sigma_i^k - \pi_i^{\mu,\sigma^{k-1}} \rangle + \mu \|\pi^{\mu,\sigma^k} - \sigma^k\|^2$$

$$+ \frac{\mu}{2} \sum_{i=1}^{N} \langle \sigma_i^{k-1} - \pi_i^{\mu,\sigma^{k-1}}, \pi_i^{\mu,\sigma^k} - \pi_i^{\mu,\sigma^{k-1}} \rangle + \frac{\mu}{2} \sum_{i=1}^{N} \langle \sigma_i^{k-1} - \pi_i^{\mu,\sigma^{k-1}}, \pi_i^{\mu,\sigma^k} - \pi_i^{\mu,\sigma^{k-1}} \rangle$$

$$\geq \sum_{i=1}^{N} \langle \nabla_{\pi_i} v_i(\pi_i^{\mu,\sigma^k}, \pi_{-i}^{\mu,\sigma^k}), \sigma_i^k - \pi_i^{\mu,\sigma^{k-1}} \rangle + \mu \|\pi^{\mu,\sigma^k} - \sigma^k\|^2$$

$$- \frac{\mu}{4} \left( \|\sigma^{k-1} - \pi^{\mu,\sigma^{k-1}}\|^2 + \|\pi^{\mu,\sigma^k} - \pi^{\mu,\sigma^{k-1}}\|^2 - \|\pi^{\mu,\sigma^k} - \pi^{\mu,\sigma^{k-1}} + \sigma^{k-1} - \pi^{\mu,\sigma^{k-1}}\|^2 \right)$$

$$- \frac{\mu}{4} \left( \|\pi^{\mu,\sigma^k} - \sigma^{k-1}\|^2 - \|\pi^{\mu,\sigma^k} - \pi^{\mu,\sigma^{k-1}}\|^2 - \|\pi^{\mu,\sigma^{k-1}} - \sigma^{k-1}\|^2 \right)$$

$$\geq \sum_{i=1}^{N} \langle \nabla_{\pi_i} v_i(\pi_i^{\mu,\sigma^k}, \pi_{-i}^{\mu,\sigma^k}), \sigma_i^k - \pi_i^{\mu,\sigma^{k-1}} \rangle + \mu \|\pi^{\mu,\sigma^k} - \sigma^k\|^2 - \frac{\mu}{4} \|\pi^{\mu,\sigma^k} - \sigma^{k-1}\|^2$$

$$\geq \sum_{i=1}^{N} \langle \nabla_{\pi_i} v_i(\pi_i^{\mu,\sigma^k}, \pi_{-i}^{\mu,\sigma^k}), \sigma_i^k - \pi_i^{\mu,\sigma^{k-1}} \rangle + \mu \|\pi^{\mu,\sigma^k} - \sigma^k\|^2 - \frac{\mu}{2} \|\pi^{\mu,\sigma^k} - \sigma^k\|^2 - \frac{\mu}{2} \|\sigma^k - \sigma^{k-1}\|^2$$

$$= \sum_{i=1}^{N} \langle \nabla_{\pi_i} v_i(\pi_i^{\mu,\sigma^k}, \pi_{-i}^{\mu,\sigma^k}), \sigma_i^k - \pi_i^{\mu,\sigma^{k-1}} \rangle + \frac{\mu}{2} \|\pi^{\mu,\sigma^k} - \sigma^k\|^2 - \frac{\mu}{2} \|\sigma^k - \sigma^{k-1}\|^2,$$

where the third inequality follows from $(a+b)^2 \leq 2(a^2 + b^2)$ for $a, b \in \mathbb{R}$. Thus,

$$\|\pi^{\mu,\sigma^k} - \sigma^k\|^2 \leq \|\sigma^k - \sigma^{k-1}\|^2 + \frac{2}{\mu} \sum_{i=1}^{N} \langle \nabla_{\pi_i} v_i(\pi_i^{\mu,\sigma^k}, \pi_{-i}^{\mu,\sigma^k}), \pi_i^{\mu,\sigma^{k-1}} - \sigma_i^k \rangle$$

$$\leq \|\sigma^k - \sigma^{k-1}\|^2 + \frac{2}{\mu} \|\pi^{\mu,\sigma^{k-1}} - \sigma^k\| \sqrt{\sum_{i=1}^{N} \|\nabla_{\pi_i} v_i(\pi_i^{\mu,\sigma^k}, \pi_{-i}^{\mu,\sigma^k})\|^2}$$

$$\leq \|\sigma^k - \sigma^{k-1}\|^2 + \frac{2\zeta}{\mu} \|\pi^{\mu,\sigma^{k-1}} - \sigma^k\|.$$

$$\square$$

### F.14    PROOF OF LEMMA F.3

*Proof of Lemma F.3.* From the first-order optimality condition for $p^{k+1}$, we have:

$$\langle \nabla_{\pi_i} v_i(\pi_i^{\mu,\sigma^k}, \pi_{-i}^{\mu,\sigma^k}) - \mu \left( \pi_i^{\mu,\sigma^k} - \sigma_i^k \right), \pi_i^* - \pi_i^{\mu,\sigma^k} \rangle \leq 0.$$

Thus, from the three-point identity $2\langle a - b, c - a \rangle = \|b - c\|^2 - \|a - b\|^2 - \|a - c\|^2$ and Young's inequality:

$$\sum_{i=1}^{N} \langle \nabla_{\pi_i} v_i(\pi_i^{\mu,\sigma^k}, \pi_{-i}^{\mu,\sigma^k}), \pi_i^* - \pi_i^{\mu,\sigma^k} \rangle \leq \mu \sum_{i=1}^{N} \langle \pi_i^{\mu,\sigma^k} - \sigma_i^k, \pi_i^* - \pi_i^{\mu,\sigma^k} \rangle$$

$$= \frac{\mu}{2} \|\pi^* - \sigma^k\|^2 - \frac{\mu}{2} \|\pi^{\mu,\sigma^k} - \sigma^k\|^2 - \frac{\mu}{2} \|\pi^* - \pi^{\mu,\sigma^k}\|^2$$

$$= \frac{\mu}{2} \|\pi^* - \sigma^k\|^2 - \frac{\mu}{2} \|\pi^{\mu,\sigma^k} - \sigma^k\|^2 - \frac{\mu}{2} \|\pi^* - \sigma^{k+1} + \sigma^{k+1} - \pi^{\mu,\sigma^k}\|^2$$

$$= \frac{\mu}{2} \|\pi^* - \sigma^k\|^2 - \frac{\mu}{2} \|\pi^{\mu,\sigma^k} - \sigma^k\|^2 - \frac{\mu}{2} \|\pi^* - \sigma^{k+1}\|^2 - \frac{\mu}{2} \|\sigma^{k+1} - \pi^{\mu,\sigma^k}\|^2 - \mu \langle \pi^* - \sigma^{k+1}, \sigma^{k+1} - \pi^{\mu,\sigma^k} \rangle$$

$$\leq \frac{\mu}{2} \|\pi^* - \sigma^k\|^2 - \frac{\mu}{2} \|\pi^{\mu,\sigma^k} - \sigma^k\|^2 - \frac{\mu}{2} \|\pi^* - \sigma^{k+1}\|^2 + \mu \|\pi^* - \sigma^{k+1}\| \|\sigma^{k+1} - \pi^{\mu,\sigma^k}\|$$

$$\leq \frac{\mu}{2} \|\pi^* - \sigma^k\|^2 - \frac{\mu}{2} \|\pi^{\mu,\sigma^k} - \sigma^k\|^2 - \frac{\mu}{2} \|\pi^* - \sigma^{k+1}\|^2 + \frac{\mu}{64K^3} \|\pi^* - \sigma^{k+1}\|^2 + \frac{32\mu K^3}{2} \|\sigma^{k+1} - \pi^{\mu,\sigma^k}\|^2.$$

Here, since $T_\sigma = \max(\frac{3}{\ln c}\ln K + \frac{\ln 64}{\ln c}, 1)$, we have $c^{2T_\sigma} \geq 64c^{\frac{3}{\ln c}\ln K} = 64K^3$. Therefore, we get:

$$\sum_{i=1}^{N}\langle\nabla_{\pi_i}v_i(\pi_i^{\mu,\sigma^k}, \pi_{-i}^{\mu,\sigma^k}), \pi_i^* - \pi_i^{\mu,\sigma^k}\rangle$$

$$\leq \frac{\mu}{2}\|\pi^* - \sigma^k\|^2 - \frac{\mu}{2}\|\pi^{\mu,\sigma^k} - \sigma^k\|^2 - \frac{\mu}{2}\|\pi^* - \sigma^{k+1}\|^2 + \frac{\mu}{64K^3}\|\pi^* - \sigma^{k+1}\|^2 + \frac{32\mu K^3}{2}\|\sigma^k - \pi^{\mu,\sigma^k}\|^2\left(\frac{1}{c}\right)^{2T_\sigma}$$

$$\leq \frac{\mu}{2}\|\pi^* - \sigma^k\|^2 - \frac{\mu}{2}\|\pi^{\mu,\sigma^k} - \sigma^k\|^2 - \frac{\mu}{2}\|\pi^* - \sigma^{k+1}\|^2 + \frac{\mu}{64K^3}\|\pi^* - \sigma^{k+1}\|^2 + \frac{\mu}{4}\|\sigma^k - \pi^{\mu,\sigma^k}\|^2$$

$$= \frac{\mu}{2}\|\pi^* - \sigma^k\|^2 - \frac{\mu}{2}\|\pi^* - \sigma^{k+1}\|^2 - \frac{\mu}{4}\|\pi^{\mu,\sigma^k} - \sigma^k\|^2 + \frac{\mu}{64K^3}\|\pi^* - \sigma^{k+1}\|^2.$$

Summing up this inequality from $k = 0$ to $K - 1$ yields:

$$\frac{\mu}{2}\|\pi^* - \sigma^0\|^2 + \frac{\mu}{64K^3}\sum_{k=0}^{K-1}\|\pi^* - \sigma^{k+1}\|^2 - \frac{\mu}{4}\sum_{k=0}^{K-1}\|\pi^{\mu,\sigma^k} - \sigma^k\|^2$$

$$\geq \sum_{k=0}^{K-1}\sum_{i=1}^{N}\langle\nabla_{\pi_i}v_i(\pi_i^{\mu,\sigma^k}, \pi_{-i}^{\mu,\sigma^k}), \pi_i^* - \pi_i^{\mu,\sigma^k}\rangle$$

$$\geq \sum_{k=0}^{K-1}\sum_{i=1}^{N}\langle\nabla_{\pi_i}v_i(\pi_i^*, \pi_{-i}^*), \pi_i^* - \pi_i^{\mu,\sigma^k}\rangle$$

$$\geq 0.$$

Then, from Cauchy–Schwarz inequality, we have:

$$\frac{\mu}{4}\sum_{k=0}^{K-1}\|\pi^{\mu,\sigma^k} - \sigma^k\|^2$$

$$\leq \frac{\mu}{2}\|\pi^* - \sigma^0\|^2 + \frac{\mu}{64K^3}\sum_{k=0}^{K-1}\|\pi^* - \sigma^{k+1}\|^2$$

$$\leq \frac{\mu}{2}\|\pi^* - \sigma^0\|^2 + \frac{\mu}{64K^3}\sum_{k=0}^{K-1}\left(\sum_{\tau=0}^{k}\|\sigma^{\tau+1} - \sigma^\tau\| + \|\pi^* - \sigma^0\|\right)^2$$

$$\leq \frac{\mu}{2}\|\pi^* - \sigma^0\|^2 + \frac{\mu}{32K^3}\sum_{k=0}^{K-1}\left(\left(\sum_{\tau=0}^{k}\|\sigma^{\tau+1} - \sigma^\tau\|\right)^2 + \|\pi^* - \sigma^0\|^2\right)$$

$$\leq \frac{\mu}{2}\|\pi^* - \sigma^0\|^2 + \frac{\mu}{32K^3}\sum_{k=0}^{K-1}\left(K\sum_{\tau=0}^{k}\|\sigma^{\tau+1} - \sigma^\tau\|^2 + \|\pi^* - \sigma^0\|^2\right)$$

$$\leq \frac{\mu}{2}\|\pi^* - \sigma^0\|^2 + \frac{\mu}{32K^3}\left(K^2\sum_{k=0}^{K-1}\|\sigma^{k+1} - \sigma^k\|^2 + K\|\pi^* - \sigma^0\|^2\right)$$

$$\leq \frac{\mu}{2}\|\pi^* - \sigma^0\|^2 + \frac{\mu}{32K^3}\left(K^2\sum_{k=0}^{K-1}\left(\|\sigma^{k+1} - \pi^{\mu,\sigma^k}\| + \|\pi^{\mu,\sigma^k} - \sigma^k\|\right)^2 + K\|\pi^* - \sigma^0\|^2\right).$$

By applying $\|\pi^{\mu,\sigma^k} - \sigma^{k+1}\| \leq \|\pi^{\mu,\sigma^k} - \sigma^k\|\left(\frac{1}{c}\right)^{T_\sigma}$ to the above inequality, we get:

$$\frac{\mu}{4}\sum_{k=0}^{K-1}\|\pi^{\mu,\sigma^k} - \sigma^k\|^2$$

$$\leq \frac{\mu}{2}\|\pi^* - \sigma^0\|^2 + \frac{\mu}{32K^3}\left(K^2\sum_{k=0}^{K-1}\left(\frac{\|\sigma^k - \pi^{\mu,\sigma^k}\|}{c^{T_\sigma}} + \|\pi^{\mu,\sigma^k} - \sigma^k\|\right)^2 + K\|\pi^* - \sigma^0\|^2\right)$$

$$\leq \frac{\mu}{2}\|\pi^* - \sigma^0\|^2 + \frac{\mu}{32K^3}\left(K^2 \sum_{k=0}^{K-1}\left(2\|\pi^{\mu,\sigma^k} - \sigma^k\|\right)^2 + K\|\pi^* - \sigma^0\|^2\right)$$

$$= \frac{\mu}{2}\|\pi^* - \sigma^0\|^2 + \frac{\mu}{32K^3}\left(4K^2 \sum_{k=0}^{K-1}\|\pi^{\mu,\sigma^k} - \sigma^k\|^2 + K\|\pi^* - \sigma^0\|^2\right)$$

$$= \mu\left(\frac{1}{2} + \frac{1}{K^2}\right)\|\pi^* - \sigma^0\|^2 + \frac{\mu}{8K}\sum_{k=0}^{K-1}\|\pi^{\mu,\sigma^k} - \sigma^k\|^2.$$

Therefore, for $K \geq 1$, we get:

$$\sum_{k=0}^{K-1}\|\pi^{\mu,\sigma^k} - \sigma^k\|^2 \leq 16\|\pi^* - \sigma^0\|^2.$$

$\square$

### F.15    PROOF OF LEMMA F.4

*Proof of Lemma F.4.* From the first-order optimality condition for $p^{k+1}$, we have:

$$\langle \nabla_{\pi_i} v_i(\pi_i^{\mu,\sigma^k}, \pi_{-i}^{\mu,\sigma^k}) - \mu\left(\pi_i^{\mu,\sigma^k} - \sigma_i^k\right), \pi_i^* - \pi_i^{\mu,\sigma^k}\rangle \leq 0.$$

Thus, from the three-point identity $2\langle a - b, c - a\rangle = \|b - c\|^2 - \|a - b\|^2 - \|a - c\|^2$ and Young's inequality:

$$\sum_{i=1}^{N}\langle \nabla_{\pi_i} v_i(\pi_i^{\mu,\sigma^k}, \pi_{-i}^{\mu,\sigma^k}), \pi_i^* - \pi_i^{\mu,\sigma^k}\rangle \leq \mu\sum_{i=1}^{N}\langle \pi_i^{\mu,\sigma^k} - \sigma_i^k, \pi_i^* - \pi_i^{\mu,\sigma^k}\rangle$$

$$= \frac{\mu}{2}\|\pi^* - \sigma^k\|^2 - \frac{\mu}{2}\|\pi^{\mu,\sigma^k} - \sigma^k\|^2 - \frac{\mu}{2}\|\pi^* - \pi^{\mu,\sigma^k}\|^2$$

$$= \frac{\mu}{2}\|\pi^* - \sigma^k\|^2 - \frac{\mu}{2}\|\pi^{\mu,\sigma^k} - \sigma^k\|^2 - \frac{\mu}{2}\|\pi^* - \sigma^{k+1} + \sigma^{k+1} - \pi^{\mu,\sigma^k}\|^2$$

$$= \frac{\mu}{2}\|\pi^* - \sigma^k\|^2 - \frac{\mu}{2}\|\pi^{\mu,\sigma^k} - \sigma^k\|^2 - \frac{\mu}{2}\|\pi^* - \sigma^{k+1}\|^2 - \frac{\mu}{2}\|\sigma^{k+1} - \pi^{\mu,\sigma^k}\|^2 - \mu\langle \pi^* - \sigma^{k+1}, \sigma^{k+1} - \pi^{\mu,\sigma^k}\rangle$$

$$\leq \frac{\mu}{2}\|\pi^* - \sigma^k\|^2 - \frac{\mu}{2}\|\pi^{\mu,\sigma^k} - \sigma^k\|^2 - \frac{\mu}{2}\|\pi^* - \sigma^{k+1}\|^2 + \mu\|\pi^* - \sigma^{k+1}\|\|\sigma^{k+1} - \pi^{\mu,\sigma^k}\|$$

$$\leq \frac{\mu}{2}\|\pi^* - \sigma^k\|^2 - \frac{\mu}{2}\|\pi^{\mu,\sigma^k} - \sigma^k\|^2 - \frac{\mu}{2}\|\pi^* - \sigma^{k+1}\|^2 + \mu \cdot \mathrm{diam}(\mathcal{X})\|\sigma^{k+1} - \pi^{\mu,\sigma^k}\|$$

Here, since $T_\sigma \geq \max(K^4, 3)$, we have $\frac{\ln T_\sigma}{T_\sigma} \leq \frac{4\ln K}{K^4}$. Therefore, we get:

$$\mathbb{E}\left[\sum_{i=1}^{N}\langle \nabla_{\pi_i} v_i(\pi_i^{\mu,\sigma^k}, \pi_{-i}^{\mu,\sigma^k}), \pi_i^* - \pi_i^{\mu,\sigma^k}\rangle \mid \sigma_k\right]$$

$$\leq \mathbb{E}\left[\frac{\mu}{2}\|\pi^* - \sigma^k\|^2 - \frac{\mu}{2}\|\pi^{\mu,\sigma^k} - \sigma^k\|^2 - \frac{\mu}{2}\|\pi^* - \sigma^{k+1}\|^2 \mid \sigma^k\right] + \mu \cdot \mathrm{diam}(\mathcal{X}) \cdot c\sqrt{\frac{\ln T_\sigma}{T_\sigma}}$$

$$\leq \mathbb{E}\left[\frac{\mu}{2}\|\pi^* - \sigma^k\|^2 - \frac{\mu}{2}\|\pi^{\mu,\sigma^k} - \sigma^k\|^2 - \frac{\mu}{2}\|\pi^* - \sigma^{k+1}\|^2 \mid \sigma^k\right] + \mu c \cdot \mathrm{diam}(\mathcal{X}) \cdot \frac{2\sqrt{\ln K}}{K^2}.$$

Summing up this inequality from $k = 0$ to $K - 1$ and taking its expectation yields:

$$\frac{\mu}{2}\|\pi^* - \sigma^0\|^2 - \mathbb{E}\left[\frac{\mu}{2}\sum_{k=0}^{K-1}\|\pi^{\mu,\sigma^k} - \sigma^k\|^2\right] + \mu c \cdot \mathrm{diam}(\mathcal{X}) \cdot \frac{2\sqrt{\ln K}}{K^2}$$

$$\geq \mathbb{E}\left[\sum_{k=0}^{K-1}\sum_{i=1}^{N}\langle \nabla_{\pi_i} v_i(\pi_i^{\mu,\sigma^k}, \pi_{-i}^{\mu,\sigma^k}), \pi_i^* - \pi_i^{\mu,\sigma^k}\rangle\right]$$

$$\geq \mathbb{E}\left[\sum_{k=0}^{K-1}\sum_{i=1}^{N}\langle \nabla_{\pi_i} v_i(\pi_i^*, \pi_{-i}^*), \pi_i^* - \pi_i^{\mu,\sigma^k}\rangle\right]$$

$$\geq 0.$$

Therefore, for $K \geq 1$, we get:

$$\mathbb{E}\left[\sum_{k=0}^{K-1} \|\pi^{\mu,\sigma^k} - \sigma^k\|^2\right] \leq \|\pi^* - \sigma^0\|^2 + \text{diam}(\mathcal{X}) \cdot \frac{4c\sqrt{\ln K}}{K^2}.$$

$\square$

### F.16  PROOF OF LEMMA F.5

*Proof of Lemma F.5.* By using the first-order optimality condition for $\sigma_i^{k+1}$, we have for all $\pi \in \mathcal{X}$:

$$\sum_{i=1}^{N}\langle \nabla_{\pi_i} v_i(\sigma_i^{k+1}, \sigma_{-i}^{k+1}) - \mu \nabla_{\pi_i} G(\sigma_i^{k+1}, \sigma_i^k), \pi_i - \sigma_i^{k+1}\rangle \leq 0,$$

and then

$$\sum_{i=1}^{N}\langle \nabla_{\pi_i} v_i(\sigma_i^{k+1}, \sigma_{-i}^{k+1}), \pi_i - \sigma_i^{k+1}\rangle \leq \mu \sum_{i=1}^{N}\langle \nabla_{\pi_i} G(\sigma_i^{k+1}, \sigma_i^k), \pi_i - \sigma_i^{k+1}\rangle.$$

When $G$ is $\alpha$-divergence, we have for all $\pi \in \mathcal{X}$:

$$\sum_{i=1}^{N}\langle \nabla_{\pi_i} G(\sigma_i^{k+1}, \sigma_i^k), \pi_i - \sigma_i^{k+1}\rangle = \frac{1}{1-\alpha}\sum_{i=1}^{N}\sum_{j=1}^{d_i}(\sigma_{ij}^{k+1} - \pi_{ij})\left(\frac{\sigma_{ij}^k}{\sigma_{ij}^{k+1}}\right)^{1-\alpha}$$

$$= \frac{1}{1-\alpha}\sum_{i=1}^{N}\sum_{j=1}^{d_i}(\sigma_{ij}^{k+1} - \pi_{ij}) = 0,$$

where we use the assumption that $\sigma^{k+1} = \sigma^k$ and $\mathcal{X}_i = \Delta^{d_i}$. Similarly, when $G$ is Rényi-divergence, we have for all $\pi \in \mathcal{X}$:

$$\sum_{i=1}^{N}\langle \nabla_{\pi_i} G(\sigma_i^{k+1}, \sigma_i^k), \pi_i - \sigma_i^{k+1}\rangle = \frac{\alpha}{1-\alpha}\sum_{i=1}^{N}\frac{1}{\sum_{j=1}^{d_i}(\sigma_{ij}^{k+1})^\alpha(\sigma_{ij}^k)^{1-\alpha}}\sum_{j=1}^{d_i}(\sigma_{ij}^{k+1} - \pi_{ij})\left(\frac{\sigma_{ij}^k}{\sigma_{ij}^{k+1}}\right)^{1-\alpha}$$

$$= \frac{\alpha}{1-\alpha}\sum_{i=1}^{N}\frac{1}{\sum_{j=1}^{d_i}(\sigma_{ij}^{k+1})^\alpha(\sigma_{ij}^k)^{1-\alpha}}\sum_{j=1}^{d_i}(\sigma_{ij}^{k+1} - \pi_{ij}) = 0.$$

Furthermore, if $G$ is reverse KL divergence, we have for all $\pi \in \mathcal{X}$:

$$\sum_{i=1}^{N}\langle \nabla_{\pi_i} G(\sigma_i^{k+1}, \sigma_i^k), \pi_i - \sigma_i^{k+1}\rangle = \sum_{i=1}^{N}\sum_{j=1}^{d_i}(\sigma_{ij}^{k+1} - \pi_{ij})\frac{\sigma_{ij}^k}{\sigma_{ij}^{k+1}}$$

$$= \sum_{i=1}^{N}\sum_{j=1}^{d_i}(\sigma_{ij}^{k+1} - \pi_{ij}) = 0,$$

Thus, we have for all $\pi \in \mathcal{X}$:

$$\sum_{i=1}^{N}\langle \nabla_{\pi_i} v_i(\sigma_i^{k+1}, \sigma_{-i}^{k+1}), \pi_i - \sigma_i^{k+1}\rangle \leq 0.$$

This is equivalent to the first-order optimality condition for $\pi^* \in \Pi^*$. Therefore, $\sigma^{k+1} = \sigma^k$ is a Nash equilibrium of the original game. $\square$

### F.17  PROOF OF LEMMA F.6

*Proof of Lemma F.6.* First, we prove the statement for $\alpha$-divergence: $G(\sigma_i^{k+1}, \sigma_i^k) = \frac{1}{\alpha(1-\alpha)}\left(1 - \sum_{j=1}^{d_i}(\sigma_{ij}^{k+1})^\alpha(\sigma_{ij}^k)^{1-\alpha}\right)$. From the definition of $\alpha$-divergence, we have for all

$\pi^* \in \Pi^*$:

$$\sum_{i=1}^{N} \langle \nabla_{\pi_i} G(\sigma_i^{k+1}, \sigma_i^k), \sigma_i^{k+1} - \pi_i^* \rangle = \frac{1}{1-\alpha} \sum_{i=1}^{N} \sum_{j=1}^{d_i} (\pi_{ij}^* - \sigma_{ij}^{k+1}) \left( \frac{\sigma_{ij}^k}{\sigma_{ij}^{k+1}} \right)^{1-\alpha}$$

$$= \frac{1}{1-\alpha} \sum_{i=1}^{N} \sum_{j=1}^{d_i} \pi_{ij}^* \left( \frac{\sigma_{ij}^k}{\sigma_{ij}^{k+1}} \right)^{1-\alpha} - \frac{1}{1-\alpha} \sum_{i=1}^{N} \sum_{j=1}^{d_i} (\sigma_{ij}^{k+1})^\alpha (\sigma_{ij}^k)^{1-\alpha}.$$

Here, when $\alpha \in (0, 1)$, we get $\sum_{j=1}^{d_i} (\sigma_{ij}^{k+1})^\alpha (\sigma_{ij}^k)^{1-\alpha} \leq 1$. Thus,

$$\sum_{i=1}^{N} \langle \nabla_{\pi_i} G(\sigma_i^{k+1}, \sigma_i^k), \sigma_i^{k+1} - \pi_i^* \rangle \geq \frac{1}{1-\alpha} \sum_{i=1}^{N} \sum_{j=1}^{d_i} \pi_{ij}^* \left( \frac{\sigma_{ij}^k}{\sigma_{ij}^{k+1}} \right)^{1-\alpha} - \frac{N}{1-\alpha}$$

$$= \frac{N}{1-\alpha} \exp \left( \ln \left( \frac{1}{N} \sum_{i=1}^{N} \sum_{j=1}^{d_i} \pi_{ij}^* \left( \frac{\sigma_{ij}^k}{\sigma_{ij}^{k+1}} \right)^{1-\alpha} \right) \right) - \frac{N}{1-\alpha}$$

$$\geq \frac{N}{1-\alpha} \exp \left( \frac{1-\alpha}{N} \sum_{i=1}^{N} \sum_{j=1}^{d_i} \pi_{ij}^* \ln \frac{\sigma_{ij}^k}{\sigma_{ij}^{k+1}} \right) - \frac{N}{1-\alpha}$$

$$= \frac{N}{1-\alpha} \exp \left( \frac{1-\alpha}{N} \left( \mathrm{KL}(\pi^*, \sigma^{k+1}) - \mathrm{KL}(\pi^*, \sigma^k) \right) \right) - \frac{N}{1-\alpha},$$

where the second inequality follows from the concavity of the $\ln(\cdot)$ function and Jensen's inequality for concave functions. Since $\ln(\cdot)$ is strictly concave, the equality holds if and only if $\sigma^{k+1} = \sigma^k$. Therefore, under the assumption that $\sigma^{k+1} \neq \sigma^k$, we get:

$$\mathrm{KL}(\pi^*, \sigma^{k+1}) - \mathrm{KL}(\pi^*, \sigma^k) < \frac{N}{1-\alpha} \ln \left( 1 + \frac{1-\alpha}{N} \sum_{i=1}^{N} \langle \nabla_{\pi_i} G(\sigma_i^{k+1}, \sigma_i^k), \sigma_i^{k+1} - \pi_i^* \rangle \right)$$

$$\leq \sum_{i=1}^{N} \langle \nabla_{\pi_i} G(\sigma_i^{k+1}, \sigma_i^k), \sigma_i^{k+1} - \pi_i^* \rangle, \tag{41}$$

where the second inequality follows from $\ln(1+x) \leq x$ for $x > -1$. From the first-order optimality condition for $\sigma_i^{k+1}$, we have for all $\pi^* \in \Pi^*$:

$$\sum_{i=1}^{N} \langle \nabla_{\pi_i} v_i(\sigma_i^{k+1}, \sigma_{-i}^{k+1}) - \mu \nabla_{\pi_i} G(\sigma_i^{k+1}, \sigma_i^k), \pi_i^* - \sigma_i^{k+1} \rangle \leq 0.$$

Then,

$$\sum_{i=1}^{N} \langle \nabla_{\pi_i} G(\sigma_i^{k+1}, \sigma_i^k), \sigma_i^{k+1} - \pi_i^* \rangle \leq \frac{1}{\mu} \sum_{i=1}^{N} \langle \nabla_{\pi_i} v_i(\sigma_i^{k+1}, \sigma_{-i}^{k+1}), \sigma_i^{k+1} - \pi_i^* \rangle$$

$$\leq \frac{1}{\mu} \sum_{i=1}^{N} \langle \nabla_{\pi_i} v_i(\pi_i^*, \pi_{-i}^*), \sigma_i^{k+1} - \pi_i^* \rangle, \tag{42}$$

where the second inequality follows from (1). Moreover, since $\pi^*$ is the Nash equilibrium, from the first-order optimality condition, we get:

$$\sum_{i=1}^{N} \langle \sigma_i^{k+1} - \pi_i^*, \nabla_{\pi_i} v_i(\pi_i^*, \pi_{-i}^*) \rangle \leq 0. \tag{43}$$

By combining (41), (42), and (43), if $\sigma^{k+1} \neq \sigma^k$, we have any $\pi^* \in \Pi^*$:

$$\mathrm{KL}(\pi^*, \sigma^{k+1}) - \mathrm{KL}(\pi^*, \sigma^k) < 0.$$

Next, we prove the statement for Rényi-divergence: $G(\sigma_i^{k+1}, \sigma_i^k) = \frac{1}{\alpha-1} \ln \left( \sum_{j=1}^{d_i} (\sigma_{ij}^{k+1})^\alpha (\sigma_{ij}^k)^{1-\alpha} \right)$. We have for all $\pi^* \in \Pi^*$:

$$\sum_{i=1}^N \langle \nabla_{\pi_i} G(\sigma_i^{k+1}, \sigma_i^k), \sigma_i^{k+1} - \pi_i^* \rangle = \frac{\alpha}{1-\alpha} \sum_{i=1}^N \frac{1}{\sum_{j=1}^{d_i} (\sigma_{ij}^{k+1})^\alpha (\sigma_{ij}^k)^{1-\alpha}} \sum_{j=1}^{d_i} (\pi_{ij}^* - \sigma_{ij}^{k+1}) \left( \frac{\sigma_{ij}^k}{\sigma_{ij}^{k+1}} \right)^{1-\alpha}$$

$$= \frac{\alpha}{1-\alpha} \sum_{i=1}^N \frac{1}{\sum_{j=1}^{d_i} (\sigma_{ij}^{k+1})^\alpha (\sigma_{ij}^k)^{1-\alpha}} \sum_{j=1}^{d_i} \pi_{ij}^* \left( \frac{\sigma_{ij}^k}{\sigma_{ij}^{k+1}} \right)^{1-\alpha} - \frac{N\alpha}{1-\alpha}.$$

Again, by using $\sum_{j=1}^{d_i} (\sigma_{ij}^{k+1})^\alpha (\sigma_{ij}^k)^{1-\alpha} \leq 1$ when $\alpha \in (0,1)$, we get:

$$\sum_{i=1}^N \langle \nabla_{\pi_i} G(\sigma_i^{k+1}, \sigma_i^k), \sigma_i^{k+1} - \pi_i^* \rangle \geq \frac{\alpha}{1-\alpha} \sum_{i=1}^N \sum_{j=1}^{d_i} \pi_{ij}^* \left( \frac{\sigma_{ij}^k}{\sigma_{ij}^{k+1}} \right)^{1-\alpha} - \frac{N\alpha}{1-\alpha}$$

$$= \frac{N\alpha}{1-\alpha} \exp \left( \ln \left( \frac{1}{N} \sum_{i=1}^N \sum_{j=1}^{d_i} \pi_{ij}^* \left( \frac{\sigma_{ij}^k}{\sigma_{ij}^{k+1}} \right)^{1-\alpha} \right) \right) - \frac{N\alpha}{1-\alpha}$$

$$\geq \frac{N\alpha}{1-\alpha} \exp \left( \frac{1-\alpha}{N} \sum_{i=1}^N \sum_{j=1}^{d_i} \pi_{ij}^* \ln \frac{\sigma_{ij}^k}{\sigma_{ij}^{k+1}} \right) - \frac{N\alpha}{1-\alpha}$$

$$= \frac{N\alpha}{1-\alpha} \exp \left( \frac{1-\alpha}{N} \left( \mathrm{KL}(\pi^*, \sigma^{k+1}) - \mathrm{KL}(\pi^*, \sigma^k) \right) \right) - \frac{N\alpha}{1-\alpha},$$

where the second inequality follows from Jensen's inequality for $\ln(\cdot)$ function. Since $\ln(\cdot)$ is strictly concave, the equality holds if and only if $\sigma^{k+1} = \sigma^k$. Therefore, under the assumption that $\sigma^{k+1} \neq \sigma^k$, we get:

$$\mathrm{KL}(\pi^*, \sigma^{k+1}) - \mathrm{KL}(\pi^*, \sigma^k) < \frac{N\alpha}{1-\alpha} \ln \left( 1 + \frac{1-\alpha}{N\alpha} \sum_{i=1}^N \langle \nabla_{\pi_i} G(\sigma_i^{k+1}, \sigma_i^k), \sigma_i^{k+1} - \pi_i^* \rangle \right)$$

$$\leq \sum_{i=1}^N \langle \nabla_{\pi_i} G(\sigma_i^{k+1}, \sigma_i^k), \sigma_i^{k+1} - \pi_i^* \rangle, \tag{44}$$

where the second inequality follows from $\ln(1+x) \leq x$ for $x > -1$. Thus, by combining (42), (43), and (44), if $\sigma^{k+1} \neq \sigma^k$, we have any $\pi^* \in \Pi^*$:

$$\mathrm{KL}(\pi^*, \sigma^{k+1}) - \mathrm{KL}(\pi^*, \sigma^k) < 0.$$

Finally, we prove the statement for reverse KL divergence: $G(\sigma_i^{k+1}, \sigma_i^k) = \sum_{j=1}^{d_i} \sigma_{ij}^k \ln \frac{\sigma_{ij}^k}{\sigma_{ij}^{k+1}}$. We have for all $\pi^* \in \Pi^*$:

$$\sum_{i=1}^N \langle \nabla_{\pi_i} G(\sigma_i^{k+1}, \sigma_i^k), \sigma_i^{k+1} - \pi_i^* \rangle = \sum_{i=1}^N \sum_{j=1}^{d_i} (\pi_{ij}^* - \sigma_{ij}^{k+1}) \frac{\sigma_{ij}^k}{\sigma_{ij}^{k+1}}$$

$$= \sum_{i=1}^N \sum_{j=1}^{d_i} \pi_{ij}^* \frac{\sigma_{ij}^k}{\sigma_{ij}^{k+1}} - N$$

$$= N \exp \left( \ln \left( \frac{1}{N} \sum_{i=1}^N \sum_{j=1}^{d_i} \pi_{ij}^* \frac{\sigma_{ij}^k}{\sigma_{ij}^{k+1}} \right) \right) - N$$

$$\geq N \exp \left( \frac{1}{N} \sum_{i=1}^N \sum_{j=1}^{d_i} \pi_{ij}^* \ln \frac{\sigma_{ij}^k}{\sigma_{ij}^{k+1}} \right) - N$$

$$= N \exp \left( \frac{1}{N} \left( \mathrm{KL}(\pi^*, \sigma^{k+1}) - \mathrm{KL}(\pi^*, \sigma^k) \right) \right) - N,$$

where the inequality follows from Jensen's inequality for $\ln(\cdot)$ function. Thus, under the assumption that $\sigma^{k+1} \neq \sigma^k$, we get:

$$\mathrm{KL}(\pi^*, \sigma^{k+1}) - \mathrm{KL}(\pi^*, \sigma^k) < N \ln \left( 1 + \frac{1}{N} \sum_{i=1}^{N} \langle \nabla_{\pi_i} G(\sigma_i^{k+1}, \sigma_i^k), \sigma_i^{k+1} - \pi_i^* \rangle \right)$$

$$\leq \sum_{i=1}^{N} \langle \nabla_{\pi_i} G(\sigma_i^{k+1}, \sigma_i^k), \sigma_i^{k+1} - \pi_i^* \rangle, \tag{45}$$

where the second inequality follows from $\ln(1 + x) \leq x$ for $x > -1$. Thus, by combining (42), (43), and (45), if $\sigma^{k+1} \neq \sigma^k$, we have any $\pi^* \in \Pi^*$:

$$\mathrm{KL}(\pi^*, \sigma^{k+1}) - \mathrm{KL}(\pi^*, \sigma^k) < 0.$$

$\square$

## G  CONVERGENCE RESULTS WITH OTHER DIVERGENCE FUNCTIONS

In this section, we establish the convergence results for our algorithm where $G$ is not squared $\ell^2$-distance, and $T_\sigma$ is sufficiently large. From Theorems 4.2 and 4.9, when $T_\sigma$ is large enough, updating $\sigma^k$ becomes equivalent to setting $\sigma^{k+1}$ to $\pi^{\mu,\sigma^k}$. First, we provide convergence results for the case where $G$ is a Bregman divergence:

**Theorem G.1.** *Assume that $G$ is a Bregman divergence $D_{\psi'}$ for some strongly convex function $\psi'$, and $\sigma^{k+1} = \pi^{\mu,\sigma^k}$ for $k \geq 0$. Then, there exists $\pi^* \in \Pi^*$ such that $\sigma^K \to \pi^*$ as $K \to \infty$.*

Next, we consider divergence functions $G$ other than Bregman divergence for games with probability simplex strategy spaces, i.e., $\mathcal{X}_i = \Delta^{d_i}$. Specifically, we provide the convergence results when $G$ is one of the following divergences; 1) $\alpha$-divergence $G(\pi_i, \sigma_i) = \frac{1}{\alpha(1-\alpha)} \left( 1 - \sum_{j=1}^{d_i} (\pi_{ij})^\alpha (\sigma_{ij})^{1-\alpha} \right)$; 2) Rényi-divergence $G(\pi_i, \sigma_i) = \frac{1}{\alpha-1} \ln \left( \sum_{j=1}^{d_i} (\pi_{ij})^\alpha (\sigma_{ij})^{1-\alpha} \right)$; 3) reverse KL divergence.

**Theorem G.2.** *Let us define $\mathcal{X}_i = \Delta^{d_i}$. Assume that $\sigma^{k+1} = \pi^{\mu,\sigma^k}$ for $k \geq 0$, and $G$ is one of the following divergence: 1) $\alpha$-divergence with $\alpha \in (0, 1)$; 2) Rényi-divergence with $\alpha \in (0, 1)$; 3) reverse KL divergence. If the initial slingshot strategy $\sigma^0$ is in the interior of $\mathcal{X}$, the sequence $\{\sigma^k\}_{k \geq 1}$ converges to the set of Nash equilibria $\Pi^*$ of the original game.*

## H  ADDITIONAL EXPERIMENTAL RESULTS AND DETAILS

### H.1  PAYOFF MATRIX IN THREE-PLAYER BIASED RPS GAME

Table 2: Three-Player Biased RPS game matrix.

|   | R | P | S |
|---|---|---|---|
| R | 0 | $-1/3$ | 1 |
| P | $1/3$ | 0 | $-1/3$ |
| S | $-1$ | $1/3$ | 0 |

### H.2  EXPERIMENTAL SETTING FOR SECTION 6

The experiments in Section 6 are conducted in Ubuntu 20.04.2 LTS with Intel(R) Core(TM) i9-10850K CPU @ 3.60GHz and 64GB RAM.

In the full feedback setting, we use a constant learning rate $\eta = 0.1$ for MWU and OMWU, and FTRL-SP in all three games. For FTRL-SP, we set $\mu = 0.1$ and $T_\sigma = 100$ for KL and reverse KL divergence perturbation, and set $\mu = 0.1$ and $T_\sigma = 20$ for squared $\ell^2$-distance perturbation. As

an exception, $\eta = 0.01$, $\mu = 1.0$, and $T_\sigma = 200$ are used for FTRL-SP with squared $\ell^2$-distance perturbation in the random payoff games with 50 actions.

For the noisy feedback setting, we use the lower learning rate $\eta = 0.01$ for all algorithms, except FTRL-SP with squared $\ell^2$-distance perturbation for the random payoff games with 50 actions. We update the slingshot strategy $\sigma^k$ every $T_\sigma = 1000$ iterations in FTRL-SP with KL and reverse KL divergence perturbation, and update it every $T_\sigma = 200$ iterations in FTRL-SP with squared $\ell^2$-distance perturbation. For FTRL-SP with $\ell^2$-distance perturbation in the random payoff games with 50 actions, we set $\eta = 0.001$ and $T_\sigma = 2000$.

### H.3 ADDITIONAL EXPERIMENTS

In this section, we compare the performance of FTRL-SP and MD-SP to MWU, OMWU, and optimistic gradient descent (OGD) (Daskalakis et al., 2018; Wei et al., 2021) in the full/noisy feedback setting. The parameter settings for MWU, OMWU, and FTRL-SP are the same as Section 6. For MD-SP, we use the squared $\ell^2$-distance and the parameter is the same as FTRL-SP with squared $\ell^2$-distance perturbation. For OGD, we use the same learning rate as FTRL-SP with squared $\ell^2$-distance perturbation.

Figure 3 shows the logarithmic exploitability of $\pi^t$ averaged over 100 instances with full feedback. We observe that FTRL-SP and MD-SP with squared $\ell^2$-distance perturbation exhibit competitive performance to OGD. The experimental results in the noisy feedback setting are presented in Figure 4. Surprisingly, in the noisy feedback setting, all FTRL-SP-based algorithms and the MD-SP-based algorithm exhibit overwhelmingly superior performance to OGD in all three games.

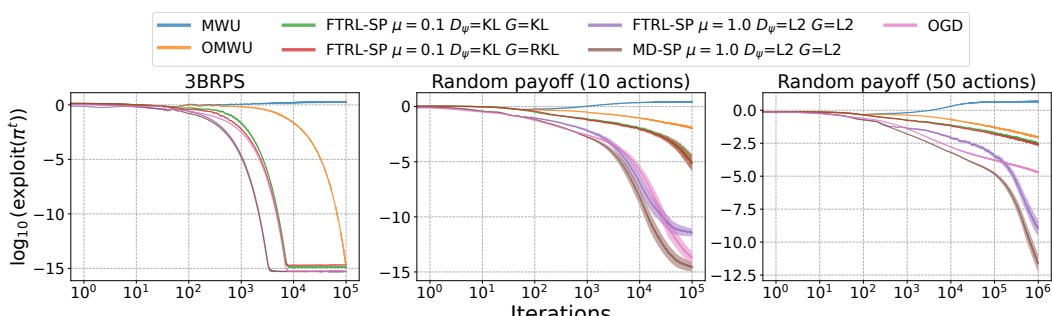

Figure 3: Exploitability of $\pi^t$ for FTRL-SP, MD-SP, MWU, OMWU, and OGD with full feedback. The shaded area represents the standard errors. Note that the KL divergence, reverse KL divergence, and squared $\ell^2$-distance are abbreviated to KL, RKL, and L2, respectively.

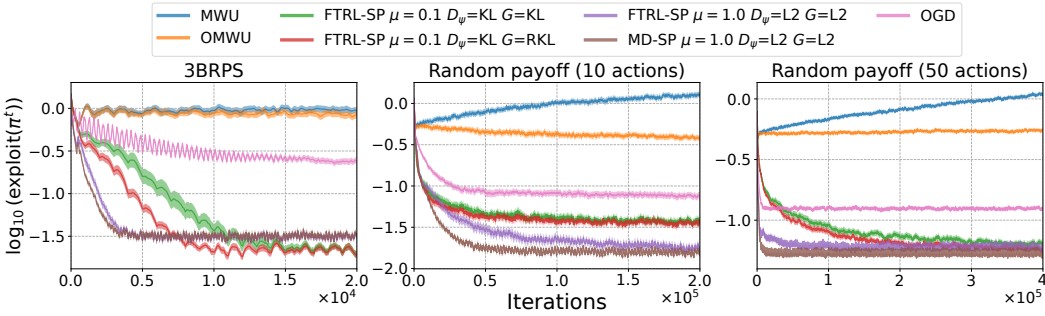

Figure 4: Exploitability of $\pi^t$ for FTRL-SP, MD-SP, MWU, OMWU, and OGD with noisy feedback. The shaded area represents the standard errors.

### H.4 COMPARISON WITH THE AVERAGED STRATEGIES OF NO-REGRET LEARNING ALGORITHMS

This section compares the last-iterate strategies $\pi^t$ of FTRL-SP and MD-SP with the average of strategies $\frac{1}{t} \sum_{\tau=1}^{t} \pi^\tau$ of MWU, regret matching (RM) (Hart & Mas-Colell, 2000), and regret matching plus (RM+) (Tammelin, 2014). The parameter settings for MWU, FTRL-SP, and MD-SP, as used in Section H.3, are maintained. Figure 5 illustrates the logarithmic exploitability averaged over 100 instances with full feedback. The results show that the last-iterate strategies of FTRL-SP and MD-SP squared $\ell^2$-distance perturbation exhibit lower exploitability than the averaged strategies of MWU, RM, and RM+.

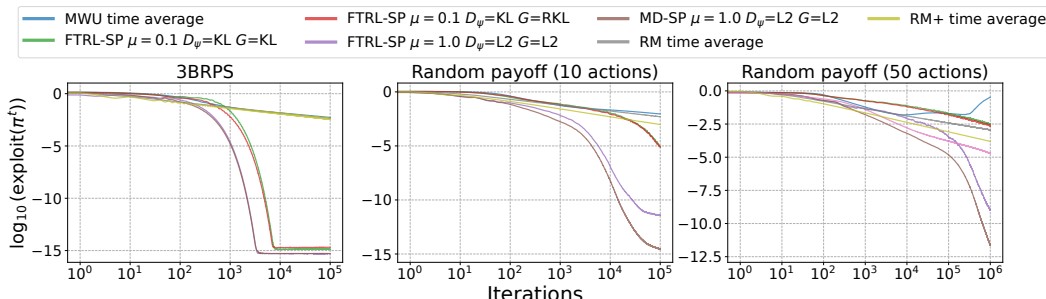

Figure 5: Comparison between exploitability of the last-iterate strategy profile of FTRL-SP, MD-SP, and the averaged strategy profile of MWU, RM, and RM+ with full feedback. The shaded area represents the standard errors.

### H.5 SENSITIVITY ANALYSIS OF UPDATE INTERVAL FOR THE SLINGSHOT STRATEGY

In this section, we investigate the performance when changing the update interval of the slingshot strategy. We vary the $T_\sigma$ of FTRL-SP with KL perturbation in 3BRPS with full feedback to be $T_\sigma \in \{10, 100, 1000, 10000\}$, and with noisy feedback to be $T_\sigma \in \{10, 100, 1000, 10000\}$. All other parameters are the same as in Section 6. Figure 6 shows the logarithmic exploitability of $\pi^t$ averaged over 100 instances in 3BRPS with full/noisy feedback. We observe that the smaller the $T_\sigma$, the faster the exploitability converges. However, if $T_\sigma$ is too small, exploitability does not converge (See $T_\sigma = 10$ with full feedback, and $T_\sigma = 100$ with noisy feedback in Figure 6).

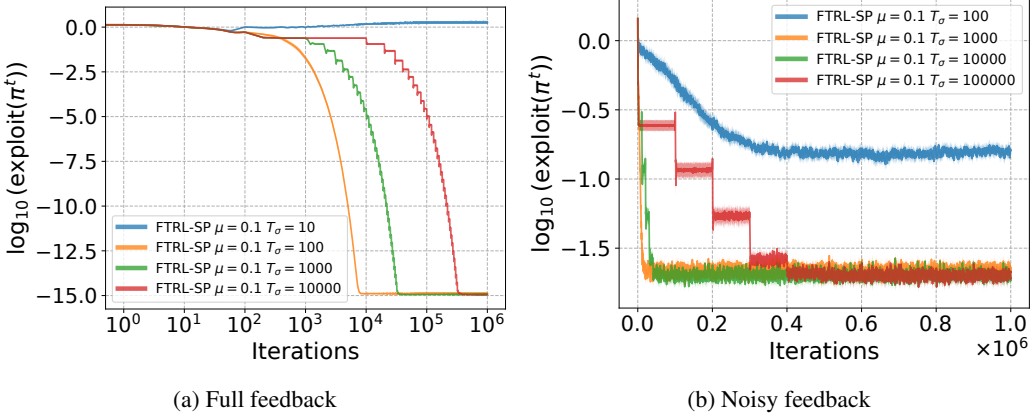

(a) Full feedback                                     (b) Noisy feedback

Figure 6: Exploitability of $\pi^t$ for FTRL-SP with varying $T_\sigma$ in 3BRPS with full/noisy feedback. The shaded area represents the standard errors.

