# OpenReview forum: "Slingshot Perturbation to Learning in Monotone Games"
_ICLR.cc/2024/Conference — Submitted to ICLR 2024_

### Official Review · Reviewer_yRC2 · 2023-10-30

**Soundness:** 3 good
**Presentation:** 4 excellent
**Contribution:** 3 good
**Rating:** 6
**Confidence:** 4

**Summary:**

This paper gives a unified framework to construct uncoupled learning algorithms that can provably converge to Nash equilibria in monotone games, for the full information feedback model and the noisy feedback model. The central idea of this framework is to introduce a slingshot strategy, so when doing (regularized) gradient descent, one does not only consider the gradient of the payoff function but also considers the gradient of the distance $G(\pi_i, \sigma_i)$ from the current strategy $\pi_i$ to the slingshot strategy $\sigma_i$. When the slingshot strategy is fixed, the algorithm can find an approximate equilibria, depending on how close $\sigma_i$ is to a Nash equilibrium. By updating the slingshot strategy periodically, the algorithm can then converge to an exact Nash equilibrium. In particular, if $G$ is the sqaured $\ell^2$-distance, then the convergence rate is $O(1/\sqrt T)$ with full information feedback and $O(T^{-1/10})$ with noisy feedback.

**Strengths:**

(1) The problem studied -- last iterate convergence to Nash equilibria via uncoupled learning in games -- is doubtless important.

(2) The framework, techniques, and result of this paper are very general: One can use any function $G$ to measure the distance from a strategy $\pi_i$ to the slingshot strategy $\sigma_i$, as long as $G$ is smooth and strongly convex. One can use any standard regularizer $\psi$. And the result holds for any monotone games. This is a nice combination and abstraction of the techniques in a wide range of previous works.

(3) Experimental results support the theoretical claims.

(4) The writing is really clear.

**Weaknesses:**

(1) The authors didn't give any impossibility results (lower bounds) on the convergence rate. I would suggest the authors to present some lower bound results to contrast with the upper bound results they give, even if the lower bounds can be from previous work.

(2) The $O(T^{-1/10})$ bound for finding exact Nash equilibrium in the noisy feedback case (Theorem 5.2) does not seem to be tight. It would be great if the authors can provide some discussion on the tightness of this result.

(3) While the convergence rates to approximate equilibria (Section 4) are presented for any general $G$ functions, the convergence rates for exact Nash equilibria (Section 5) are only for $G$ being the squared $\ell^2$-distance. The authors say that Theorem G.1 and G.2 are for other $G$ functions, but they are only asymptotic results, with no quantitative convergence rates. Given that the main contribution of this paper is a unified framework, I feel that a quantitative convergence rate for general $G$ function is needed.

**Questions:**

**Questions:**

(Q1) The cited related work [Cai-Luo-Wei-Zheng, 2023, Uncoupled and Convergent Learning in Two-Player Zero-Sum Markov Games] shows that the Nash equilibria of 2-player zero-sum games can be found by an uncoupled learning algorithm with bandit feedback with $O(T^{-1/8})$ convegence rate. This is better than the $O(T^{-1/10})$ bound in this paper. Of course I'm not saying [Cai-Luo-Wei-Zheng, 2023] outperforms this paper, because this paper's result holds for more general monotone games. But I wonder whether [Cai-Luo-Wei-Zheng, 2023]'s ideas can be used to improve this paper's result? In particular, improving the convergence rate or proving results for the bandit feedback case?

(Q2) Do the authors have any responses to the 3 weaknesses I mentioned above?

**Suggestion:**

(S1) Page 3, "Other notations" paragraph: "$\langle \nabla \psi(\pi_i'), \pi - \pi_i'\rangle$" should be "$\langle \nabla \psi(\pi_i'), \pi_i - \pi_i'\rangle$"

---

> ### Author Response · Authors · 2023-11-18
> **Response to Reviewer yRC2**
>
> We thank you for your positive feedback and constructive comments. The detailed answers to each of the questions can be found below.
>
> ---
>
> ### Weakness 1
>
> > (1) The authors didn't give any impossibility results (lower bounds) on the convergence rate. I would suggest the authors to present some lower bound results to contrast with the upper bound results they give, even if the lower bounds can be from previous work.
>
> ### Answer
>
> Thank you for your valuable comment regarding the tightness of our derived convergence rates. We agree that establishing lower bounds for our algorithm in both full and noisy feedback settings would be a promising future direction. In fact, to the best of our knowledge, no existing research has derived a lower bound for the noisy feedback setting. We will strive to provide a lower bound for our algorithm in full and noisy feedback settings in future work.
>
> ---
>
> ### Weakness 2
>
> > (2) The $O(T^{-1/10})$ bound for finding exact Nash equilibrium in the noisy feedback case (Theorem 5.2) does not seem to be tight. It would be great if the authors can provide some discussion on the tightness of this result.
>
> ### Answer
>
> Our convergence rate in the noisy feedback could be improved because we have roughly derived the upper bound on $\langle \nabla_{\pi_i}v_i(\pi^{\mu, \sigma^k}), \pi^{\mu, \sigma^{k-1}} - \sigma^k\rangle$ by using Cauchy–Schwarz inequality in the proof of Lemma F.2. We will mention the potential for improvement in our convergence rate in the subsequent revision.
>
> ---
>
> ### Weakness 3
>
> > (3) While the convergence rates to approximate equilibria (Section 4) are presented for any general $G$ functions, the convergence rates for exact Nash equilibria (Section 5) are only for $G$ being the squared $\ell^2$-distance. The authors say that Theorem G.1 and G.2 are for other $G$ functions, but they are only asymptotic results, with no quantitative convergence rates. Given that the main contribution of this paper is a unified framework, I feel that a quantitative convergence rate for general $G$ function is needed.
>
> ### Answer
>
> We would like to emphasize that existing studies have only provided asymptotic last-iterate convergence results for special cases of $G$, such as KL divergence [Perolat et al., 2021] or reverse KL divergence [Abe et al., 2023]. Our work significantly contributes to this field by (i) generalizing these results to a wider class of $G$ functions, and (ii) deriving a non-asymptotic last-iterate convergence rate for a specific $G$ function. We acknowledge the importance and challenge of extending Theorems 5.1 and 5.2 to general $G$ functions, which we view as an intriguing open problem for future work.
>
> ---
>
> ### Question 1
>
> > The cited related work [Cai-Luo-Wei-Zheng, 2023, Uncoupled and Convergent Learning in Two-Player Zero-Sum Markov Games] shows that the Nash equilibria of 2-player zero-sum games can be found by an uncoupled learning algorithm with bandit feedback with $O(T^{-1/8})$ convegence rate.
>
> ### Answer
>
> As you pointed out, Cai et al. [2023] have analyzed settings that are subtly different from ours, both in terms of the feedback type (bandit feedback vs noisy feedback), and the type of game (two-player zero-sum matrix games or Markov games vs $N$-player monotone games). This distinction makes it challenging to draw a straightforward comparison between the convergence rate of theirs and ours in Theorem 5.2.
>
> Moreover, the algorithm proposed by Cai et al. [2023] involves a decreasing perturbation strength $\mu_t$ over iteration $t$. However, in our method, as indicated in Theorem 4.2, a smaller $\mu$ necessitates a correspondingly smaller learning rate $\eta$. This could potentially slow down the convergence rate and make it difficult to find an appropriate learning rate. Therefore, for the sake of practicality, we have opted to maintain $\mu$ as a constant independent of $t$ in our algorithm.

---

> > ### Comment · Reviewer_yRC2 · 2023-11-22
> > **Happy with authors' rebuttal and keep rating 6**
> >
> > I am happy with the authors' response and keep a positive rating 6.

---

### Official Review · Reviewer_GKmU · 2023-10-31

**Soundness:** 3 good
**Presentation:** 4 excellent
**Contribution:** 3 good
**Rating:** 8
**Confidence:** 3

**Summary:**

The authors studied equilibrium computation in monotone games by addressing the cases where the feedback can be contaminated by some noise through the perturbation of payoffs. To this end, they focused on perturbing the payoffs based on the distance from a slingshot strategy. They characterized the convergence rate toward a near equilibrium. They proposed updating the slingshot strategy to converge exact equilibrium with last-iterate convergence guarantees.

**Strengths:**

- The entire paper (including the Appendix) is well-written.
- The balance between the preliminary information and new content is very good, which makes the paper accessible.
- Addressing noisy feedback is of interest.
- Updating the slingshot strategy for exact equilibrium convergence is interesting.

**Weaknesses:**

- In-line mathematical expressions makes the paper difficult to read.

**Questions:**

No clarification is needed.

---

> ### Author Response · Authors · 2023-11-15
> **Response to Reviewer GKmU**
>
> Many thanks for your encouraging and positive comments on our paper! We appreciate your positive remarks on the writing, the balance of content, and our approach. Your comments greatly encourage our work.

---

### Official Review · Reviewer_53rn · 2023-10-31

**Soundness:** 3 good
**Presentation:** 3 good
**Contribution:** 3 good
**Rating:** 6
**Confidence:** 3

**Summary:**

This paper considers the problem of computing Nash equilibria in monotone games. Monotone games are of utter importance in the (algorithmic) game theory and algorithms that can be executed in a rather decentralized manner and have provable efficient convergence rates are always sought after.

In particular, this work unifies pre-existing literature that relies on perturbing agents' payoffs and develops a general meta-algorithm, "FTRL with Slingshot Perturbation". This framework can retrieve multiple algorithms proposed in recent works, e.g., [1], [2], [3].

The algorithm (in the full-information setting) works as follows:
* A regularization function, $\psi$, and a regularization parameter, $\mu$, are selected. Every player initializes a "slingshot" strategy.
* For a fixed number of steps, players run an instance of the FTRL algorithm, with the spin-off of perturbing their utility gradient by adding the gradient of the Bregman distance generated by $\psi$ times the parameter $\mu$.
* Every player updates their "slingshot strategy" and returns to the second step.

The guarantees of this algorithm are:
* an exponential convergence rate of the sequence of strategies to the unique Nash equilibrium of the perturbed problem for a fixed slingshot strategy
* the convergence of the slingshot strategies (i.e., the strategies as well) to an approximate equilibrium.

Further, if the algorithm is run for an infinite number of steps, it is guaranteed to reach an exact Nash equilibrium of the unperturbed game.

The results are also extended for a meta-algorithm where Mirror Descent is used in place of FTRL.

[1] Tuyls, K., Hoen, P.J.T. and Vanschoenwinkel, B., 2006. An evolutionary dynamical analysis of multi-agent learning in iterated games.
[2] Perolat, J., Munos, R., Lespiau, J.B., Omidshafiei, S., Rowland, M., Ortega, P., Burch, N., Anthony, T., Balduzzi, D., De Vylder, B. and Piliouras, G., 2021, July. From Poincaré recurrence to convergence in imperfect information games: Finding equilibrium via regularization.
[3] Abe, Kenshi, Sakamoto, Mitsuki, Iwasaki Atsushi, 2022. Mutation-driven follow the regularized leader
for last-iterate convergence in zero-sum games.

**Strengths:**

* The paper unifies existing work and its main claims are quite general in their statement.
* The idea of the FTRL-slingshot-perturbation algorithm is natural and elegant.
* The paper is more or less self-contained.
* The authors establish convergence rates for both the full information and the noisy information feedback setting.
* The algorithm assets a simplified analysis of convergence.

**Weaknesses:**

* At first glance, the claims about "exact Nash equilibrium" computation can seem misleading.
* The algorithm is not fully decentralized. It could be possible to discuss whether the process would converge when the frequency of update of the slingshot strategy and $\mu$ varied across players.
* There already exist algorithms with last-iterate convergence and no-regret guarantees for monotone games. This work does not have any no-regret guarantees nor does it try to consider more decentralized settings (i.e., players have to change their slingshot strategy at the same time, and $\mu$ is the same for everyone as well as $\psi$ --- at least according to my understanding)

**Questions:**

* Do you think that the slingshot perturbation framework achieves no adversarial regret?
* What would happen if a fixed number of agents did not update their slingshot strategy every time or every player had their own frequency of updating the slingshot strategy?
* Would it be possible to tune $\mu$ as well to achieve better convergence rates?

---

> ### Author Response · Authors · 2023-11-15
> **Response to Reviewer 53rn**
>
> We thank you for your positive feedback and constructive comments. The detailed answers to each of the questions can be found below.
>
> ---
>
> ### Weakness 1
>
> > At first glance, the claims about "exact Nash equilibrium" computation can seem misleading.
>
> ### Answer
>
> Thanks for pointing this out to us. We have replaced the term ‘’exact Nash equilibrium’’ with ‘’Nash equilibrium of the underlying game’’ in our latest version (highlighted in red) to clarify what the term means.
>
> ---
>
> ### Weakness 2
>
> > The algorithm is not fully decentralized. It could be possible to discuss whether the process would converge when the frequency of update of the slingshot strategy and $\mu$ varied across players.
>
> ### Answer
>
> When each player may have a distinct perturbation strength $\mu_i>0$, convergence results very similar to the theorems in Section 4 are expected to be obtained. This is because strong convexity for every perturbed payoff function is still maintained. Consequently, we expect the behavior to be consistent with these theorems, substituting $\mu$ with the smallest perturbation strength $\min_{i\in [N]}\mu_i$ across all players.
>
> For the scenarios where the frequency of updates of the slingshot strategy, $T_{\sigma}$, varies among players, that is, each player updates the slingshot strategy at his or her own timing, we assume that each player independently adheres to the lower bound on $T_{\sigma}$ as defined in Theorems 5.1 or 5.2. Under this condition, we are expected to obtain similar convergence rates as in these theorems.
>
> Therefore, we believe that we can fully decentralize our algorithm with ease. We also agree that further analysis of this asymmetric setup would present an intriguing and promising avenue for future research.
>
> ---
>
> ### Question 1
>
> > Do you think that the slingshot perturbation framework achieves no adversarial regret?
>
> ### Answer
>
> Although a no-regret guarantee is not crucial for ensuring last-iterate convergence, we guess the FTRL-SP would achieve the no-regret property. This is because FTRL-SP updates the slingshot strategy so that it converges to the best response against the current strategy of the opponent. However, this does not always admit an optimal rate. We will mention this issue in the subsequent revision upon your request.
>
> ---
>
> ### Question 2
>
> > What would happen if a fixed number of agents did not update their slingshot strategy every time or every player had their own frequency of updating the slingshot strategy?
>
> ### Answer
>
> We anticipate that if some agents do not update their slingshot strategy, or if $T_{\sigma}=\infty$ for some agents, it would disrupt the last-iterate convergence property. Specifically, the strategy of the player with $T_{\sigma}=\infty$  at least would not reach a Nash equilibrium strategy.
>
> With regards to the scenario where each player has their own $T_{\sigma}$, we have addressed this in our response to Weakness 2.
>
> ---
>
> ### Question 3
>
> > Would it be possible to tune $\mu$ as well to achieve better convergence rates?
>
> ### Answer
>
> We appreciate your insightful suggestion regarding the tuning of $\mu$ for better convergence rates. Indeed, setting $\mu$ to a small value that depends on $T$ could potentially lead to improved convergence rates. However, as we have implied in Theorem 4.2, a smaller $\mu$ would necessitate a smaller learning rate $\eta$, which could make the balancing between $\mu$ and $\eta$ quite challenging. For the sake of practicality, we have opted to maintain $\mu$ as a constant independent of $T$.

---

### Official Review · Reviewer_vVKF · 2023-11-02

**Soundness:** 2 fair
**Presentation:** 2 fair
**Contribution:** 2 fair
**Rating:** 5
**Confidence:** 4

**Summary:**

The paper studies last-iterate convergence to NE in monotone games, under full but noisy feedback. The techniques used focus on regularising the payoffs by a reference strategy named "slingshot strategy". The slingshot is then updated as the current maintained strategy at exponential intervals and thus converges last-iterate to the true NE of the game rater then to the perturbed one.

**Strengths:**

The per studies the problem of finding a NE under perturbed feedbacks in an important subclass of games, which are monotone games. This class includes many important examples and thus is an important achievement. The paper is fairly well written and the results seems correct (although not surprising). The topic of the per is in line with the interests of the ICLR community

**Weaknesses:**

My first concern is with the motivation for the work. The authors rarely mention way one should consider full but noisy feedback. You only have full feedback when you posses a great understanding of the environment, but then noise wouldn't make much sense. I would get is you considered noisy bandit feedback, but the authors do not seem to discuss the problem.
Also, it is not clear if noise ruins the guarantees of the algorithms already present in the literature. Usually no-regret algorithm still converge in full feedback, to some sort of equilibria even in the presence of noisy feedback as long as the loss used are unbiased. The authors should discuss better why existing techniques fail, why now they only fail in terms of experimental evaluation.

My second, and far more pressing, concern is on the technical innovation of the work. The "slingshot" perturbation was studied extensively in recent papers, as also noted by the authors (Sokota et al. 2023, Bernasconi et al. 2022, Perlolat at al. 2021, Liu et al. 2023).
The main technical contribution of the paper is the introduction of the noise into the feedback, but it not clearly well explained why this is challenging or important and what technical novelty does the problem require.

Moreover, I'm puzzled about some results about section 5.  In particular how is it possible to have some meaningful guarantees as the gradient of $G(\pi,\sigma)$ can be arbitrary large in general. For example if $\sigma$ is close to the boundary and $G$ is the KL divergence, probably the results of section 5 only require very nice properties of $G$ and thus it limits its applicability.

Is also not clear to me why you do the slingshot update. You give guarantees on the exploitability of the last iterate strategy, but not on the convergence rate to the strategy itself, as you do in theorem 4.2 and 4.10 for approximate nash.

**Questions:**

1) Way should one care about full but noisy feedback?
2) What are the new techniques introduced here that are designed to help with noisy feedback?
3) Way don't you consider dynamic $\mu=\mu_k$ in section 5? Diminishing it every $T_\sigma$ turns would help as long as you can bound the distance (with either $G$ of $D$) between $\pi^{\mu_k}$ and $\pi^{\mu_{k+1}}$. This is a natural question that should give asymptotic rates to exact nash.
4) Do you require $G$ to have bounded gradient in section 5? Otherwise I think you cannot use some statements such as theorem 4.2 and 4.9 in section 5.
5) Have you considered using smaller and smaller values of $\mu=\mu_k$? This would create a sequence of NE equilibria that converge to the exact one. Similar techniques have been used in Bernasconi et al. 2022 and Liu et al. 2023, and I think should be discussed.
6) Way MWU does not converge in figure 1, without noise?

---

> ### Author Response · Authors · 2023-11-14
> **Response to Reviewer vVKF (1/2)**
>
> Thank you for taking the time to review our work and for providing your valuable feedback. We appreciate your insights and understand that you may have some concerns regarding certain aspects of our study. Our responses below address these concerns.
>
> ---
>
> ### Weakness 1
>
> > Usually no-regret algorithm still converge in full feedback, to some sort of equilibria even in the presence of noisy feedback as long as the loss used are unbiased. The authors should discuss better why existing techniques fail, why now they only fail in terms of experimental evaluation.
>
> ### Answer
>
> Standard no-regret algorithms like MWU converge to a Nash equilibrium only when one averages strategies over iterations to obtain the exact equilibrium strategy, since the dynamics make an orbit around it. While this property is well-known as average-iterate convergence, we focus on last-iterate convergence, where an algorithm converges to an equilibrium without taking the average of the strategies. We hope this clarifies the difference between the concepts of average-iterate and last-iterate.
>
> Previous studies [Daskalakis and Panageas, 2019, Mertikopoulos et al., 2019, Wei et al., 2021] have shown that optimistic no-regret algorithms, such as OWMU and OGD, enjoy last-iterate convergence only with full feedback, but fail with noisy feedback [Abe et al. 2023]. Roughly speaking, the success with full feedback is due to the proper bound of the variation of the gradient feedback vectors $\sum_{i=1}^N\\|\widehat{\nabla}\_{\pi_i}v_i(\pi^t) -\widehat{\nabla}\_{\pi_i}v_i(\pi^{t-1})\\|^2$ over iterations. More formally, we define the path length of the gradient feedbacks as $\sum_{s=1}^t\sum_{i=1}^N\\|\widehat{\nabla}\_{\pi_i}v_i(\pi^s) -\widehat{\nabla}\_{\pi_i}v_i(\pi^{s-1})\\|^2$. Noisy feedback makes the path length large, leading to the failure of optimistic no-regret algorithms to converge to an equilibrium.
>
> Thus, achieving last-iterate convergence with noisy feedback is still a significantly challenging task. We will explain the intuition in our paper's Introduction or related literature section.
>
> ---
>
> ### Weakness 2
>
> > You give guarantees on the exploitability of the last iterate strategy, but not on the convergence rate to the strategy itself, as you do in theorem 4.2 and 4.10 for approximate nash.
>
> ### Answer
>
> First, we would like to clarify that the convergence of exploitability means that the last-iterate strategy itself converges to an equilibrium. Therefore, our results (Theorems 5.1 and 5.2) inherently indicate that we have provided the convergence rate of the strategy itself. Exploitability is a standard measure of proximity to a Nash equilibrium, as used in several studies [Cai et al., 2022a,b, Abe et al., 2023, Cai and Zheng, 2023].
>
> ---
>
> ### Question 1
>
> > What are the new techniques introduced here that are designed to help with noisy feedback?
>
> ### Answer
>
> The techniques devised in this paper are manifold.
>
> - We have developed a subtly different perturbation technique from the existing ones you mentioned so that it enables the FTRL or MD dynamics to converge to an approximate equilibrium in underlying games (an equilibrium in perturbed games).
> - The existing studies only show the sort of convergence only with full feedback, and we have been successful with noisy feedback. So, we are the first to establish last-iterate convergence with noisy feedback.
> - Our perturbation technique also provides a comprehensive view of payoff-regularized algorithms such as Perorat et al. [2021] and Abe et al. [2022].
> - Furthermore, the idea of the slingshot strategy update leads us to find a Nash equilibrium in underlying games (not an approximate one). In other words, it is innovative that we mix the perturbation technique with the slingshot strategy update and exhibit last-iterate convergence with both full and noisy feedback, which has not been achieved so far.

---

> ### Author Response · Authors · 2023-11-14
> **Response to Reviewer vVKF (2/2)**
>
> ### Question 2
>
> > Do you require $G$ to have bounded gradient in section 5? Otherwise I think you cannot use some statements such as theorem 4.2 and 4.9 in section 5.
>
> ### Answer
>
> No, we do **not** require the gradient of $G$ to be bounded for Theorems 5.1, 5.2, G.1, and G.2. Therefore, we believe that these theorems hold for a wide range of divergence functions.
>
> ---
>
> ### Question 3
>
> > Way don't you consider dynamic $\mu=\mu_k$ in section 5? Diminishing it every $T_{\sigma}$ turns would help as long as you can bound the distance (with either $G$ of $D$) between $\pi^{\mu_k}$ and $\pi_{\mu_{k+1}}$.
>
> ### Answer
>
> Thank you for your insightful comment! We would first like to point out that we have already given the **non-asymptotic convergence rates to exact Nash equilibria** in Theorems 5.1 and 5.2 for our approach (of course, they also mean the asymptotic rates). As in Theorem 4.2, when the perturbation strength $\mu$ is small, a correspondingly small learning rate $\eta$ should be used. Consequently, the convergence speed would slow down if we were to adopt an approach where $\mu_k$ decreases as $k$ increases, and finding a proper learning rate would be difficult. On the contrary, our slingshot update approach does not require a decaying $\mu_k$. Hence, our method does not suffer from the stringent requirement of selecting appropriate $\mu$ and $\eta$ and is more robust than decaying perturbation approaches in terms of hyperparameter settings.
>
> ---
>
> ### Question 4
>
> > Way MWU does not converge in figure 1, without noise?
>
> ### Answer
>
> Figure 1 shows the exploitability of the last-iterate strategy of each algorithm. As we previously addressed in our response to Weakness 1, MWU’s last-iterate strategy indeed fails to converge even with full feedback. This phenomenon has been widely owhebserved and discussed in several studies [Perolat et al., 2021, Abe et al., 2023, Liu et al., 2023]. Furthermore, it has even been theoretically substantiated by Mertikopoulos et al. [2018] and Bailey and Piliouras [2018].

---

> > ### Comment · Reviewer_vVKF · 2023-11-20
> >
> > After the authors' response I still have doubts about the paper's contributions.
> > 1) The algorithm converge last iterate to perturbed equilibrium, not to exact equilibrium. (Convergence to exact eq is given only in section 5 where the authors rely on some strange assumptions about the rate of change of parametric equilibria).
> > 2) Section 5 (the one in which you actually use slingshot perturbation) only works with $G(a,b)=\|a-b\|_2^2$. If the authors think that the procedure could also work for other $G$ (see response to review yRC2, weakness 3) they should prove it or tone down the achievements about that part of the paper.
> > 3) "First, we would like to clarify that the convergence of exploitability means that the last-iterate strategy itself converges to an equilibrium" This is not true. You converge last iterate to an approximate equilibrium, that has comparable exploitability as the true equilibrium, but it could be very far from the actual equilibrium, so last iterate property is less appealing that if the algorithm would converge to the true equilibrium
> > 4) In Th 5.1 and 5.2 the assumption on the difference $\pi^{\mu,\sigma^k}-\sigma^{k+1}$ seems to bypass the main difficulty of doing slingshot update, which is that that equilibria can jump from one place to another when the parameter (or losses) are changes, even slightly. It can also be that the assumption are written in a confusing way or that I'm missing something.
> > The authors should definitely discuss if these assumptions are ever met, because they seem pretty strong written as such.
> > 5) I think it would be more fair to wither consider the average policy in MWU or not to include it at all, when you known that the last iterate convergence would not converge. This is because, the presence of noise could be unknown to the user of the algorithm, but  the user would certainly know that it has to take the average strategy for MWU to work. Also it would be fair to include RM, RM+ and equivalent in the experimental evaluations.

---

> ### Author Response · Authors · 2023-11-21
> **Response to the comments of Reviewer vVKF (1/2)**
>
> Thank you for your further comments! Firstly, we would like to summarize the key points of our response to your comments as follows:
>
> - **The assumptions of Theorems 5.1 and 5.2 are naturally satisfied by Theorems 4.2 and 4.9, hence they are not strong assumptions.**
> - **Theorems 5.1 and 5.2 guarantee the last-iterate convergence to an actual Nash equilibrium, not a perturbed equilibrium.**
> - **We have also provided the last-iterate convergence results for $G$ other than squared $\ell^2$-distance in Theorems G.1 and G.2.**
>
> The details of our responses to each of your comments can be found below.
>
> ---
>
> ### Comment 1
>
> > The algorithm converge last iterate to perturbed equilibrium, not to exact equilibrium. (Convergence to exact eq is given only in section 5 where the authors rely on some strange assumptions about the rate of change of parametric equilibria).
>
> ### Answer
>
> In Section 5, we have presented Theorems 5.1 and 5.2 that ensure the last-iterate convergence to an actual equilibrium, not to a perturbed one. Your concern might be regarding the assumptions on the difference $\pi^{\mu,\sigma^k}-\sigma^{k+1}$. To clarify, we have addressed this point in our response to Comment 4, where we explain that these assumptions are generally satisfied and thus, are not strong assumptions.
>
> ---
>
> ### Comment 2
>
> > Section 5 (the one in which you actually use slingshot perturbation) only works with $G(a,b)=|a-b|_2^2$. If the authors think that the procedure could also work for other $G$ (see response to review yRC2, weakness 3) they should prove it or tone down the achievements about that part of the paper.
>
> ### Answer
>
> While our last-iterate convergence rates are indeed provided for the case where $G$ is squared $\ell^2$-distance, we have also established the asymptotic last-iterate convergence results for more general functions $G$ in Theorems G.1 and G.2 (i.e., Bregman divergence, $\alpha$-divergence, Rényi-divergence, and reverse KL divergence). Notably, these results significantly extend the scope of the asymptotic convergence results previously established for KL divergence and reverse KL divergence in existing studies [Perorat et al., 2021, Abe et al., 2022].

---

> ### Author Response · Authors · 2023-11-21
> **Response to the comments of Reviewer vVKF (2/2)**
>
> ### Comment 3
>
> > "First, we would like to clarify that the convergence of exploitability means that the last-iterate strategy itself converges to an equilibrium" This is not true. You converge last iterate to an approximate equilibrium, that has comparable exploitability as the true equilibrium, but it could be very far from the actual equilibrium, so last iterate property is less appealing that if the algorithm would converge to the true equilibrium
>
> ### Answer
>
> As indicated by Theorems 5.1 and 5.2, the exploitability of the last-iterate strategy $\pi^t$ converges $0$. Moreover, according to the definition of exploitability (as we have described in Section 2), the value of exploitability $\mathrm{exploit}(\pi)$ of a given strategy $\pi$ equals $0$ **if and only if $\pi$ is an actual Nash equilibrium in the original game**. Therefore, Theorems 5.1 and 5.2 mean that the last-iterate strategy $\pi^t$ updated by our FTRL-SP does indeed converge to an actual Nash equilibrium, not an approximate equilibrium.
>
> We will be happy if this clarifies your concerns, although we are still uncertain about why you think last-iterate could be far from the actual equilibrium. Let us know if you think the last iterate property established so far in this literature lacks appeal, please.
>
> ---
>
> ### Comment 4
>
> > In Th 5.1 and 5.2 the assumption on the difference $\pi^{\mu,\sigma^k}-\sigma^{k+1}$ seems to bypass the main difficulty of doing slingshot update, which is that that equilibria can jump from one place to another when the parameter (or losses) are changes, even slightly. It can also be that the assumption are written in a confusing way or that I'm missing something. The authors should definitely discuss if these assumptions are ever met, because they seem pretty strong written as such.
>
> ### Answer
>
> We would like to emphasize that in our slingshot strategy update framework, the perturbed equilibrium $\pi^{\mu,\sigma^k}$ at $k$-th slingshot update is not far from the next slingshot strategy $\sigma^{k+1}$. Specifically, the assumption that $\|\pi^{\mu,\sigma^k}-\sigma^{k+1}\|\leq \|\pi^{\mu,\sigma^k}-\sigma^k\|\left(\frac{1}{c}\right)^{T_{\sigma}}$ is directly satisfied when Theorem 4.2 holds. This is due to the following reasons:
>
> - Let us set $\sigma=\sigma^k$ and $\pi^0=\sigma^k$ in Theorem 4.2.
> - From this theorem, we can immediately deduce that $\|\pi^{\mu,\sigma^k}-\pi^t\| \leq \|\pi^{\mu,\sigma^k} - \sigma^k\|\left(\frac{1}{c}\right)^t$ holds for any $t\geq 1$.
> - Therefore, by taking $t=T_{\sigma}$, we obtain $\|\pi^{\mu,\sigma^k}-\pi^{T_{\sigma}}\| \leq \|\pi^{\mu,\sigma^k} - \sigma^k\|\left(\frac{1}{c}\right)^{T_{\sigma}}$.
> - Since we use $\pi^{T_{\sigma}}$ as the next slingshot strategy $\sigma^{k+1}$ in our framework, it follows that $\|\pi^{\mu,\sigma^k}-\sigma^{k+1}\|\leq \|\pi^{\mu,\sigma^k}-\sigma^k\|\left(\frac{1}{c}\right)^{T_{\sigma}}$.
>
> Hence, FTRL-SP always satisfies this assumption in the full feedback setting, and we believe that the assumption regarding the difference $\pi^{\mu,\sigma^k}-\sigma^{k+1}$ is not strong. A similar reasoning can be applied to Theorem 5.2 for the noisy feedback setting.
>
> ---
>
> ### Comment 5
>
> > I think it would be more fair to wither consider the average policy in MWU or not to include it at all, when you known that the last iterate convergence would not converge. This is because, the presence of noise could be unknown to the user of the algorithm, but the user would certainly know that it has to take the average strategy for MWU to work. Also it would be fair to include RM, RM+ and equivalent in the experimental evaluations.
>
> ### Answer
>
> Thank you for your suggestion! We will remove MWU’s results from Figures 1 and 2. Regarding your suggestion to include RM and RM+ in our experimental evaluations, we are currently conducting these experiments. As soon as the results are available, we will update our paper accordingly to provide a more comprehensive analysis.

---

> ### Author Response · Authors · 2023-11-22
> **Manuscript Revision - Experimental Comparison Included as Requested**
>
> Dear Reviewer vVKF,
>
>
> As you requested, we have included an experimental comparison with the averaged strategies of MWU, RM, and RM+ in Section H.4 of our revised manuscript (highlighted in red).
> If you're interested, feel free to take a look at these results.
>
> Sincerely,
>
> The Authors

---

### Comment · Area_Chair_CytW · 2023-11-19
**Some questions to the authors**

Dear authors,

Thank you for your timely responses to the reviewers' comments.

I would first like to ask the reviewers to go through the posted rebuttals and follow up on their comments and concerns, or ask for any further clarifications before the discussion phase closes.

In the meantime, I would also like to take this opportunity to ask some questions of my own:

1. There is a fair number of papers achieving last-iterate convergence in merely monotone games with noisy gradient feedback, see e.g., [1,2,3] below. In more detail, [1] proves last-iterate convergence to a minimum-norm Nash equilibrium in merely monotone games with access to a stochastic first order oracle, while [2] extends this to a payoff-based information structure. [3] treats certain unconstrained (non-compact) cases, but also provides convergence rates. The algorithms considered are quite close, so I am wondering about the positioning of your results relative to these prior works – especially [1,2].

2. When $G$ is the Bregman divergence induced by $\psi$, is the slingshot strategy different to a Bregman regularization step – like e.g., [2] for the Tikhonov / Euclidean case or [4,5] for general case?

3. How do the results in Section 4 relate to e.g., [6] who established both convergence and convergence rates, even for more general constraints? [In particular, [6] treats the payoff-based case, which is harder to deal with because of the increased variance of the estimator]

4. I am having trouble of coming up with $G$ functions that are not Bregman divergences. Did you provide an example that I missed? Even if so, what is the benefit of considering non-divergence $G$ functions?

Kind regards,

The AC

---

### **References**
[1] Koshal, Jayash, Angelia Nedić, and Uday V. Shanbhag. "Single timescale regularized stochastic approximation schemes for monotone Nash games under uncertainty." 49th IEEE Conference on Decision and Control (CDC). IEEE, 2010.

[2] Tatarenko, Tatiana, and Maryam Kamgarpour. "Learning Nash equilibria in monotone games." 2019 IEEE 58th Conference on Decision and Control (CDC). IEEE, 2019.

[3] Hsieh, Yu-Guan Hsieh, Iutzeler, Franck, Malick, Jérôme, and Mertikopoulos, Panayotis. "Explore aggressively, update conservatively: Stochastic extragradient methods with variable stepsize scaling." NeurIPS '20: Proceedings of the 34th International Conference on Neural Information Processing Systems, 2020.

[4] David S. Leslie and E. J. Collins, Individual Q-learning in normal form games, SIAM Journal on Control and Optimization 44 (2005), no. 2, 495–514.

[5] Coucheney, Pierre, Gaujal, Bruno, and Mertikopoulos, Panayotis. "Penalty-regulated dynamics and robust learning procedures in games." Mathematics of Operations Research 40 (2015), no. 3, 611– 633.

[6] Drusvyatskiy, Dmitriy, Maryam Fazel, and Lillian J. Ratliff. "Improved Rates for Derivative Free Gradient Play in Strongly Monotone Games." 2022 IEEE 61st Conference on Decision and Control (CDC). IEEE, 2022.

---

> ### Author Response · Authors · 2023-11-20
> **Response to Area Chair CytW**
>
> We thank you for your insightful questions. The detailed answers to each of the questions can be found below.
>
> ---
>
> ### Question 1
>
> > I am wondering about the positioning of your results relative to these prior works – especially [1,2]
>
> ### Answer
>
> Koshal et al. [2010] have provided last-iterate convergence results for the noisy feedback setting in monotone games. However, their results are limited to **asymptotic convergence**. Similarly, Tatarenko and Kamgarpour [2019] have also demonstrated asymptotic convergence in a payoff-based setting, where only the exact utility value of sampled actions is observable, which is also referred to as bandit feedback [Bravo et al., 2018]. This setting is fundamentally distinct from our noisy feedback setting.
>
> Regarding Hsieh et al. [2020], it is certain that they establish **asymptotic convergence results** for general monotone games. However, the concrete convergence rates are derived just under certain stronger assumptions, such as strongly monotone games. In contrast, as we stated in our Theorem 5.2, we have derived the **non-asymptotic convergence rates** for general monotone games.
>
> From an algorithmic standpoint, both Koshal et al. [2010] and Tatarenko and Kamgarpour [2019] have analyzed an iterative Tikhonov regularization method, where the perturbation strength $\mu$ diminishes as the iteration $t$ progresses. However, as we have implied in Theorem 4.2, a smaller $\mu$ would require a smaller learning rate $\eta$. This could potentially decelerate the convergence rate and complicate the task of finding an appropriate $\eta$. For practicality, we have opted to keep $\mu$ as a constant independent of $t$ in our algorithm.
>
> ---
>
> ### Question 2
>
> > When $G$ is the Bregman divergence induced by $\psi$, is the slingshot strategy different to a Bregman regularization step – like e.g., [2] for the Tikhonov / Euclidean case or [4,5] for general case?
>
> ### Answer
>
> Our slingshot strategy update technique is fundamentally different from [2,4,5]. Specifically, the learning dynamics proposed by Leslie and Collins [2005] and Coucheney et al. [2015] **do not update the slingshot strategy**. Indeed, the Boltzmann Q-learning dynamics presented in Example 3.3 of our paper represents one such dynamic they proposed. Hence, a part of their dynamics is contained in our FTRL-SP.
>
> More importantly, while Leslie and Collins [2005] and Coucheney et al. [2015] have demonstrated convergence to **an approximate Nash equilibrium**, we have achieved convergence to **a Nash equilibrium of the original game** by introducing a slingshot strategy update. We believe this key distinction underscores the unique contributions of our work.
>
> ---
>
> ### Question 3
>
> > How do the results in Section 4 relate to e.g., [6] who established both convergence and convergence rates, even for more general constraints?
>
> ### Answer
>
> The study of Drusvyatskiy et al. [2022] is distinguished from ours with regard to both the types of games and feedback. They focus on **strongly monotone games**, which is a special case of our **(non-strictly)** **monotone games**, and the payoff-based setting (a.k.a. bandit feedback), which is inevitably different from our noisy feedback setting where the observed gradient vector is contaminated by additive noise.
>
> ---
>
> ### Question 4
>
> > Did you provide an example that I missed? Even if so, what is the benefit of considering non-divergence $G$ functions?
>
> ### Answer
>
> We have already provided some examples of $G$ functions that are not Bregman divergence in our paper; $\alpha$-divergence, Rényi-divergence, and reverse KL divergence [Abe et al., 2022, 2023]. Also, the detailed last-iterate convergence results for these functions can be found in Theorems G.1 and G.2.
>
> Our motivation to generalize $G$ functions stems from the observation that while Perolat et al. [2021] utilize the KL divergence, Abe et al. [2022, 2023] employ the reverse KL divergence. A unique characteristic distinguishing these studies is that the dynamics in Abe et al. [2022, 2023] is equivalent to replicator mutator dynamics [Zagorsky et al., 2013], while the FTRL dynamics is known to coincide with replicator dynamics without mutation (or perturbation, as referred to in our paper).
>
> Thus, developing a unified framework for payoff-perturbed algorithms is of significant importance. This generalization would provide us with a more profound understanding of such algorithms, particularly in terms of last-iterate convergence.
>
> ---
>
> ### Reference
> [1] Benjamin M. Zagorsky, Johannes G. Reiter, Krishnendu Chatterjee, and Martin A. Nowak. Forgiver triumphs in alternating prisoner’s dilemma. PLOS ONE, pages 1–8, 2013.

---

> > ### Comment · Area_Chair_CytW · 2023-11-20
> >
> > Thank you for your reply.
> >
> > I appreciate the explanations, but there is an important point of disconnect with your paper: In Section 4 (the longest section in your paper, and the basis for the moving-slingshot analysis of Section 5), you are considering a *fixed* slingshot, in a *perturbed* game, which is *strongly monotone* (because the perturbation is), and you are proving convergence to a *perturbed* equailibrium (essentially a quantal response equilibrium). In this regard, the results of Section 4 are much closer to the existing literature than your reply suggests, and the lack of discussion and positioning relative to these works is somewhat perplexing.
> >
> > Your answer regarding stochastic oracle feedback versus bandit feedback likewise misses the fact that the bandit case requires a more sophisticated analysis which typically subsumes the stochastic oracle analysis (to be clear, I am not saying that the bandit model subsumes the oracle model, but that the bandit analysis typically subsumes the oracle one). However, if you are looking for a recent work with stochastic oracle feedback, you might want to look into
> >
> > Huang K, Zhang S. New first-order algorithms for stochastic variational inequalities. SIAM Journal on Optimization. 2022;32(4):2745-72.
> >
> > Thank you again for your input. Regards,
> >
> > The AC

---

> ### Author Response · Authors · 2023-11-21
> **Response to the comments of Area Chair CytW**
>
> Thanks for your comments regarding the standing of Section 4 and the pointer to the related work.
>
> In Section 4, we will include the relationship between our Theorem 4.9 and the convergence results for the studies on strongly monotone games with noisy and bandit feedback [Hsieh et al., 2020, Drusvyatskiy et al., 2022, Huang and Zhang, 2022].
>
> However, we would like to clarify that Theorem 4.9 represents only a part of our contributions. In addition to this, our work also includes:
>
> - We have proposed a unified framework covering a wide array of existing payoff-regularized algorithms.
> - Our algorithm achieves **last-iterate convergence with rates in general (non-strictly) monotone games with both full and noisy feedback**, by introducing the slingshot strategy update.

---

### Meta-Review · Area_Chair_CytW · 2023-12-06

**Metareview:**

This paper is examining the problem of equilibrium learning in monotone games with a "slingshot perturbation" method. The main challenge in this class of games - as opposed to strictly or strongly monotone games - is that standard gradient methods fail to converge, even with deterministic gradient feedback. To address this, the authors introduce a strongly convex regularization penalty term - the so-called "slingshot perturbation" - which allows them to obtain convergence to a near-equilibrium point. The accuracy of the approximation depends on the weight of the regularization penalty and the position of the "slingshot" point, so, by reducing the weight and updating the anchor of the regularization term appropriately, the authors ultimately achieve convergence in the class of merely monotone games.

This paper was discussed extensively, and opinions oscillated between borderline positive and negative. On the positive side, the paper's premise is an interesting one, as achieving convergence in merely monotone games with noisy gradient feedback is a challenging problem. On the other hand, the discussion also raised several concerns:
1. The "slingshot perturbation" described in Equation (3) essentially amounts to introducing a strongly convex penalty to the players' utility functions. This is a textbook technique in optimization and game theory, described already in the classical textbooks of Van Damme (1987) and Facchinei-Pang (2003), and going back to several decades before. The specific algorithm considered in Section 4 by the authors has already been examined in the literature, by Koshal et al. (2010) and Tatarenko and Kamgarpour (2019) in the Euclidean case, Leslie and Collins (2005), Tuyls et al. (2006), and Coucheney et al. (2015) for general perturbation functions in finite games, and many others. As such, referencing only a few very recent works, using a new name for a classical concept and ignoring the existing corpus of literature on regularization leads to an unbalanced presentation which makes it quite difficult to appreciate the paper's contributions.

2. The "exploitability gap" considered by the authors is the classical Nikaidô-Isoda gap function, which goes back all the way to the 1950s. Again, this creates confusion, as it makes the positioning of the paper's contributions and results in the context of existing work quite difficult.

3. The results of Section 4 concern a fixed regularization weight, so the problem becomes strongly monotone in this case. As a result, the analysis follows the standard path of first-order methods applied to strongly monotone problems, but the authors do not make this clear, leading again to an unbalanced presentation that occludes what's going on.

4. In Section 5, the requirement of a fixed regularization penalty function is lifted. However, the authors' results become less transparent as a result, and the exact requirements on the perturbation schedule in Theorems 5.1 and 5.2 are quite opaque. [And a question also remains on how this relates to vanishing regularization schedules used e.g., by Tatarenko and Kamgarpour]

Overall, even though this paper contains interesting ideas, the current treatment is not yet at the point where it can be accepted "as is" and a fresh round of reviews would be required before considering this question again. For this reason, a decision was reached to make a "reject" recommendation at this stage while encouraging the authors to revise their paper from the ground up and resubmit.

**Justification For Why Not Higher Score:**

The paper has some interesting contributions but the positioning creates an unbalanced presentation (especially in Section 4).

**Justification For Why Not Lower Score:**

N/A

---

### Decision · Program_Chairs · 2024-01-16

Reject